# Dynamical versus Bayesian Phase Transitions in a Toy Model of Superposition

## Abstract

We investigate phase transitions in a Toy Model of Superposition (TMS) (Elhage et al., 2022) using Singular Learning Theory (SLT). We derive a closed formula for the theoretical loss and, in the case of two hidden dimensions, discover that regular $k$-gons are critical points. We present supporting theory indicating that the local learning coefficient (a geometric invariant) of these $k$-gons determines phase transitions in the Bayesian posterior as a function of training sample size. We then show empirically that the same $k$-gon critical points also determine the behavior of SGD training. The picture that emerges adds evidence to the conjecture that the SGD learning trajectory is subject to a sequential learning mechanism. Specifically, we find that the learning process in TMS, be it through SGD or Bayesian learning, can be characterized by a journey through parameter space from regions of high loss and low complexity to regions of low loss and high complexity.

## 1 Introduction

The apparent simplicity of the Toy Model of Superposition (TMS) proposed in Elhage et al. (2022) conceals a remarkably intricate *phase structure*. During training, a plateau in the loss is often followed by a sudden discrete drop, suggesting some development in the network's internal structure. To shed light on these transitions and their significance, this paper examines the dynamical transitions in TMS during SGD training, connecting them to phase transitions of the Bayesian posterior with respect to sample size $n$. While the former transitions have been observed in several recent works in deep learning (Olsson et al., 2022; McGrath et al., 2022; Wei et al., 2022a), their formal status has remained elusive. In contrast, phase transitions of the Bayesian posterior are mathematically well-defined in Singular Learning Theory (SLT) (Watanabe, 2009).

Using SLT, we can show formally that the Bayesian posterior is subject to an *internal model selection* mechanism in the following sense: the posterior prefers, for small training sample size $n$, critical points with low complexity but potentially high loss. The opposite is true for high $n$ where the posterior prefers low loss critical points at the cost of higher complexity. The measure of complexity here is very specific: it is the *local learning coefficient*, $\lambda$, of the critical points, first alluded to by Watanabe (2009, §7.6) and clarified recently in Lau et al. (2023). We can think of this internal model selection as a discrete dynamical process: at various critical sample sizes the posterior concentration "jumps" from one region $\mathcal{W}_\alpha$ of parameter space to another region $\mathcal{W}_\beta$. We refer to an event of this kind as a *Bayesian phase transition $\alpha \to \beta$*.

For the TMS model with two hidden dimensions we show that these Bayesian phase transitions actually occur and do so between phases dominated by weight configurations representing regular polygons (termed here $k$-gons). The main result of SLT, the asymptotic expansion of the free energy (Watanabe, 2018), predicts phase transitions as a function of the loss and local learning coefficient of each phase. For TMS, we are in the fortunate position of being able to derive theoretically the exact local learning coefficient of the $k$-gons which are most commonly encountered during MCMC sampling of the posterior, and thereby verify that the mathematical theory correctly predicts the empirically observed phases and phase transitions. Altogether, this forms a mathematically well-founded toolkit for reasoning about phase transitions in the Bayesian posterior of TMS.

It has been observed empirically in TMS that SGD training also undergoes "phase transitions" (Elhage et al., 2022) in the sense that we often see steady plateaus in the training (and test) loss separated by sudden transitions, associated with geometric transformations in the configuration of the columns

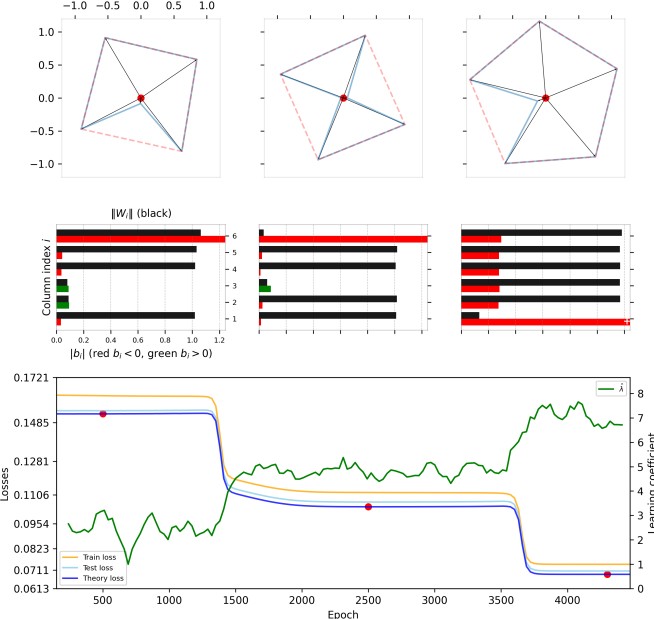

Figure 1: For $r = 2$ hidden dimensions and $c = 6$ feature dimensions, neural network parameters $w = (W, b)$ for TMS with high loss are tolerated at the beginning of training because they have low complexity (low local learning coefficient $\hat{\lambda}$) but at the end of training low loss solutions are attractive despite their high complexity (high $\hat{\lambda}$). *First row*: columns $W_i \in \mathbb{R}^2$ of $W$ are shown as black arrows for three snapshots (timestamps shown as red dots in the third row). Adjacent columns are connected by a blue line, the red dashed line shows the convex hull. *Second row*: for columns $W_i$ ordered $1 \le i \le 6$ from the negative $x$-axis in a counter-clockwise direction, the norm $\|W_i\|$ (black) and $|b_i|$ (red, green) are shown. A white plus sign indicates a bias that exceeds $1.25 * \max_{i \in I} \|W_i\|^2$ where $I$ is the set of all columns $i$ where $b_i \ge \|W_i\|$. All snapshots share the same axes. *Third row:* losses and local learning coefficient, with the latter smoothed over a window of size 6, where $\hat{\lambda}$ is measured every 30 epochs. For more examples see Appendix B.

of the weight matrix. Figure 1 shows a typical example. We refer to these as *dynamical transitions*. A striking pattern emerges when we observe the evolution of the loss and the estimated local learning coefficient, $\hat{\lambda}$, over the course of training: we see "opposing staircases" where each drop in the training and test loss is accompanied by a jump in the (estimated) local complexity measure. In essence, during the training process, as SGD reduces the loss, it exhibits an increasing tolerance for complex solutions. On these grounds we propose the **Bayesian antecedent hypothesis**, which says that these dynamical transitions have "standing behind them" a Bayesian phase transition.

We begin in Section 3.1 by recalling the TMS, and present a closed form for the population loss in the high sparsity limit. In our **first contribution**, we provide a partial classification of critical points of the population loss (Section 3.2) and document the local learning coefficients of several of these critical points (Section 3.3). In our **second contribution**, we experimentally verify that the main phase transition predicted by the internal model selection theory, using the theoretically derived local learning coefficients, actually takes place (Section 4.2). In Section 5 we present experimental results on dynamical transitions in TMS. Our **third contribution** is to show empirically that SGD training in TMS transitions from high-loss-low-complexity solutions to low-loss-high-complexity solutions, where complexity is measured by the estimated local learning coefficient. This provides support for our proposed relation between Bayesian and dynamical transitions (Section 5.1).

## 2 RELATED WORK

The TMS problem is, with the nonlinearity removed and varying importance factors, solved by computing principal components; it has long been understood that the learning dynamics of computing principal components is determined by a unique global minimum and a hierarchy of saddle points of decreasing loss (Baldi & Hornik, 1989), (Amari, 2016, §13.1.3). In recent decades an extensive literature has emerged on *Deep Linear Networks* (DLNs) building on these results, and applying them to explain phenomena in the development of both natural and artificial neural networks (Saxe et al., 2019). Under some hypotheses the saddles of a DLN are strict (Kawaguchi, 2016) and all local minima are global; this suggests a picture of gradient descent dynamics moving through neighbourhoods of saddles of ever-decreasing index until reaching a global minima. This has been termed "saddle-to-saddle" dynamics by Jacot et al. (2021). Through careful analysis of training dynamics it has been shown for DLNs that there is a general tendency of optimization trajectories towards solutions of lower loss and higher "complexity", which is generally defined in an ad-hoc way depending on the data distribution (Arora et al., 2018; Li et al., 2020; Eftekhari, 2020; Advani et al., 2020). For example, it has been shown that gradient-based optimization introduces a form of implicit regularization towards low-rank solutions in deep matrix factorization (Arora et al., 2019).

Viewing the optimization process as a search for solutions which begins at candidates of low complexity, the tendency to gradually increase complexity "only when necessary" has been put forward as a potential explanation for the generalization performance of neural networks (Gissin et al., 2019). This intuition is backed by results such as (Gidel et al., 2019; Saxe et al., 2013), which show that for DLNs the singular values of the model are learned separately at different rates, with features corresponding to larger singular values learned first.

Outside of the DLN models, saddle-to-saddle dynamics of SGD training have been studied in toy non-linear models often referred to as single-index or multi-index models. In these models, the target function for input $x \in \mathbb{R}^d$ is generated by a non-linear, low dimensional function $\varphi : \mathbb{R}^k \to \mathbb{R}$ via $f(x) = \varphi(\theta^T x)$ with $\theta \in \mathbb{R}^{d \times k}$ where $k \ll d$. Single-index refers to $k = 1$. In a very recent work, Abbe et al. (2023) showed that for a particular multi-index model with certain restrictions on the input data distribution, SGD follows a saddle-to-saddle dynamic where the learning process adaptively selects target functions of increasing complexity. Their Figure 1 tells the same story as our Figure 1: at the beginning of training, low complexity solutions are preferred, and the opposite preference develops as training progresses.

One attempt to put these intuitions in a broader context is (Zhang et al., 2018) which relates the above phenomena to entropy-energy competition in statistical physics. However this approach suffers from a lack of theoretical justification due to an incorrect application of the Laplace approximation (Wei et al., 2022b; Lau et al., 2023). The internal model selection principle (Section 4.1) in singular learning theory provides the correct form of entropy-energy competition for neural networks and potentially gives a theoretical backing for the intuitions developed in the DLN literature.

## 3 TOY MODEL OF SUPERPOSITION

### 3.1 THE TMS POTENTIAL

We recall the Toy Model of Superposition (TMS) setup from (Elhage et al., 2022) and derive a closed-form expression for the population loss in the high sparsity limit. The TMS is an autoencoder with input and output dimension $c$ and hidden dimension $r < c$:

$$f : X \times \mathcal{W} \longrightarrow \mathbb{R}^c,$$
$$f(x, w) = \text{ReLU}(W^T W x + b), \tag{1}$$

where $w = (W, b) \in \mathcal{W} \subseteq M_{r,c}(\mathbb{R}) \times \mathbb{R}^c$ and inputs are taken from $x \in X = [0, 1]^c$. We suppose that the (unknown) true generating mechanism of $x$ is given by the distribution

$$q(x) = \sum_{i=1}^{c} \frac{1}{c} \delta_{x \in C_i} \tag{2}$$

where $C_i$ denotes the $i$th coordinate axis intersected with $X$. Sampling from $q(x)$ can be described as follows: uniformly sample a coordinate $1 \leq i \leq c$ and then uniformly sample a length $0 \leq \mu \leq 1$,

return $\mu e_i$ where $e_i$ is the $i$th unit vector. This is the high sparsity limit of the TMS input distribution of Elhage et al. (2022), see also Henighan et al. (2023). We posit the probability model

$$p(x|w) \propto \exp\left(-\tfrac{1}{2}\|x - f(x,w)\|^2\right), \tag{3}$$

which leads to the expected negative log likelihood $-\int q(x) \log p(x|w)\, dx$. Dropping terms constant with respect to $w$ we arrive at the population loss function

$$L(w) = \int q(x)\|x - f(x,w)\|^2 dx\,.$$

Given $W \in M_{r,c}(\mathbb{R})$ we denote by $W_1, \ldots, W_c$ the columns of $W$. We set

1. $P_{i,j} = \{(W,b) \in M_{r,c}(\mathbb{R}) \times \mathbb{R}^c \mid W_i \cdot W_j > 0 \text{ and } -W_i \cdot W_j \leq b_i \leq 0\}$;
2. $P_i = \{(W,b) \in M_{r,c}(\mathbb{R}) \times \mathbb{R}^c \mid \|W_i\|^2 > 0 \text{ and } -\|W_i\|^2 \leq b_i \leq 0\}$;
3. $Q_{i,j} = \{(W,b) \in M_{r,c}(\mathbb{R}) \times \mathbb{R}^c \mid -W_i \cdot W_j > b_i > 0\}$

For $w = (W,b)$ we set $\delta(P_{i,j})$ to be 1 if $w \in P_{i,j}$ and 0 otherwise, similarly for $\delta(P_i), \delta(Q_{i,j})$.

**Lemma 3.1.** *For $w = (W,b) \in M_{r,c}(\mathbb{R}) \times \mathbb{R}^c$ we have $L(w) = \frac{1}{3c}H(w)$ where*

$$H(W,b) = \sum_{i=1}^{c} \delta(b_i \leq 0) H_i^-(W,b) + \delta(b_i > 0) H_i^+(W,b) \tag{4}$$

*and*

$$H_i^-(W,b) = \sum_{j \neq i} \delta(P_{i,j})\left[\frac{1}{W_i \cdot W_j}(W_i \cdot W_j + b_i)^3\right]$$

$$+ \delta(P_i)\left[\frac{b_i^3}{\|W_i\|^4} + \frac{b_i^3}{\|W_i\|^2}\right] + (1 - \delta(P_i)) + \delta(P_i)N_i$$

$$H_i^+(W,b) = \sum_{j \neq i} \delta(Q_{i,j})\left[-\frac{1}{W_i \cdot W_j}b_i^3\right]$$

$$+ \sum_{j \neq i}(1 - \delta(Q_{i,j}))\left[(W_i \cdot W_j)^2 + 3(W_i \cdot W_j)b_i + 3b_i^2\right] + N_i$$

*where $N_i = (1 - \|W_i\|^2)^2 - 3(1 - \|W_i\|^2)b_i + 3b_i^2$*

*Proof.* See Appendix G. $\qquad\square$

We refer to $H(w)$ as the **TMS potential**. While this function is analytic at many of the critical points of relevance when $r = 2$, it is not analytic at the 4-gons (see Appendix J).

### 3.2 $k$-GON CRITICAL POINTS

We prove that various $k$-gons are critical points for $H$ when $r = 2$. Recall that $w^* \in \mathcal{W}$ is a *critical point* of $H$ if $\nabla H|_{w=w^*} = 0$. The function $H$ is clearly $O(r)$-invariant: if $O$ is an orthogonal matrix then $H(OW, b) = H(W, b)$. The potential is also invariant to jointly permuting the columns and biases. Due to these *generic symmetries* we may without loss of generality assume that the columns $W_i$ of $W$ are ordered anti-clockwise in $\mathbb{R}^2$ with zero columns coming last.

For $i = 1, \ldots, c$, let $\theta_i \in [0, 2\pi)$ denote the angle between nonzero columns $W_i$ and $W_{i+1}$, where $c + 1$ is defined to be 1. Let $l_i \in \mathbb{R}_{\geq 0}$ denote $\|W_i\|$. In this parametrization $W$ has coordinate $(l_1, \ldots, l_c, \theta_1, \ldots, \theta_c, b_1, \ldots, b_c)$ with constraint $\theta_1 + \cdots + \theta_c = 2\pi$. Since $O(2)$ has dimension 1 any critical point of $H$ is automatically part of a 1-parameter family. For convenience we refer to a critical point as non-degenerate (resp. minimally singular) if it has these properties *modulo the generic symmetries*, that is, in the $\theta, l, b$ parametrization. Thus, a critical point is non-degenerate (resp. minimally singular) if in a local neighbourhood in the $\theta, l, b$ parametrization $H$ can be written

as a full sum of squares with nonzero coefficients (resp. a non-full sum of squares). For background on the minimal singularity condition see (Wei et al., 2022b, §4) and (Lau et al., 2023, Appendix A).

We call $w \in M_{2,c}(\mathbb{R}) \times \mathbb{R}^c$ a **standard $k$-gon** for $k \in \{4, 5, 6, 7, 8\}$ and $k \leq c$ if it has coordinate

$$
\begin{aligned}
l_1 = \cdots = l_k = l^*, && l_{k+1} = \cdots = l_c = 0, \\
\theta_1 = \cdots = \theta_{k-1} = \tfrac{2\pi}{k}, && \theta_k + \cdots + \theta_c = \tfrac{2\pi}{k}, \\
b_1 = \cdots = b_k = b^*, && b_{k+1} < 0, \ldots, b_c < 0
\end{aligned}
$$

where $l^* \in \mathbb{R}_{>0}, b^* \in \mathbb{R}_{\leq 0}$ are the unique joint solution to $-(l^*)^2 \cos(s\tfrac{2\pi}{k}) \leq b^*$ where $s$ is the unique integer in $[\tfrac{k}{4} - 1, \tfrac{k}{4})$ (see Theorem H.1). For values of $l^*, b^*$ see Table A.1. Any parameter of this form is proven to be a critical point of $H$ in Appendix H. For $k$ as above and $0 \leq \sigma \leq c - k$ a $k^{\sigma+}$-**gon** is a parameter with the same specification as the standard $k$-gon except that $\sigma$ of the biases $b_{k+1}, \ldots, b_c$ are equal to $1/(2c)$ and the rest have arbitrary negative values. We usually write for example $k^{++}$ when $\sigma = 2$, noting that the $k^{0+}$-gon is the standard $k$-gon. These parameters are proven to be critical points of $H$ when $k \geq 5$ in Appendix I and for $k = 4$ in Appendix J.2.

For $k = 4$ there are a number of additional "exotic" 4-gons. They are parametrized by $0 \leq \sigma \leq c - k$ and $0 \leq \phi \leq 4$. A $4^{\sigma+,\phi-}$-**gon** has the same specification as the $4^{\sigma+}$-gon, except that a subset of the biases $I \subseteq \{1, 2, 3, 4\}$ of size $|I| = \phi$ are special in the following sense: for any $i \notin I$ the bias $b_i$ has the optimal value $b_i = b^* = 0$ and the corresponding length is standard $l_i = l^* = 1$, but if $i \in I$ then $b_i < 0$ and $l_i$ is subject only to the constraint $l_i^2 < -b_i$. We write for example $4^{++-}$ for the $4^{2+,1-}$-gon. These are proven to be critical points of $H$ in Appendix J.2. In Appendix A, we provide visualizations and a quick guide for recognizing these critical points and their variants.

What we know is the following: the standard $k$-gon for $k = c$ is a non-degenerate critical point (modulo the generic symmetries) for $c \in \{5, 6, 7, 8\}$ in the sense that in local coordianates in the $l, \theta, b$-parametrization near the critical point $H$ can be written as a full sum of squares (Section H.1). For $c > 8$ and $c$ being a multiple of 4, we conjecture that the $c$-gon is a critical point (Section H.2), and we also conjecture that for $c > 8$ and $c$ not a multiple of 4 there is no $c$-gon which is a critical point. When $k \in \{5, 6, 7, 8\}$ and $k < c$ the standard $k$-gon is minimally singular (Appendix H.3).

### 3.3 LOCAL LEARNING COEFFICIENTS

The learning coefficient $\lambda$ was established by Watanabe (2009) as a central invariant of Bayesian statistics. In (Lau et al., 2023) a *local* form of the learning coefficient $\lambda(w^*)$ was proposed as a general measure of the degeneracy of a critical point $w^*$ in singular models (such as neural networks) and an estimator $\hat{\lambda}(w^*)$ was proposed based on the WBIC (Watanabe, 2013).

Table 1 summarises theoretical local learning coefficients $\lambda$ and losses $L$ for some critical points[1]. For more theoretical values see Table H.2 and Appendix H, and for empirical estimates Appendix K. In minimally singular cases (including $5, 5^+, 6$) the local learning coefficient agrees with a simple dimension count (half the number of normal directions to the level set, which is locally a manifold). This explains why the coefficient increases by $\tfrac{3}{2}$ as we move from the $k$-gon to the $(k+1)$-gon: this transition fixes one column of $W$ (2 parameters) and the corresponding entry in the bias $b$, and so reduces by 3 the number of free parameters, increasing the learning coefficient by $\tfrac{3}{2}$ (for further discussion see Appendix E).

| Critical points | Local learning coefficient $\lambda$ | Loss $L$ |
|:---:|:---:|:---:|
| 4 | $4, 4.5, 5, 5.5$ | 0.11111 |
| $4^+$ | $5, 5.5, 6, 6.5$ | 0.10417 |
| 5 | 7 | 0.06874 |
| $5^+$ | 8 | 0.06180 |
| 6 | 8.5 | 0.04819 |

Table 1: Critical points and their theoretical $\lambda$ and $L$ values for the $r = 2, c = 6$ TMS potential.

---

[1]The 4-gons are on the boundary of multiple chambers (see Appendix J). We list $4, 4.5, 5, 5.5$ as the local learning coefficient for the standard 4-gon as these are the effective dimensions when the parameter is approached from the four incident chambers.

## 4 BAYESIAN PHASE TRANSITIONS

In Bayesian statistics there is a fundamental distinction between the learning process for regular models and singular models. In regular models, as the number of samples $n$ increases, the posterior concentrates at the MAP estimator and looks increasingly Gaussian. In singular models, which include neural networks, we expect rather that the learning process is dominated by *phase transitions*, where at some critical values $n \approx n_{cr}$ the posterior "jumps" from one region of parameter space to another.[2] This is a universal phenomena in singular learning theory (Watanabe, 2009; 2020).

### 4.1 INTERNAL MODEL SELECTION

We present phase transitions of the Bayesian posterior in SLT building on (Watanabe, 2009, §7.6), (Watanabe, 2018, §9.4), Watanabe (2020). We assume $(p(x|w), q(x), \varphi(w))$ is a model-truth-prior triplet with parameter space $\mathcal{W} \subseteq \mathbb{R}^d$ satisfying the fundamental conditions of (Watanabe, 2009) and the relative finite variance condition (Watanabe, 2018). Given a dataset $\mathcal{D}_n = \{x_1, \ldots, x_n\}$, we define the empirical negative log likelihood function $L_n(w) = -\frac{1}{n} \sum_{i=1}^{n} \log p(x_i|w)$. The posterior distribution $p(w|\mathcal{D}_n)$ is, up to a normalizing constant, given by $\exp(-nL_n(w))\varphi(w)$. The marginal likelihood is the intractable normalizing constant of the posterior distribution. The free energy $F_n$ is defined to be the negative log of the marginal likelihood:

$$F_n = -\log \int_{\mathcal{W}} \exp(-nL_n(w))\varphi(w)dw. \tag{5}$$

The asymptotic expansion in $n$ is (Watanabe, 2018, §6.3) given by

$$F_n = nL_n(w_0) + \lambda \log n - (m-1) \log \log n + O_p(1) \tag{6}$$

where $w_0$ is an optimal parameter, $\lambda$ is the learning coefficient and $m$ is the multiplicity. We refer to this as the **free energy formula**.

The philosophy behind using the marginal likelihood (or equivalently, the free energy) to perform model selection is well established. Thus we could use the first two terms in (6) to choose between two competing models on the basis of their fit (as measured by $nL_n$) and their complexity (as measured by $\lambda$). We can also apply the same principle to different regions of the parameter space in the same model. Let $\{\mathcal{W}_\alpha\}_\alpha$ be a finite collection of compact semi-analytic subsets of $\mathcal{W}$ with nonempty interior, whose interiors cover $\mathcal{W}$. We assume each $\mathcal{W}_\alpha$ contains in its interior a point $w_\alpha^*$ minimising $L$ on $\mathcal{W}_\alpha$ and that the triple $(p, q, \varphi)$ restricted to $\mathcal{W}_\alpha$ in the obvious sense has relative finite variance. We refer to the $\alpha$ rather loosely as *phases*. We can choose a partition of unity $\rho_\alpha$ subordinate to a suitably chosen cover, so as to define $\varphi_\alpha(w) = \rho_\alpha(w)\varphi(w)$ with

$$F_n = -\log \int_{\mathcal{W}} e^{-nL_n(w)}\varphi(w)dw = -\log \sum_\alpha \int_{\mathcal{W}_\alpha} e^{-nL_n(w)}\varphi_\alpha(w)dw$$

$$= -\log \sum_\alpha V_\alpha \int_{\mathcal{W}_\alpha} e^{-nL_n(w)}\overline{\varphi}_\alpha(w)dw = -\log \sum_\alpha e^{-F_n(\mathcal{W}_\alpha)-v_\alpha}$$

where $\overline{\varphi}_\alpha = \frac{1}{V_\alpha}\varphi_\alpha$ for $V_\alpha = \int_{\mathcal{W}_\alpha} \varphi_\alpha dw$, $v_\alpha = -\log(V_\alpha)$ and

$$F_n(\mathcal{W}_\alpha) = -\log \int_{\mathcal{W}_\alpha} e^{-nL_n(w)}\overline{\varphi}_\alpha(w)dw \tag{7}$$

denotes the free energy of the restricted tuple $(p, q, \overline{\varphi}_\alpha, \mathcal{W}_\alpha)$. We will refer to $F_n(\mathcal{W}_\alpha)$ as the **local free energy**. Using the log-sum-exp approximation, we can write $F_n = -\log \sum_\alpha e^{-F_n(\mathcal{W}_\alpha)-v_\alpha} \approx \min_\alpha [F_n(\mathcal{W}_\alpha) + v_\alpha]$. Since (6) applies to the restricted tuple $(p, q, \overline{\varphi}_\alpha, \mathcal{W}_\alpha)$ we have

$$F_n(\mathcal{W}_\alpha) = nL_n(w_\alpha^*) + \lambda_\alpha \log n - (m_\alpha - 1) \log \log n + O_p(1) \tag{8}$$

which we refer to as the **local free energy formula**.[3]

---

[2]Another important class of phase transitions, where the posterior jumps as a hyperparameter in the prior or true distribution is varied, will not be discussed here; see (Watanabe, 2018, §9.4), (Carroll, 2021).

[3]In general deriving the free energy formula requires some sophisticated mathematics (Watanabe, 2009; 2018) but when the critical point $w_\alpha^*$ dominating the phase $\mathcal{W}_\alpha$ is minimally singular, simpler techniques similar to (Balasubramanian, 1997) suffice; see (Lau et al., 2023, Appendix A). Many, but not all, of the singularities appearing in this paper are minimally singular.

In this paper we absorb the volume constant $v_\alpha$ and terms of order $\log \log n$ or lower in (8) into a term $c_\alpha$ that we treat as effectively constant, giving

$$F_n \approx \min_\alpha \left[ nL_n(w_\alpha^*) + \lambda_\alpha \log n + c_\alpha \right]. \qquad (9)$$

A principle of *internal model selection* is suggested by (9) whereby the Bayesian posterior "selects" a phase $\alpha$ based on the local free energy of the phase, in the sense that this phase contains most of the probability mass of the posterior for this value of $n$ (Watanabe, 2009, §7.6).[4] At a given value of $n$ we can order the phases by their posterior concentration, or what is the same, their free energies $F_n(\mathcal{W}_\alpha)$. We say there is a *local phase transition* between phases $\alpha, \beta$ at *critical sample size* $n_{cr}$, written $\alpha \to \beta$, if the position of $\alpha, \beta$ in this ordered list of phases swaps. That is, for $n \approx n_{cr}$ and $n < n_{cr}$ the Bayesian posterior prefers $\alpha$ to $\beta$, and the reverse is true for $n > n_{cr}$. We say that a phase $\alpha$ *dominates the posterior* at $n$ if it has the highest posterior mass, that is, $F_n(\mathcal{W}_\alpha) < F_n(\mathcal{W}_\beta)$ for all $\beta \neq \alpha$. A *global* phase transition is a local phase transition where $\alpha$ dominates the posterior for $n < n_{cr}$ and $\beta$ dominates for $n > n_{cr}$ with $n$ near $n_{cr}$. Generally when we speak of a phase transition in this paper we mean a *local* transition.

Generically, phase transitions occur when, as $n$ increases, phases with lower loss and higher complexity are preferred; this expectation is verified in TMS in the next section. For more on the theory of Bayesian phase transitions see Appendix C.

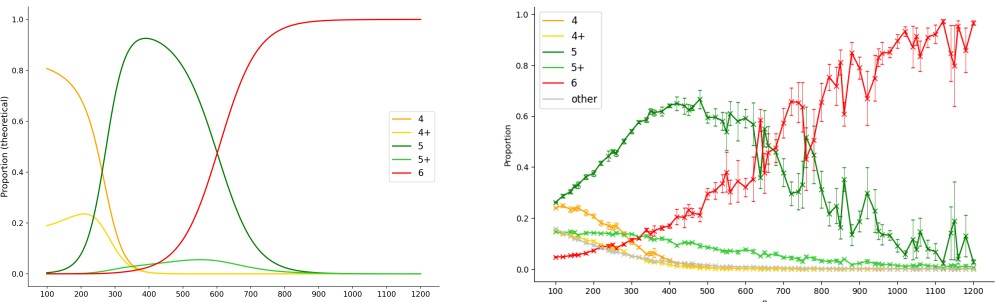

Figure 2: Proportion of Bayesian posterior concentrated in regions $\mathcal{W}_{k,\sigma}$ for $r = 2, c = 6$ according to the free energy formula (theory, left) and MCMC sampling of the posterior (experimental, right). Theory predicts, and experiments show, a phase transition $5 \to 6$ in the range $600 \leq n \leq 700$.

## 4.2 EXPERIMENTS

There is a tension in the internal model selection story: the free energy formula is asymptotic in $n$, but the discussion of phase transitions involves $F_n(\mathcal{W}_\alpha)$ at finite $n$. Whether this is valid, in a given range of $n$ and for a given system, is a question that may be difficult to resolve purely theoretically. We show experimentally for $r = 2, c = 6$ that a Bayesian phase transition actually takes place between the 5-gon and the 6-gon, within a range of $n$ values consistent with the free energy formula.

In this section we focus on the case $r = 2, c = 6$. For $c \in \{4, 5\}$ see Appendix F.4. We first define regions of parameter space $\{\mathcal{W}_\alpha\}_\alpha$. Given a matrix $W$ we write $\text{ConvHull}(W)$ for the number of points in the convex hull of the set of columns. For $3 \leq k \leq c, 0 \leq \sigma \leq c - k$ we define

$$\mathcal{W}_{k,\sigma} = \left\{ w = (W, b) \in \mathcal{W} \,|\, \text{ConvHull}(W) = k \text{ and } b \text{ has } \sigma \text{ positive entries} \right\}.$$

The set $\mathcal{W}_{k,\sigma} \subseteq \mathbb{R}^d$ is semi-analytic and contains the $k^{\sigma+}$-gon in its interior. For $\alpha = (k, \sigma)$ we let $w_\alpha^*$ denote the parameter of the $k^{\sigma+}$-gon. We verify experimentally the hypothesis that this parameter dominates the Bayesian posterior of $\mathcal{W}_\alpha$ (see Appendix F) by which we mean that most samples from the posterior for a relevant range of $n$ values are "close" to $w_\alpha^*$.[5] In this sense the choice of phases $\mathcal{W}_\alpha$ is appropriate for the range of sample sizes we consider.

---

[4]We often replace $L_n(w_\alpha^*)$ by $L(w_\alpha^*)$ in comparing phases; see (Watanabe, 2018, §9.4).

[5]The $k^{\sigma+,\phi-}$-gons for $\phi > 0$ have high loss but may nonetheless dominate the posterior for very low $n$, however this is outside the scope of our experiments, which ultimately dictates the choice of the set $\mathcal{W}_{k,\sigma}$.

We draw posterior samples using MCMC-NUTS (Homan & Gelman, 2014) with prior distribution $N(0, 1)$ and sample sizes $n$. Each posterior sample is then classified into some $\mathcal{W}_{k,\sigma}$ (our classification algorithm for the deciding the appropriate value of $k$ is not error-free.) For each $n$, 10 datasets are generated and the average proportion of the $k$-gons, and standard error, is reported in Figure 2. Details of the theoretical proportion plot are given in Appendix F.1.

Let $w_\alpha^*, w_\beta^*$ be $k$-gons dominating phases $\mathcal{W}_\alpha, \mathcal{W}_\beta$. A *Bayesian phase transition* $\alpha \to \beta$ occurs when the difference between the free energies $F_n(\mathcal{W}_\beta) - F_n(\mathcal{W}_\alpha)$ changes from positive to negative.

The most distinctive feature in the experimental plot is the $5 \to 6$ transition in the range $600 \le n \le 700$. The free energy formula predicts this transition at $n_{cr} \approx 600$ (Appendix C.2). An alternative visualization of the $5 \to 6$ transition using t-SNE is given in Appendix F.2. As $n$ decreases past 400 the MCMC classification becomes increasingly uncertain, and it is less clear that we should expect the free energy formula to be a good model of the Bayesian posterior, so we should not read too much into any correspondence between the plots for $n \le 400$ (see Appendix F.2).

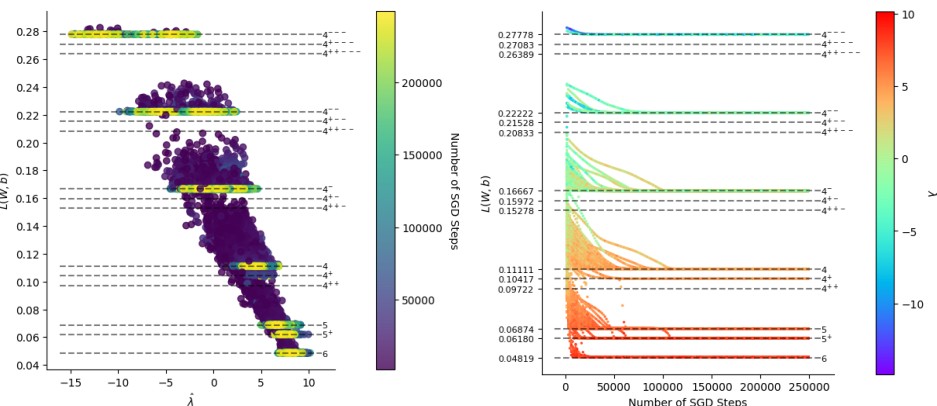

Figure 3: Visualization of 400 SGD trajectories initialized at MCMC samples from the Bayesian posterior for $r = 2, c = 6$ at sample size $n = 100$. We see that SGD trajectories are dominated by plateaus at loss values corresponding to our classification of critical points (Appendix A) and that lower loss critical points have higher estimated local learning coefficients. Note that for highly singular critical points we see that $\hat\lambda$ is unable to provide non-negative values without additional hyperparameter tuning, but the ordinality (more positive is less degenerate) is nonetheless correct. See Appendix K for details and caveats for the $\hat\lambda$ estimation.

## 5 DYNAMICAL PHASE TRANSITIONS

A *dynamical* transition $\alpha \to \beta$ occurs in a trajectory if it is near a critical point $w_\alpha^*$ of the loss at some time $\tau_1$ (e.g. there is a visible plateau in the loss) and at some later time $\tau_2 > \tau_1$ it is near $w_\beta^*$ without encountering an intermediate critical point. We conduct an empirical investigation into whether the $k$-gon critical points of the TMS potential dominate the behaviour of SGD trajectories for $r = 2, c = 6$, and the existence of dynamical transitions.

There are two sets of experiments. In the first we draw a training dataset $\mathcal{D}_n = \{x_1, \ldots, x_n\}$ where $n = 1000$ from the true distribution $q(x)$. We also draw a test set of size 5000. We use minibatch-SGD initialized at a 4-gon plus Gaussian noise of standard deviation 0.01, and run for 4500 epochs with batch size 20 and learning rate 0.005. This initialisation is chosen to encourage the SGD trajectory to pass through critical points with high loss after a small number of SGD steps, allowing us to observe phase transitions more easily. Along the trajectory, we keep track of each iterate's training loss, test set loss and theoretical test loss. Figure 1 is a typical example, additional plots are collected in Figures B.1-B.7. In the second set of experiments, summarized in Figure 3, we take the same size training dataset but initialize SGD trajectories differently, at random MCMC samples from the Bayesian posterior at $n = 100$ (a small value of $n$). The number of epochs is 5000.

In both cases we estimate the local learning coefficient of the training iterates $w_t$. This is a newly-developed estimator Lau et al. (2023) that uses SGLD (Welling & Teh, 2011) to estimate a version of the WBIC (Watanabe, 2013) localized to $w_t$ and then forms an estimate of the local learning coefficient $\hat{\lambda}(w_t)$ based on the approximation that $\mathrm{WBIC}(w_t) \approx nL_n(w_t) + \lambda(w_t)\log n$ where $\lambda(w_t)$ is the local RLCT. In the language of Lau et al. (2023), we use a full-batch version of SGLD with hyperparameters $\epsilon = 0.001, \gamma = 1$ and 500 steps to estimate the local WBIC.

These experiments support the following description of SGD training for the TMS potential when $r = 2, c = 6$: trajectories are characterised by plateaus associated to the critical points described in Section 3.2 and further discussed in Appendix A. The dynamical transitions encountered are

$$4^{++---} \longrightarrow 4^{+--} \longrightarrow 4^-\,, \qquad 4^{+---} \longrightarrow 4^{--}\,,$$
$$4^{++--} \longrightarrow 4^{+-} \longrightarrow 4\,, \qquad 4^{++-} \longrightarrow 4^+ \longrightarrow 5 \longrightarrow 5^+\,. \tag{10}$$

The dominance of the classified critical points, and the general relationship of decreasing loss and increasing complexity, can be seen in Figure 3.

## 5.1 Relation Between Bayesian and Dynamical Transitions

Phases of the Bayesian posterior for TMS with $r = 2, c = 6$ are dominated by $k$-gons which are critical points of the TMS potential (Section 4). The same critical points explain plateaus of the SGD training curves (Section 5). This is not a coincidence: on the one hand SLT predicts that phases of the Bayesian posterior will be associated to singularities of the KL divergence, and on the other hand it is a general principle of nonlinear dynamics that singularities of a potential dictate the global behaviour of solution trajectories (Strogatz, 2018; Gilmore, 1981).

However, the relation between *transitions* of the Bayesian posterior and *transitions* over SGD training is more subtle. There is no *necessary* relation between these two kinds of transitions. A Bayesian transition $\alpha \to \beta$ might not have an associated dynamical transition if, for example, the regions $\mathcal{W}_\alpha, \mathcal{W}_\beta$ are distant or separated by high energy barriers. For example, the Bayesian phase transition $5 \to 6$ has not been observed as a dynamical transition (it may occur, just with low probability per SGD step). However, it seems reasonable to expect that for many dynamical transitions there exists a Bayesian transition between the same phases. We call this the *Bayesian antecedent* of the dynamical transition if it exists. This leads us to:

**Bayesian Antecedent Hypothesis (BAH).** The dynamical transitions $\alpha \to \beta$ encountered in neural network training have Bayesian antecedents.

Since a dynamical transition decreases the loss, the main obstruction to having a Bayesian antecedent is that in a Bayesian phase transition $\alpha \to \beta$ the local learning coefficient should increase (Appendix D). Thus the BAH is in a similar conceptual vein to the expectation, discussed in Section 2, that SGD prefers higher complexity critical points as training progresses. While the dynamical transitions in (10) are all associated with increases in our estimate of the local learning coefficient, we also know that at low $n$, the constant terms can play a nontrivial role in the free energy formula. Our analysis (Appendix D.1) suggests that all dynamical transitions in (10) have Bayesian antecedents, with the possible exception of $4^{++---} \to 4^{+--}$ and $4^{+---} \to 4^{--}$ where the analysis is inconclusive.

## 6 Conclusion

Phase transitions and emergent structure are among the most interesting phenomena in modern deep learning (Wei et al., 2022a; Barak et al., 2022; Liu et al., 2022) and provide an interesting avenue for fundamental progress in neural network interpretability (Olsson et al., 2022; Nanda et al., 2023) and AI safety (Hoogland et al., 2023). Building on Elhage et al. (2022) we have shown that the Toy Model of Superposition with two hidden dimensions has, in the high sparsity limit, phase transitions in both stochastic gradient-based and Bayesian learning. We have shown that phases are in both cases dominated by $k$-gon critical points which we have classified, and we have proposed with the BAH a relation between transitions in SGD training and phase transitions in the Bayesian posterior.

Our analysis of TMS also demonstrates the practical utility of the local complexity measure $\hat{\lambda}$ introduced in (Lau et al., 2023), which is an all-purpose tool for measuring model complexity.

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

# A   FANTASTIC CRITICAL POINTS AND WHERE TO FIND THEM

In this section we provide a brief guide, using loose intuitive language, to recognise known critical parameters and their variations. Rigorous derivation and details are given in Appendix H, I, and J.

Broadly speaking, known critical parameters, $(W^*, b^*)$, are classified by three discrete numbers:

- $k$: the number of vertices in the *regular* polygon formed by the convex hull of the columns of the $W^*$ matrix, interpreted as a vector in $\mathbb{R}^2$. The length of these vectors (for any $k \leq c$) have to be at the optimal values derived in Appendix H and listed in Table A.1.

- $\sigma$: the number of positive values in the bias vector. These positive biases are required to take on the optimal value at $b^* = 1/(2c)$ and have to occur at indices that *do not* correspond to the $k$-gon vertices.

- $\phi$: the number of large negative values in the bias vectors. So far, we've only observed $\phi > 0$ when $k = 4$, i.e. this discrete subcategory only applies to 4-gons. These biases have to occur at indices that *do* correspond to the 4-gon vertices.

For $r = 2, c = 6$, the above description and constraints result in the 18 families of critical points whose representative members are shown in Figure A.1 and their loss or potential energy levels are shown in Figure A.2.

Next, we discuss possible variations within these families of critical points. Aside from the ever-present rotational and permutation symmetries discussed elsewhere, there are variations of these standard descriptions that allow the parameter to stay on the same critical submanifold. Figure A.3 shows some examples of irregular versions of known critical points. One can cross-check that their potential values $L$ are the same as their regular counterpart. Most of these variation is the result of having negative values in the bias vectors allowing for extra variability without changing the loss value. To explain the examples in Figure A.3,

- 5-gon (top left). In the standard 5-gon family, the vestigial bias $b'$ can have arbitrary negative value and corresponding weight column can be any vector so long as its length $l'$ is smaller than $\sqrt{\min\{|b'|, |b^*|\}}$ where $b^*$ is the optimal negative bias for the main columns (see Table A.1).

- 4-gon (top right). The two vestigial biases can take arbitrary negative values.

- $4^{+--}$-gon (bottom left). Having two negative biases with large magnitude afford a few other degrees of freedom. The weight columns $W_3, W_4$ with those large negative biases can be any vector as long as (1) they lengths is smaller than $\sqrt{|b_i|}$ for their respective biases and (2) they form obtuse angle relative other columns $W_1$ and $W_2$, i.e the other two vertices of the 4-gon.

- $4^{--}$-gon (bottom right). Other than the variation in the main columns $W_3, W_6$ with large negative biases, the two vestigial columns can also be any vectors as long as they stay within the sector between $W_1$ and $W_2$ and their lengths are bounded by $\min\left\{\sqrt{|b_i|} \mid i = 3, 4, 5, 6\right\}$.

| Critical point | $l^*$ | $b^*$ |
|---|---|---|
| 4-gon | 1 | 0 |
| 5-gon | 1.17046 | $-0.28230$ |
| 6-gon | 1.32053 | $-0.61814$ |

Table A.1: Parameters of certain $k$-gons.

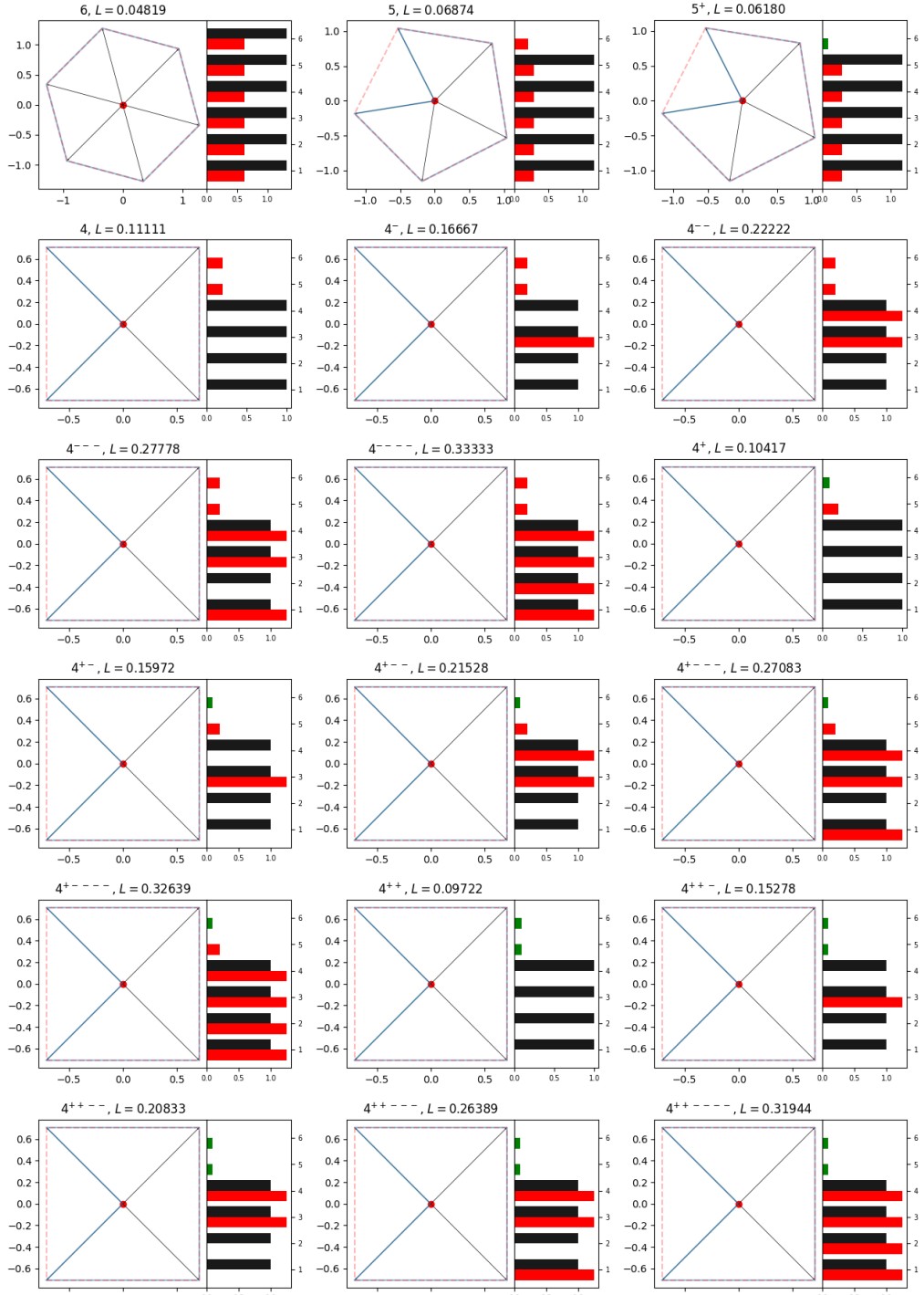

Figure A.1: Representative of each known class of critical parameters in $r = 2, c = 6$.

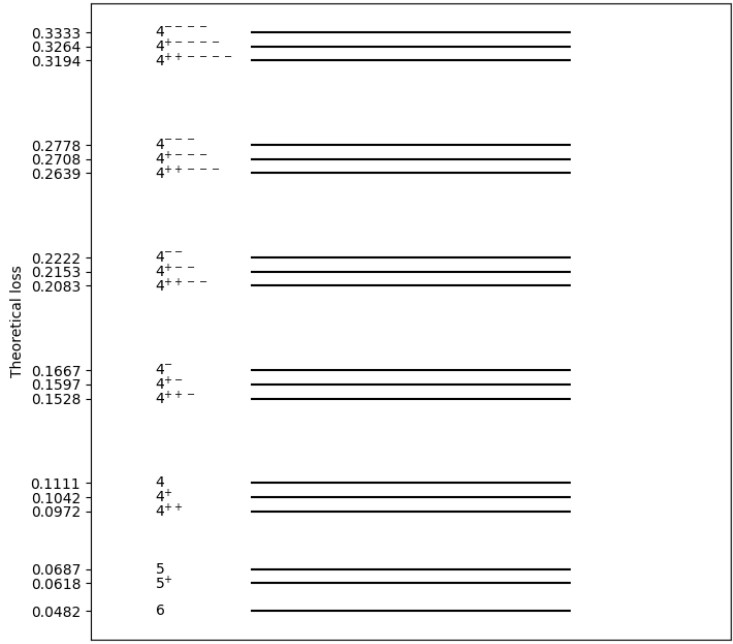

Figure A.2: Potential energy levels $L$ for known critical points in $r = 2, c = 6$.

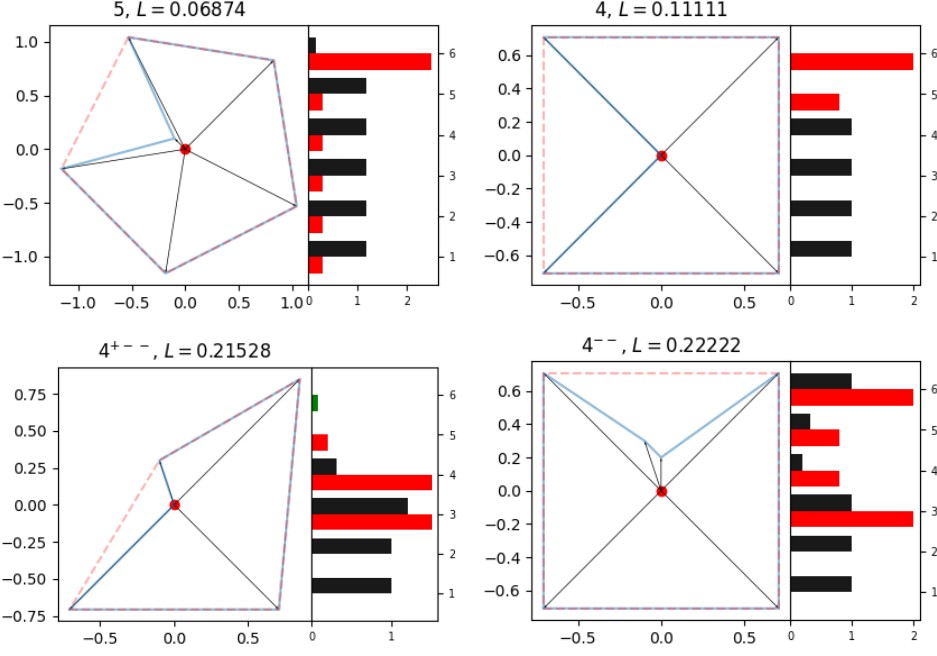

Figure A.3: Irregular versions of known critical points in $r = 2, c = 6$.

# B    ADDITIONAL EXAMPLES OF SGD TRAJECTORIES FOR THE $r = 2, c = 6$ TMS POTENTIAL

In this Appendix we collect additional individual SGD trajectories for the $r = 2, c = 6$ TMS potential, with the same hyperparameters as discussed in Section 5. These are all of the runs from 30 random seeds that had a dynamical transition. We note that each of the critical points encountered in a plateau fall into the classification discussed in Appendix A and the estimator $\hat{\lambda}$ for the local learning coefficient jumps in each transition. Note that the transitions in Figure 1 are $4^{++-} \to 4^{+} \to 5$.

The instructions for reading the figures in this appendix, for example Figure B.1, are the same as for Figure 1 in the main text.

We note that in some runs containing particularly degenerate $k$-gons, such as the $4^{++---}$ in Figure B.1, the estimator $\hat{\lambda}$ produces negative values for the standard hyperparameter $\epsilon = 0.001$. By adapting this hyperparameter to the level of degeneracy we can correct for this and avoid invalid estimates (see Appendix K). But since we cannot predict the trajectory of SGD iterates, we choose to use a fixed hyperparameter $\gamma = 1.0$, $\epsilon = 0.001$, number of SGLD steps $= 500$ in all Figures of this form.

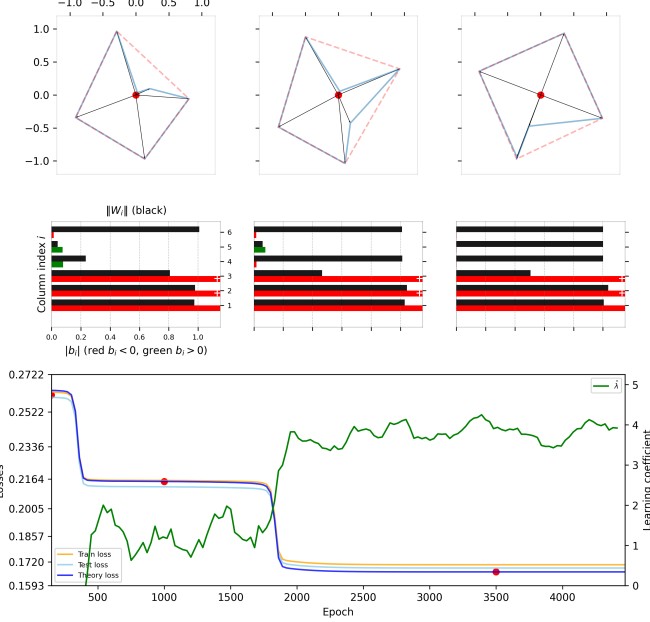

Figure B.1: Trajectory with dynamical transitions $4^{++---} \to 4^{+--} \to 4^{-}$.

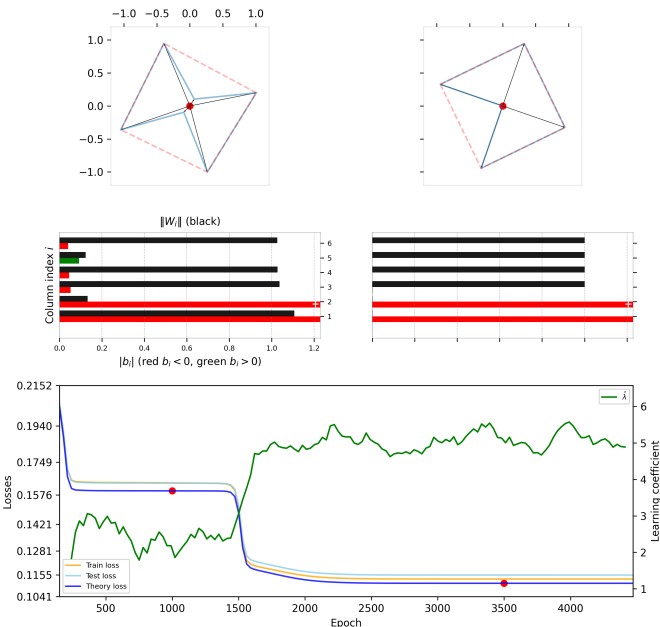

Figure B.2: Trajectory with dynamical transition $4^{+-} \to 4$.

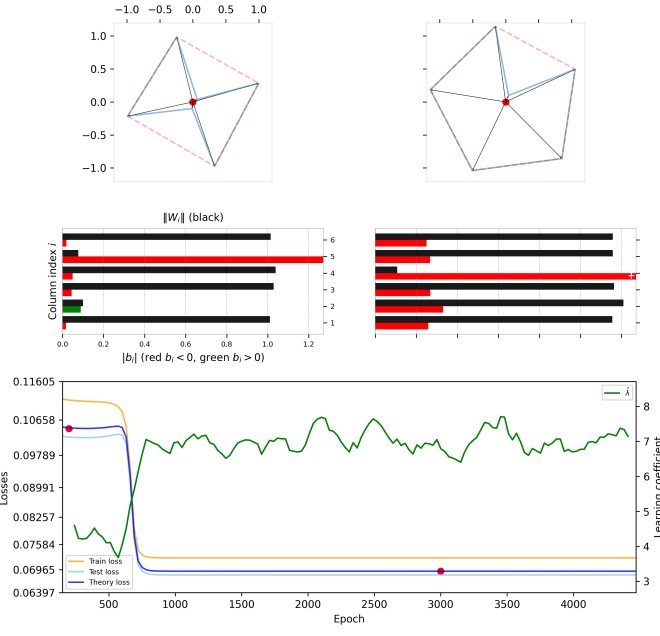

Figure B.3: Trajectory with dynamical transition $4^+ \to 5$.

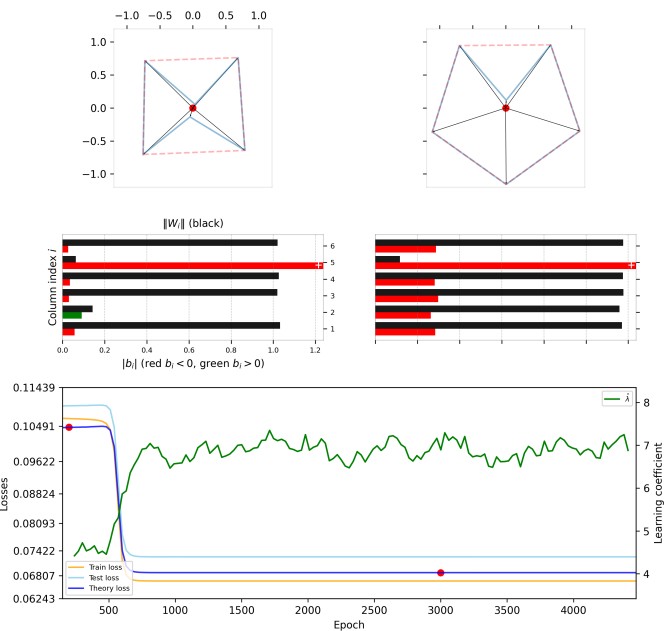

Figure B.4: Trajectory with dynamical transition $4^+ \to 5$.

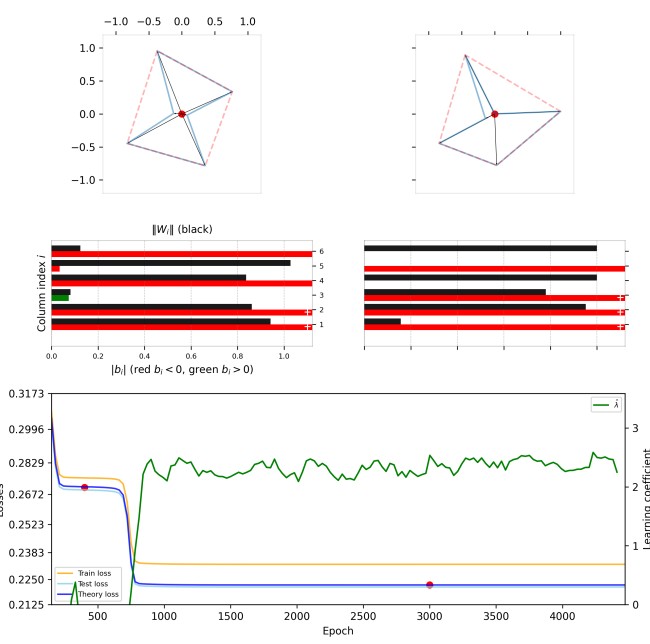

Figure B.5: Trajectory with dynamical transition $4^{+---} \to 4^{--}$.

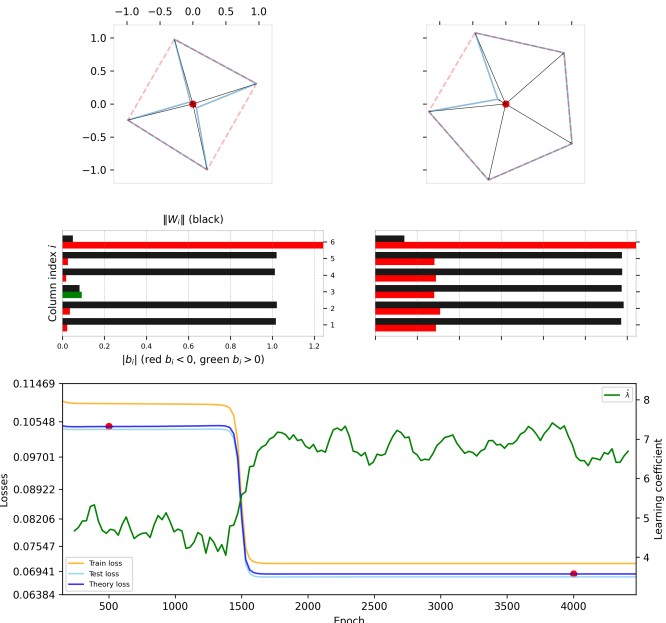

Figure B.6: Trajectory with dynamical transition $4^+ \rightarrow 5$.

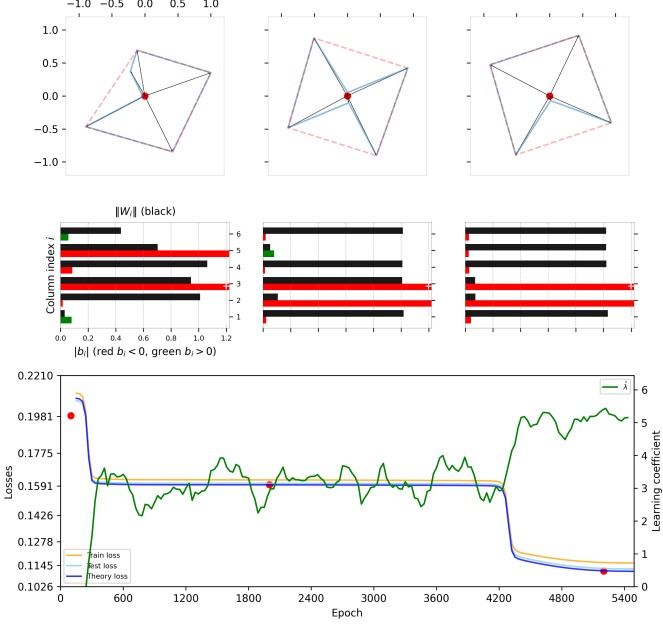

Figure B.7: Trajectory with dynamical transition $4^{++--} \rightarrow 4^{+-} \rightarrow 4$.

## C    USING THE FREE ENERGY FORMULA

In Section 4.1 we defined a (local) phase transition $\alpha \to \beta$ at a critical sample size $n_{cr}$ to take place when the local free energies swap order, as in the following table, with $n$ taken close to $n_{cr}$:

| $n < n_{cr}$ | $n = n_{cr}$ | $n > n_{cr}$ |
|---|---|---|
| $F_n(\mathcal{W}_\alpha) < F_n(\mathcal{W}_\beta)$ | $F_n(\mathcal{W}_\alpha) \approx F_n(\mathcal{W}_\beta)$ | $F_n(\mathcal{W}_\alpha) > F_n(\mathcal{W}_\beta)$ |

We assume $n$ is in a range where the local free energies for $\gamma \in \{\alpha, \beta\}$ are well-approximated by the right hand-side of the following

$$F_n(\mathcal{W}_\gamma) \approx nL(w_\gamma^*) + \lambda_\gamma \log n + c_\gamma \tag{11}$$

for some constant $c_\gamma$. While $n$ is of course an integer, to find where the free energy curves cross we may treat $n$ as a real variable. To an ordered pair $\alpha, \beta$ we may associate

$$\Delta L = L(w_\beta^*) - L(w_\alpha^*)$$
$$\Delta \lambda = \lambda_\beta - \lambda_\alpha$$
$$\Delta c = c_\beta - c_\alpha$$

Then to solve $F_n(W_\alpha) = F_n(W_\beta)$ for $n$ we may instead solve

$$n\Delta L + \Delta \lambda \log n + \Delta c = 0. \tag{12}$$

Theoretically, a phase transition $\alpha \to \beta$ exists if and only if this equation has a positive solution. However, in practice the free energy formula on which this equation is based will only well describe the Bayesian posterior for sufficiently large $n$, and it is an empirical question what this $n$ may be. In the following when we say that a phase transition is predicted to exist (or not), the reader should keep this caveat in mind.

When we refer to theoretically derived values for phase transitions, we mean that we solve (12) with the given values of $\Delta L, \Delta \lambda, \Delta c$. Note that if the phase $\beta$ has lower loss, learning coefficient and constant term (so that $\Delta L$, $\Delta \lambda$ and $\Delta c$ are all negative) then there can be no phase transition $\alpha \to \beta$ as $F_n(\mathcal{W}_\alpha)$ is never lower than $F_n(\mathcal{W}_\beta)$.

Although the constant (and lower order) terms in the free energy expansion are not well-understood, in this paper we proceed assuming that the leading contribution comes from the prior in the manner described in Section C.1 below.

### C.1    CONSTANT TERMS IN THE FREE ENERGY FORMULA

Recall from Section 4.1 that given a collection of phases $\{\mathcal{W}_\alpha\}$ the free energy is

$$F_n = -\log \sum_\alpha V_\alpha \int_{\mathcal{W}_\alpha} e^{-nL_n(w)} \overline{\varphi}_\alpha(w) dw$$

where $\overline{\varphi}_\alpha = \frac{1}{V_\alpha} \varphi_\alpha$ for $V_\alpha = \int_{\mathcal{W}_\alpha} \varphi_\alpha dw$. Suppose that the phase $\mathcal{W}_\alpha$ is dominated by a critical point $w_\alpha^*$ and that the partition of unity is chosen so that $\varphi_\alpha(w_\alpha^*) \approx \varphi(w_\alpha)$ (this is reasonable since the critical point is in the interior). We explore the following approximation to the contribution of $\alpha$ to the above integral

$$\int_{\mathcal{W}_\alpha} e^{-nL_n(w)} \overline{\varphi}_\alpha(w) dw \approx \overline{\varphi}(w_\alpha^*) \int_{\mathcal{W}_\alpha} e^{-nL_n(w)} dw.$$

This means that the prior contributes to $c_\alpha$ of (8) through $-\log(V_\alpha \overline{\varphi}(w_\alpha^*))$ as well as through the $O_P(1)$ term of the asymptotic expansion. With a normal prior $\varphi = \frac{1}{\sigma\sqrt{(2\pi)^d}} \exp(-\frac{1}{2\sigma^2} \|w\|^2)$

$$V_\alpha \overline{\varphi}(w_\alpha^*) = \varphi_\alpha(w_\alpha^*) \approx \frac{1}{\sigma\sqrt{(2\pi)^d}} \exp\left(-\frac{1}{2\sigma^2} \|w_\alpha^*\|^2\right).$$

Hence $-\log(V_\alpha \overline{\varphi}(w_\alpha^*))$ depends on $\sigma$ through the sum $\log \sigma + \frac{1}{2\sigma^2} \|w_\alpha\|^2$. Here if $w_\alpha^* = (W_\alpha^*, b_\alpha^*)$ we have $\|w_\alpha^*\|^2 = \|W_\alpha^*\|^2 + \|b_\alpha^*\|^2$. In Table C.1, Table C.2 we show the value of this contribution when $\sigma = 1$ for $c \in \{5, 6\}$. Note that for some $k$-gons there are negative biases that can take arbitrarily large values, so the shown values are lower bounds.

| Critical point | $\frac{1}{2}\|w_\alpha^*\|^2$ |
|---|---|
| 4 | $\geq 2$ |
| $4^+$ | $\geq 2.05$ |
| 5 | $\geq 3.62417$ |

Table C.1: Prior factors for $k$-gons when $c = 5$.

| Critical point | $\frac{1}{2}\|w_\alpha^*\|^2$ |
|---|---|
| 6 | 6.37767 |
| 5 | > 3.62417 |
| $5^+$ | 3.62764 |
| 4 | 2 |
| $4^-$ | > 1.5 |
| $4^{--}$ | > 1 |
| $4^{---}$ | > 0.5 |
| $4^{----}$ | > 0 |
| $4^+$ | 2.00347 |
| $4^{+-}$ | > 1.50347 |
| $4^{+--}$ | > 1.00347 |
| $4^{+---}$ | > 0.50347 |
| $4^{+----}$ | > 0.00347 |
| $4^{++}$ | 2.00694 |
| $4^{++-}$ | > 1.50694 |
| $4^{++--}$ | > 1.00694 |
| $4^{++---}$ | > 0.50694 |
| $4^{++----}$ | > 0.00694 |

Table C.2: Prior factors for $k$-gons when $c = 6$.

## C.2 THEORETICAL PREDICTIONS FOR THE 5-GON TO 6-GON TRANSITION FOR $r = 2, c = 6$

With $\alpha = 5$ and $\beta = 6$ we have from Table 1 and Table C.2 that

$$\Delta L = 0.04819 - 0.06874 = -0.02055$$
$$\Delta\lambda = 8.5 - 7 = 1.5$$
$$\Delta c = 6.37767 - 3.62417 = 2.7535$$

Solving (12) numerically gives $n_{cr} = 601$ as the closest integer.

## C.3 INFLUENCE OF CONSTANT TERMS

Dividing (12) through by $\log n$ we have

$$\frac{n}{\log n} = -\frac{1}{\log n}\frac{\Delta c}{\Delta L} - \frac{\Delta\lambda}{\Delta L} = -\frac{1}{\Delta L}\left[\frac{\Delta c}{\log n} + \Delta\lambda\right]. \tag{13}$$

In the phase transitions we analyse in this paper $\Delta L$ is on the order of 0.01, $\Delta\lambda$ is on the order of 1, and $\Delta c$ is on the order of 1, so $\Delta\lambda/\Delta L, \Delta c/\Delta L$ are on the order of 10. In Figure 2 we care about roughly $200 \leq n \leq 1000$ so $5 \leq \log n \leq 7$. Hence in practice the first term in (13) is roughly one order of magnitude lower than the second; the upshot being that the primary determinant of $n_{cr}$ is $|\Delta\lambda/\Delta L|$ but the influence of the constant terms can be significant.

In the second transition of Figure 1 from $4^+ \to 5$ we have $\Delta\lambda = 2$, $\Delta L = 0.06874 - 0.10417 = -0.03543$ (based on Table 1) and $\Delta c = 3.62417 - 2.00347 = 1.6207$ (based on Table C.2) so $-\Delta c/\Delta L \approx 45$ and $-\Delta\lambda/\Delta L \approx 56$. Solving (12) numerically yields $n_{cr} \approx 380$. Solving the equation with $\Delta c = 0$ gives $n_{cr} \approx 327$, so as suggested above including the constant term shifts the critical sample size by a lower order term.

## C.4 DOUBLE TRANSITIONS

Assume that there are transitions $\alpha \to \beta$ at critical sample size $n_1$ and $\beta \to \gamma$ at critical sample size $n_2$, both involving no change in constant terms so that (16) applies. Since $n/\log n$ is increasing, if $n_1 < n_2$ we deduce

$$-\frac{\Delta\lambda_1}{\Delta L_1} < -\frac{\Delta\lambda_2}{\Delta L_2} \tag{14}$$

where

$$\begin{aligned}
\Delta\lambda_1 &= \lambda_\beta - \lambda_\alpha\,, \\
\Delta\lambda_2 &= \lambda_\gamma - \lambda_\beta\,, \\
\Delta L_1 &= L(w_\beta^*) - L(w_\alpha^*)\,, \\
\Delta L_2 &= L(w_\gamma^*) - L(w_\beta^*)\,.
\end{aligned}$$

From (14) we obtain the inequality

$$\Delta L_2 \Delta\lambda_1 > \Delta\lambda_2 \Delta L_1 \implies \frac{\Delta L_2}{\Delta\lambda_2} > \frac{\Delta L_1}{\Delta\lambda_1} \tag{15}$$

which says that: *along any curve in the $(\lambda, L)$ plane following a sequence of Bayesian phase transitions, the slope must increase.* For example, we observe in Figure D.1 that the negative slopes become successively less negative as we move along a sequence of transitions. This is the least obvious for the pair of transitions $4^{+--} \to 4^+$ and $4^+ \to 5$ which corresponds to the fact that the gap $n_2 - n_1$ is small in Figure D.5.

## D BAYESIAN ANTECEDENTS

In this section we review whether the phase transitions we find empirically have Bayesian antecedents. To begin we consider the case where $\Delta c = 0$. Then from (13) we deduce

$$\frac{n}{\log n} = -\frac{\Delta\lambda}{\Delta L}\,. \tag{16}$$

For $n > 3$, $n/\log n$ is positive and an increasing function of $n$, and we denote the inverse function by $\mathcal{N}$. Since the critical sample size for a transition $\alpha \to \beta$ must be positive, if $\Delta L < 0$ (the loss decreases) then (16) has a (unique) solution if and only if $\Delta\lambda > 0$ (the complexity increases). The unique solution is the critical sample size

$$n_{cr} = \mathcal{N}\Big(-\frac{\Delta\lambda}{\Delta L}\Big)\,.$$

If $\Delta L < 0$ and $\Delta c \neq 0$ we simply plot the free energy curves and see if they intersect. Given the orders of magnitude discussed in Section C.3 we expect if $\Delta c > 0$, $\Delta\lambda > 0$ then there is likely to be a solution, whereas if $\Delta c < 0$, $\Delta\lambda < 0$ then the right hand side of (13) is negative and no transition can exist. The mixed cases are harder to argue about in general terms.

### D.1 THE BAH FOR $r = 2, c = 6$

We examine the evidence for the existence of Bayesian antecedents of the dynamical transitions in TMS for $r = 2, c = 6$ exhibited in Section 5. The known dynamical transitions are summarised in Figure D.1. The slope of the lines is, in the notation of (16), equal to $\frac{\Delta L}{\Delta\lambda}$ and so the fact that all observed phase transitions go down and to the right would indicate, if the constant terms were ignored, that the critical sample size is positive and a Bayesian phase transition exists. Here the $L$ values are from Section A and the $\hat{\lambda}$ values from Table K.1 (note the caveats there) for those $k$-gons where we do not have theoretically derived values (for $\alpha \in \{5, 5^+, 6\}$ see Table 1).

To perform a more refined analysis which includes the constant terms we use Table C.2 and compare plots of free energy curves. In the cases where we use an empirical estimate of the learning coefficient, we display the curve as part of a shaded region made up of curves with coefficients of $\log n$ within one standard deviation of the estimate. The results are shown in Figures D.2-D.5.

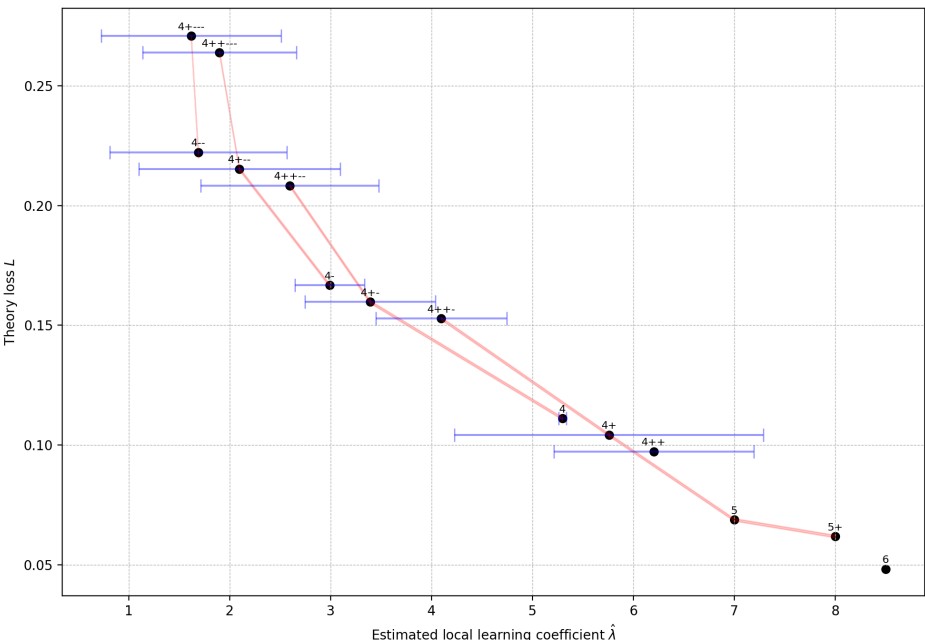

Figure D.1: Summary of known dynamical transitions and the phases involved. Each scatter point $(\hat{\lambda}_\alpha, L_\alpha)$ corresponds to one of our classified critical points $w_\alpha^*$ and a red line is drawn between phases with dynamical transitions connecting them (in the direction that goes right and down) as listed in (10). These "curves" are necessarily concave up if the time order of dynamical transitions matches the sample size order of Bayesian transitions, see C.4.

For phase transitions occurring at large values of $n$, the existence of a transition is relatively insensitive to small changes in the learning coefficient or constant terms, and we can also be more confident that the predicted transition translates (via the correspondence between the free energy formula and the posterior, which is only valid for sufficiently large $n$) to an actual phase transition in the posterior. For transitions occurring at low $n$, such as those in Figure D.2 and Figure D.3, the analysis is strongly affected by small changes in learning coefficient or constant terms, and so we cannot be sure that a Bayesian transition exists.

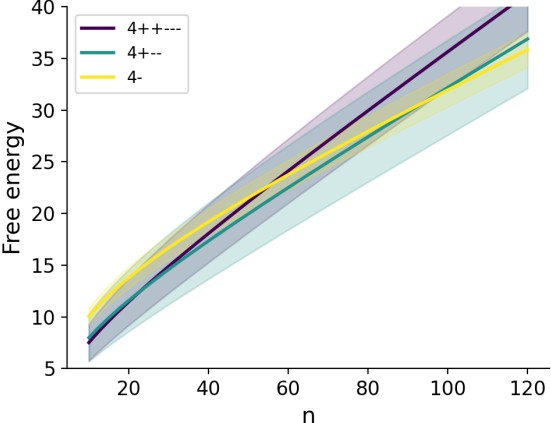

Figure D.2: Free energy plot providing evidence of Bayesian transitions $4^{++---} \to 4^{+--}$ and $4^{+--} \to 4^-$. In the former case the plot is merely suggestive, since the transition takes place at low $n$ and is very sensitive to the learning coefficients and constant terms.

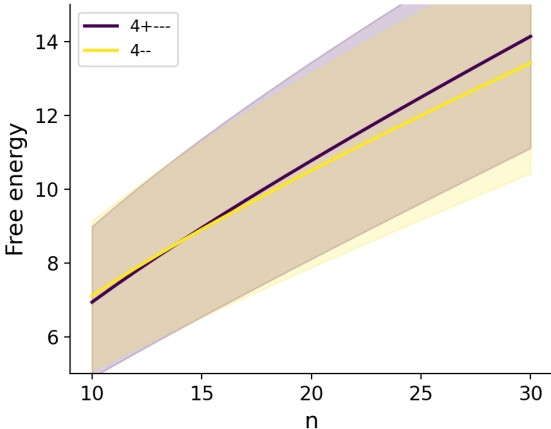

Figure D.3: Free energy plot providing weak evidence of a Bayesian transition $4^{+---} \to 4^{--}$. The transition takes place at low $n$ and is very sensitive to the learning coefficients and constant terms.

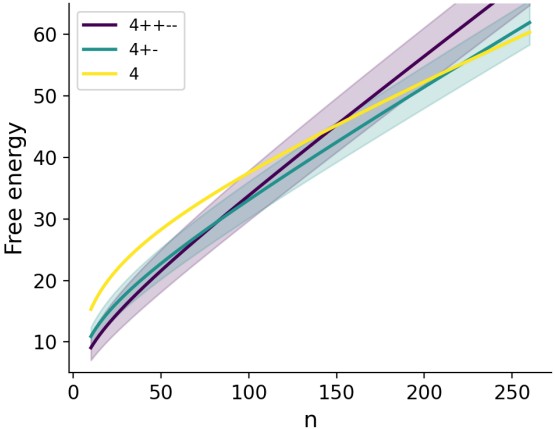

Figure D.4: Free energy plots suggesting Bayesian transitions $4^{++--} \to 4^{+-}$ and $4^{+-} \to 4$.

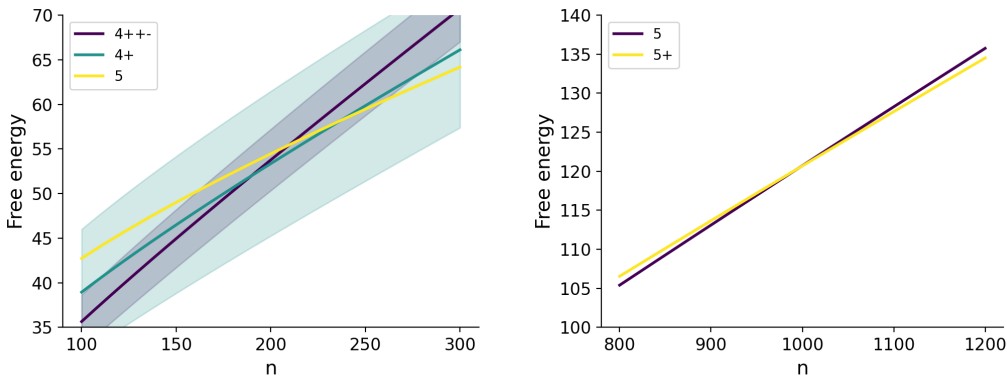

Figure D.5: Free energy plots suggesting Bayesian transitions $4^{++-} \to 4^{+}$, $4^{+} \to 5$ and $5 \to 5^{+}$.

## E  INTUITION FOR THE LOCAL LEARNING COEFFICIENT AS COMPLEXITY MEASURE FOR $c = 6$

Here we provide some intuition for why critical points in TMS with higher local learning coefficient should be thought of as more complex. It seems uncontroversial that the standard $(k+1)$-gon is more complex than the standard $k$-gon. We focus on explaining why increasing the number of positive biases on a $k$-gon causes a slight *increase* in the model complexity and increasing the number of large negative biases on the 4-gon causes a large *decrease* in the complexity. This pattern can be seen empirically in Table K.1.

The basic fact that informs this discussion is that the local learning coefficient is half the number of normal directions to the level set $L(w) = L(w_\alpha^*)$ at $w_\alpha^*$ when $L$ is Morse-Bott at $w_\alpha^*$ so that a naive count of normal directions captures the degeneracy. See (Watanabe, 2009, §7.1), (Wei et al., 2022b, §4) and (Lau et al., 2023, Appendix A) for relevant mathematical discussion. That means that if we *increase* the number of directions we can travel in the level set by 1 when we move from $w_\alpha^*$ to $w_\beta^*$ then we expect to *decrease* the learning coefficient by $\frac{1}{2}$. When the level set is more degenerate at $w_\alpha^*$ such naive dimension counts fail to be the correct measure and it is more difficult to provide simple intuitions. However in TMS we are fortunate that some of the critical points (e.g. $5, 5^+$) are minimally singular so naive counts actually do capture what is going on.

So let us do some naive counting. Recall that any positive bias $b_i$ associated with a column $W_i$ with zero norm must have the exact value $\frac{1}{2c}$, whereas negative biases associated with such columns can take on any value. Fixing the value of the bias reduces the number of free parameters by 2, since if we have a positive bias at $b_i$ then $l_i = \|W_i\|$ must be zero. This explains why the learning coefficient of the $5^+$-gon is one larger than that of the 5-gon, since both are minimally singular and we have decreased the number of free parameters (dimension of the level set) by 2.

Next we consider large negative biases. For the $4^-$-gon, note that the neuron with the large negative bias never fires (it is a "dead" neuron), so this critical point only really has representations for three inputs. In fact, when there are no positive biases, the family of parameters that we call a $4^-$-gon includes $w \in \mathcal{W}$ with (using the $l, \theta, b$ parametrization) any $b_4 < 0$ and any $l_4^2 < -b_4$ including $l_4$ arbitrarily close to zero, with a convex hull containing only three vertices. Further, in the case of the $4^{--}$-gon, this configuration only has representations for two inputs. In this case, if the two weights with negative biases are adjacent then there is an entire "dead" quarter-plane of the activation space, and the 5th and 6th columns of $W$ can take on nonzero values in that quarter plane (provided they satisfy $l_i^2 < -b_i$ for $i \in \{5, 6\}$). This extra freedom means that the number of bits required to "pin down" a $4^{--}$-gon is less than a $4^-$gon, which is less than a 4-gon. Similarly, specifying the $4^{---}$-gon and $4^{----}$-gon requires even less information, so it is appropriate that the local learning coefficient classifies them as less complex.

# F   MCMC EXPERIMENTS

We give further details about the experiments where we use MCMC to sample the Bayesian posterior to establish the phase occupancy plot in 2. For a given number of columns $c$ and sample size $n$, we generate $n$ samples $X_i$ from the true distribution $q(x)$ and obtain a likelihood function $\prod_{i=1}^n p(X_i \mid w)$ with $q(x)$ and $p(x \mid w)$ given by (2) and (3) respectively. We choose a prior on the parameter space $w = (W, b)$ to be the standard multivariate Gaussian prior (i.e. with zero mean and identity covariance matrix).

To sample from the corresponding posterior distribution, we run Markov Chain Monte Carlo (MCMC), specifically with the No U-Turn sampler (NUTS) (Homan & Gelman, 2014), with 6 randomly initialised MCMC chains, each with 5000 iterations after 500 iterations of burn-in. We thin the resulting chain by a factor of 10 resulting in a total posterior sample size of 3000. For each combination of $n$ and $c$, we run the above posterior sampling procedure for 10 different PRNG seeds, which produces different input samples $X_i$ as well as changing MCMC trajectories.

## F.1   DETAILS OF THEORETICAL PROPORTION CURVES

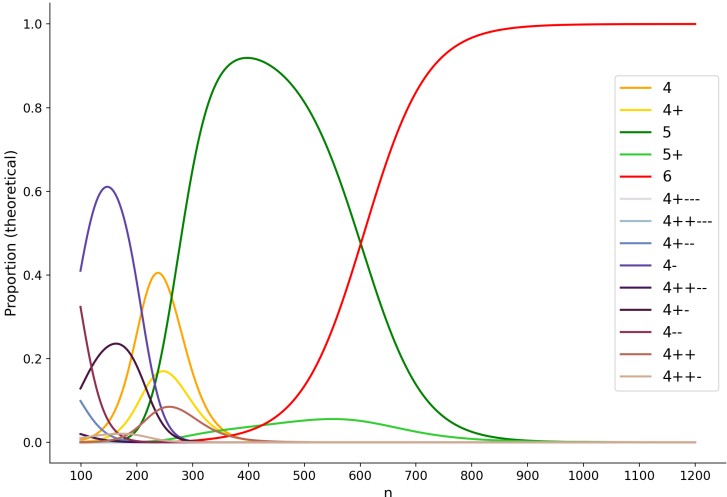

Figure F.1: Extended version of theoretical occupancy plot shown previously in Figure 2 where the effect of sub-dominant phases is now included. Note that exact theoretical values of the loss and prior contributions were used for all critical points shown, and exact values of the local learning coefficient were used for the 6-gon, $5^+$-gon and 5-gon, but estimates of the local learning coefficients were used for other critical points (Table K.1).

This section contains details of the theoretical component of Figure 2. For each $\alpha \in \{4, 4^+, 5, 5^+, 6\}$ we consider the free energy approximation

$$f_\alpha(n) = nL_\alpha + \lambda_\alpha \log n + c_\alpha$$

where $L_\alpha$ is the theoretical value taken from Section A and $c_\alpha$ are the constant terms from Table C.2. We use the theoretically derived value of $\lambda_\alpha$ in Table 1 for $\alpha \in \{5, 5^+, 6\}$ and the empirically estimated $\hat{\lambda}_\alpha$ for $\alpha \in \{4, 4^+\}$ from Table K.1. We then define

$$p_\alpha(n) = \exp(-f_\alpha(n)), \quad Z(n) = \sum_\alpha p_\alpha(n)$$

and the theory plot in Figure 2 shows the curves $\{\frac{1}{Z} p_\alpha(n)\}_\alpha$. Figure F.1 is produced in the same way, with a larger range of phases $\alpha$.

## F.2   VERIFYING DOMINANT PHASES FOR $r = 2, c = 6$

To quantify the relative frequency of each phase at a given sample size $n$, we classify all posterior samples into various phases $\mathcal{W}_{k,\sigma}$ by counting the vertices in their convex hull ($k$) and the number

of positive biases ($\sigma$), and compute the proportion of samples that falls into each phase. We then plot the frequencies of each phase as a function of sample size $n$ to visualize how preferred phases changed with $n$. Figure 2 shows the corresponding plot in the case for $r = 2, c = 6$. The lines show the frequencies of the phases $6, 5, 5^+, 4$ and $4^+$, while unclassified posterior samples are labelled as "other".

While this convex-hull and positive bias counting classification scheme is based on the characteristics of known critical points, it is only an imperfect reflection. There is the risk of mistaking posterior samples in $\mathcal{W}_{k,\sigma}$ as evidence of occupation in the $k^{\sigma+}$-gon phase when it is not. Since we do not claim to have found all possible critical points of the TMS potential and have neglected the higher loss variants of 4-gon (explained below), it is possible that MCMC samples do not reflect known phases. If this misclassification happens sufficiently often, it could invalidate the comparison of the occupancy plots with theoretical predictions.

To guard against this, we should check that every MCMC sample in $\mathcal{W}_{k,\sigma}$ is close to a known critical point in $\mathcal{W}_{k,\sigma}$, or is classified as "other". To reduce the amount of labour for this task, we run t-SNE projection (van der Maaten & Hinton, 2008) of the parameters into a 2D space with a custom metric design to remove known symmetries allowing samples that are similar to each other to show up in t-SNE projections as clusters regardless of the irrelevant differences between their angular displacement and column permutation. The custom t-SNE metric is such that distance between a pair of parameters, $(W, b), (W', b')$, is given by the sum

$$\text{HammingDistance}(b > 0, b' > 0) + \min_{i,j \in \{1,...c\}} \|\text{Normalize}(W, i) - \text{Normalize}(W, j)\|_{\text{Frobenius}}$$

where $b > 0$ denotes the binary array $(\mathbb{1}_{b_1 > 0}, \dots, \mathbb{1}_{b_c > 0})$ and $\text{Normalise}(W, i)$ denotes a normalised weight matrix where all column vectors are rotated by a fixed angle so that $i^{th}$ column vector is aligned with the positive x-axis and the columns are reordered so that the column indices reflects the order of the vectors when read counter-clockwise starting from the positive x-axis.

With this, we can verify the occupancy of dominant phases by checking several samples in each cluster to verify the phase classification of the entire cluster. To illustrate, let us verify the phase occupancy for $c = 6$ at $n = 1000$ as shown in Figure 2. Figure F.2 shows the t-SNE projection of the samples for a particular MCMC run. Looking at both the theoretical and empirical occupancy curves at $n = 1000$, the posterior is dominated by the 6-gon, followed by the 5-gon and then the $5^+$-gon. Looking at various samples in the largest (green) t-SNE cluster, they do correspond to the 6-gon all with biases near the optimal negative value. The minor cluster (in dark purple) corresponds to the 5-gon. This cluster of 5-gons includes samples with a sixth "vestigial leg" with non-zero length. However, these belong to the same phase (same critical submanifold as the 5-gon) since the corresponding bias has large negative value. The t-SNE projection also reveals a small number of $5^+$-gon samples.

Performing similar inspections for MCMC chains at $n = 500, 700$ allows us to confirm that the dominant phase switches from the 5 to 6-gon in the interval $600 \leq n \leq 700$. This inspection also confirms that clusters of $5^+$-gons coexist with the two dominant phases albeit at a much lower probability.

For sufficiently low values of $n$, we encounter two issues in establishing phase occupancy.

1. Other higher loss phases such as variants of the 4-gon with large negative biases, and potentially other higher energy phases that we have not characterised start to have non-negligible occupancy.

2. As $n$ becomes lower, the posterior distribution becomes less concentrated. This means that significantly more posterior mass, and hence a higher fraction of MCMC samples, is accounted for by regions of parameter space that are further away from critical points. These points may be close to the boundaries between different regions, increasing the chance of misclassification, or they may bear little resemblance to the critical point associated with the region they are classified into.

For $n > 400$, from inspecting t-SNE clusters, the above issues do not arise: the samples are close to known critical points, and the frequency of unclassified "other" samples is low enough that it won't significantly affect the relative frequency of the dominant phases. Furthermore, we also do not observe many samples that are close to high loss 4-gon variants. This supports the prediction

depicted in the extended theoretical occupancy curve shown in Figure F.1 which suggests that these 4-gon variants only show up in the $n < 400$ regime.

We caution the reader in regards to interpreting the phase occupancy diagrams for the range $100 < n < 400$ where one or more of the issues above could affect the empirical frequency.

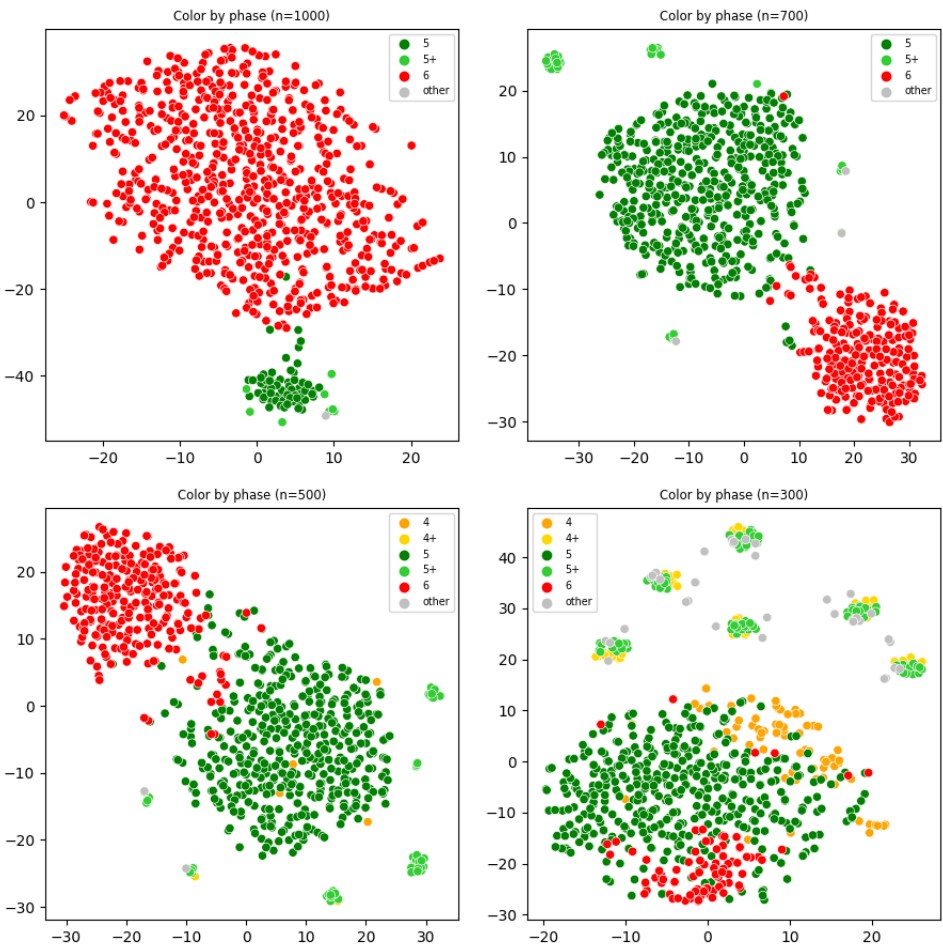

Figure F.2: t-SNE plots of MCMC samples from the posterior at a range of sample sizes $n$ encompassing the phase transition from the 5-gon to the 6-gon.

### F.3 MCMC Health

MCMC sampling for high dimensional posterior distributions is challenging. In our case there is the added challenge of the posterior being multi-modal (the posterior density has local maxima at the dominant phases) where the modes are not points, but submanifolds of varying codimension. To ensure that the proportion of MCMC samples that falls into $\mathcal{W}_{k,\sigma}$ is a good reflection of the probability of $\mathcal{W}_{k,\sigma}$, we need to make sure that our Markov chains are well mixed.

For this purpose, we produce and check two different types of diagnostic plots for each MCMC run:

- **Theoretical loss trace plots.** We plot the theoretical loss of each MCMC sample against its sample index which orders the samples in each MCMC chain in increasing order of MCMC iterations required to generate the sample. An unhealthy MCMC chain will show up on such a plot as points occupying a very narrow band of theoretical loss values.

- **Phase type trace plots.** On the same trace plots, we color each sample by their phase classification. Successful posterior sampling should produce samples in each phase with a

frequency that is roughly the same as the posterior probability of that phase. While we do not know the true probability of a given phase, we can cross reference each MCMC chain with other chains performing sampling on the same posterior to see that every MCMC chain visits phases discovered by any other MCMC chain. An unhealthy MCMC chain will show up on such a plot as a chain that only contains samples of one phase type when there is more than one phase type observed across all chains.

Figure F.3 shows a examples of such diagnostic trace plots for a few experiments (with $c = 6$ and matching those in Figure F.2) at $n = 300, 1000$. All six chains run in these experiments are plotted on the same plot and distinguished by color. At the higher sample size $n = 1000$, we expect and do indeed observe that a particular phase, the 6-gon, dominates the posterior but every chain visits sub-dominant phases as well.

The diagnostics detect no sign of problems for the experiments used to establish the phase occupancy curves in Figures 2, F.5 and F.6. However, we do observe that MCMC fails for sample sizes $n$ significantly greater than those we report in this paper. With large sample sizes, the posterior distribution becomes highly concentrated at each phase, posing a significant challenge for an MCMC chain to escape its starting point (controlled by random initialization and the trajectory of the burn-in phase). Figure F.4 shows an example at $n = 4000$, where we see

- A chain (colored pink) which, for many iterations, produces samples in a very narrow band of loss.

- Most chains have a starting point falling into the $5^+$-gon phase and rarely escape (only the red chain found the lower loss 6-gon region).

- The proportion of 6-gons is mostly determined by how many chains have their starting point already in the 6-gon phase. In this run, this proportion is dominated by the last orange chain.

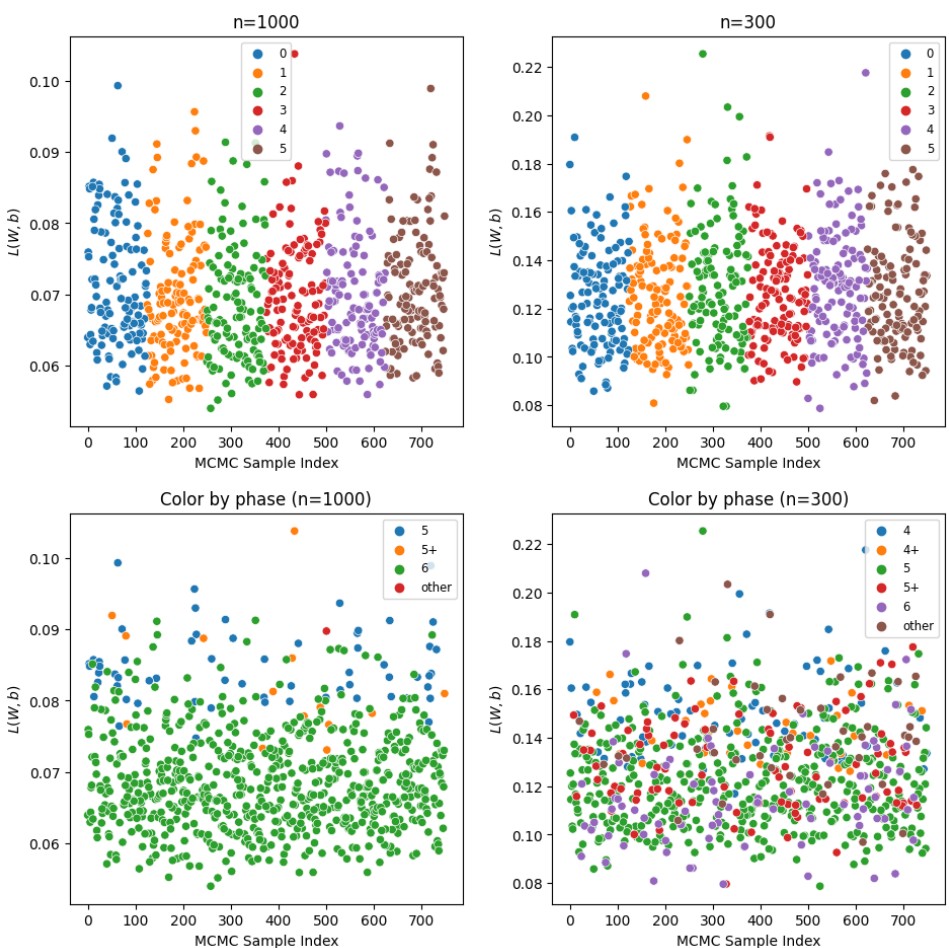

Figure F.3: Trace plots displaying theoretical loss of the MCMC samples ordered by their MCMC iteration number and colored by MCMC chain index (top) and the same scatter plot but colored by phase classification (bottom).

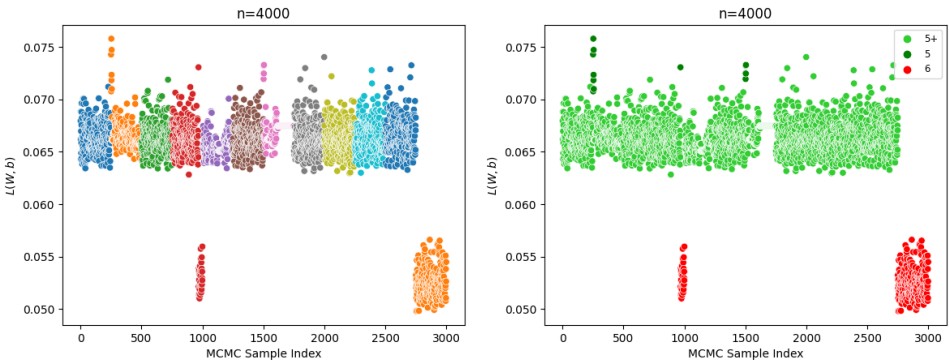

Figure F.4: Unhealthy trace plots.

### F.4 THEORY AND EXPERIMENTS FOR $c = 4$ AND $c = 5$

In this section we repeat the analysis of Section 4.2 for $c = 4$ and $c = 5$ with the same experimental setup as explained at the beginning of that section. When $c = 4$ the 4-gon is a true parameter (it has zero loss) and any 3-gon is not, so the theory predicts that the 4-gon must dominate the posterior for all $n$, as seen in Figure F.5.

| Critical point | Local learning coefficient $\lambda$ | Loss $L$ |
|---|---|---|
| 4-gon | 4, 4.5, 5, 5.5 | 0 |

Table F.1: $r = 2, c = 4$

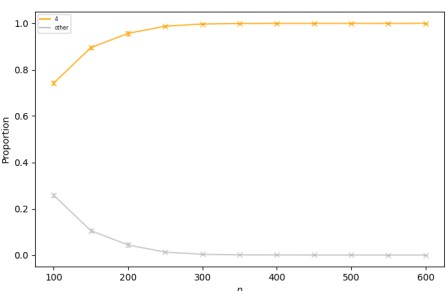

Figure F.5: $r = 2, c = 4$. The standard 4-gon dominates for all $n$.

When $c = 5$ the theory and experimental curves in Figure F.6 show the $4 \to 5$ transition. Note that $4^+$ is correctly predicted to never dominate the posterior despite having lower energy than the standard 4-gon.

| Critical point | Local learning coefficient $\lambda$ | Loss $L$ |
|---|---|---|
| 4-gon | 4, 4.5, 5, 5.5 | 0.06667 |
| $4^+$-gon | 5, 5.5, 6, 6.5 | 0.05667 |
| $5^-$-gon | 7 | 0.01583 |

Table F.2: $r = 2, c = 5$

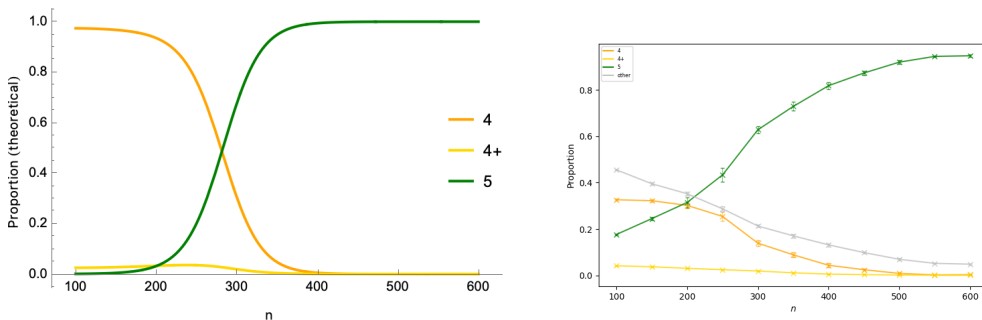

Figure F.6: Proportion of Bayesian posterior density concentrated in regions $\mathcal{W}_{k,\sigma}$ associated to $k$-gons, as a function of the number $n$ of samples for $r = 2, c = 5$.

We note that the classification of MCMC samples in $c = 4, 5$ described in this section is slightly different from what was described for $c = 6$ in Section 4.2. The main reason being that we need to handle variants of the 4-gon more carefully in $c = 4, 5$.

- For $c = 4$, we classify a sample $(W, b)$ only by the number of vertices on the convex hull formed by the column vectors with no additional subcategories defined by the number

of positive biases. Theoretically we know that there are no critical 4-gons with positive bias and the standard 4-gon is a critical point with all zero bias and is thus susceptible to misclassification even with slight perturbation when the the number of positive biases is counted.

- For $c = 5$, the situation is similar except for one extra case where need to allow for the possibility of a $4^+$-gon. A sample $(W, b)$ is classified as a $4^+$-gon when it has 4 vertices in its convex hull, and if $b_i > 0$ then $l_i = \|W_i\| < 0.5$.

In the cases $c = 4, 5$ we also manually verify the dominant phases by visually inspecting t-SNE clusters of MCMC samples at multiple sample sizes.

# G    POTENTIAL IN LOCAL COORDINATES

*Proof of Lemma 3.1.* By definition

$$L(W, b) = \frac{1}{c} \left( \sum_{i \neq j} \int_0^1 \text{ReLU}(W_j \cdot W_i x_i + b_j)^2 dx_i \right.$$
$$\left. + \sum_{i=1}^c \int_0^1 \left( x_i - \text{ReLU}(\|W_i\|^2 x_i + b_i) \right)^2 dx_i \right).$$

Let $i, j \in \{1, \ldots, c\}$ be such that $i \neq j$. To compute the integral

$$\int_0^1 \text{ReLU}(W_j \cdot W_i x_i + b_j)^2 dx_i,$$

we first find the region in $[0, 1]$ on which $W_j \cdot W_i x_i + b_j \geq 0$. Let

$$D_{j,i} = \{x_i \in [0, 1] \,|\, W_j \cdot W_i x_i + b_j \geq 0\}.$$

1. If $W_j \cdot W_i > 0$ and $-W_j \cdot W_i \leq b_j \leq 0$, then
$$D_{j,i} = \left[ \frac{-b_j}{W_j \cdot W_i}, 1 \right].$$

2. If $W_j \cdot W_i > 0$ and $b_j \leq -W_j \cdot W_i$, then
$$D_{j,i} = \emptyset.$$

3. If $W_j \cdot W_i = 0$ and $b_j = 0$, then
$$D_{j,i} = [0, 1].$$
Note that in this case, $W_j \cdot W_i x_i + b_j = 0$ for all $x_i \in [0, 1]$.

4. If $W_j \cdot W_i = 0$ and $b_j < 0$, then
$$D_{j,i} = \emptyset.$$

5. If $W_j \cdot W_i < 0$ and $b_j \leq 0$, then
$$D_{j,i} = \emptyset.$$

6. If $W_j \cdot W_i > 0$ and $b_j > 0$, then
$$D_{j,i} = [0, 1].$$

7. If $W_j \cdot W_i = 0$ and $b_j > 0$, then
$$D_{j,i} = [0, 1].$$

8. If $W_j \cdot W_i < 0$ and $b_j \geq -W_j \cdot W_i > 0$, then
$$D_{j,i} = [0, 1].$$

9. If $W_j \cdot W_i < 0$ and $-W_j \cdot W_i > b_j > 0$, then
$$D_{j,i} = \left[ 0, \frac{-b_j}{W_j \cdot W_i} \right].$$

Recall the definition of $P_{j,i}$, $P_i$, and $Q_{j,i}$ from (Lemma 3.1). Then for $b_j \leq 0$,

$$\int_0^1 \text{ReLU}(W_j \cdot W_i x_i + b_j)^2 dx_i = \delta(P_{j,i}) \int_{-b_j/(W_j \cdot W_i)}^1 (W_j \cdot W_i x_i + b_j)^2 dx_i$$
$$= \delta(P_{j,i}) \left[ \frac{1}{3W_j \cdot W_i} (W_j \cdot W_i x_i + b_j)^3 \right]_{-b_j/(W_j \cdot W_i)}^1$$
$$= \delta(P_{j,i}) \frac{1}{3W_j \cdot W_i} (W_j \cdot W_i + b_j)^3.$$

and for $b_j > 0$,

$$\int_0^1 \text{ReLU}(W_j \cdot W_i x_i + b_j)^2 dx_i = \delta(Q_{j,i}) \int_0^{-b_j/(W_j \cdot W_i)} (W_j \cdot W_i x_i + b_j)^2 dx_i$$

$$+ \left(1 - \delta(Q_{j,i})\right) \int_0^1 (W_j \cdot W_i x_i + b_j)^2 dx_i$$

$$= \delta(Q_{j,i}) \left[ \frac{1}{3W_j \cdot W_i} (W_j \cdot W_i x_i + b_j)^3 \right]_0^{-b_j/(W_j \cdot W_i)}$$

$$+ \left(1 - \delta(Q_{j,i})\right) \left[ \frac{1}{3W_j \cdot W_i} (W_j \cdot W_i x_i + b_j)^3 \right]_0^1$$

$$= \delta(Q_{j,i}) \frac{1}{3} \left( \frac{-b_j^3}{W_j \cdot W_i} \right)$$

$$+ \left(1 - \delta(Q_{j,i})\right) \frac{1}{3} [(W_j \cdot W_i)^2 + 3(W_j \cdot W_i)b_j + 3b_j^2]$$

It remains to compute

$$\int_0^1 \left( x_i - \text{ReLU}(\|W_i\|^2 x_i + b_i) \right)^2 dx_i$$

for each $i \in \{1, \ldots, c\}$. We first find the region in $[0, 1]$ on which $\|W_i\|^2 x_i + b_i \geq 0$. Let

$$D_i = \{x_i \in [0, 1] \mid \|W_i\|^2 x_i + b_i \geq 0\}.$$

1. If $\|W_i\|^2 > 0$ and $-\|W_i\|^2 \leq b_i \leq 0$, then

$$D_i = \left[ \frac{-b_i}{\|W_i\|^2}, 1 \right].$$

2. If $\|W_i\|^2 > 0$ and $b_i \leq -\|W_i\|^2$, then

$$D_i = \emptyset.$$

In this case

$$\int_0^1 \left( x_i - \text{ReLU}(\|W_i\|^2 x_i + b_i) \right)^2 dx_i = \int_0^1 x_i^2 dx_i = \frac{1}{3}.$$

3. If $\|W_i\|^2 = 0$ and $b_i = 0$, then

$$D_i = [0, 1].$$

Note that in this case, $\|W_i\|^2 x_i + b_i = 0$ for all $x_i \in [0, 1]$. So

$$\int_0^1 \left( x_i - \text{ReLU}(\|W_i\|^2 x_i + b_i) \right)^2 dx_i = \int_0^1 x_i^2 dx_i = \frac{1}{3}.$$

4. If $\|W_i\|^2 = 0$ and $b_i < 0$, then

$$D_i = \emptyset.$$

In this case

$$\int_0^1 \left( x_i - \text{ReLU}(\|W_i\|^2 x_i + b_i) \right)^2 dx_i = \int_0^1 x_i^2 dx_i = \frac{1}{3}.$$

5. If $\|W_i\|^2 > 0$ and $b_i > 0$, then

$$D_i = [0, 1].$$

6. If $\|W_i\|^2 = 0$ and $b_i > 0$, then

$$D_i = [0, 1].$$

For $i$ with $b_i \leq 0$, on $P_i$,

$$\int_0^1 \left(x_i - \text{ReLU}(\|W_i\|^2 x_i + b_i)\right)^2 dx_i = \int_{-b_i/\|W_i\|^2}^1 (x_i - \|W_i\|^2 x_i - b_i)^2 dx_i$$

$$+ \int_0^{-b_i/\|W_i\|^2} x_i^2 dx_i$$

$$= \int_{-b_i/\|W_i\|^2}^1 \left((1 - \|W_i\|^2)x_i - b_i\right)^2 dx_i$$

$$+ \left[\frac{1}{3} x_i^3\right]_0^{-b_i/\|W_i\|^2}.$$

If $\|W_i\| \neq 1$, then the integral is equal to

$$\frac{1}{3} \left\{ (1 - \|W_i\|^2)^2 - 3(1 - \|W_i\|^2)b_i + 3b_i^2 + \frac{b_i^3}{\|W_i\|^4} + \frac{b_i^3}{\|W_i\|^2} \right\}.$$

If $\|W_i\| = 1$, then the integral is equal to

$$\frac{1}{3}(3b_i^2 + 2b_i^3).$$

Since

$$\lim_{\|W_i\| \to 1} \frac{1}{3} \left\{ (1 - \|W_i\|^2)^2 - 3(1 - \|W_i\|^2)b_i + 3b_i^2 + \frac{b_i^3}{\|W_i\|^4} + \frac{b_i^3}{\|W_i\|^2} \right\}$$

$$= \frac{1}{3}(3b_i^2 + 2b_i^3),$$

We know that in $P_i$,

$$\int_0^1 \left(x_i - \text{ReLU}(\|W_i\|^2 x_i + b_i)\right)^2 dx_i$$

$$= \frac{1}{3} \left\{ (1 - \|W_i\|^2)^2 - 3(1 - \|W_i\|^2)b_i + 3b_i^2 + \frac{b_i^3}{\|W_i\|^4} + \frac{b_i^3}{\|W_i\|^2} \right\}.$$

For $i$ with $b_i > 0$,

$$\int_0^1 \left(x_i - \text{ReLU}(\|W_i\|^2 x_i + b_i)\right)^2 dx_i = \int_0^1 \left(x_i - (\|W_i\|^2 x_i + b_i)\right)^2 dx_i$$

$$= \frac{1}{3(1 - \|W_i\|^2)} \left\{ [(1 - \|W_i\|^2) - b_i]^3 + b_i^3 \right\}$$

$$= \frac{1}{3} \left[ (1 - \|W_i\|^2)^2 - 3(1 - \|W_i\|^2)b_i + 3b_i^2 \right]$$

Thus, $L(W, b) = \frac{1}{3c} H(W, b)$ as claimed. $\qquad \square$

Now we introduce a new coordinate system of the parameter space which is used to analyse the local geometry around a critical point. Let $\mathcal{C} = \{\{i, j\} \mid i \neq j \in \{1, 2, \ldots, c\}\}$. For a subset $C \subset \mathcal{C}$, define a subset $W_C$ of $M_{r,c}(\mathbb{R})$, called a **chamber**, by

$$W_C = \{W \in M_{r,c}(\mathbb{R}) \mid W_i \cdot W_j > 0 \text{ if and only if } \{i, j\} \in C\}.$$

Note that

1. Some subsets $C$ of $\mathcal{C}$ define an empty chamber $W_C = \emptyset$. For example, when $r = 2$ and $c = 4$, the set

$$C = \{\{1, 2\}, \{2, 3\}, \{3, 4\}, \{4, 1\}\} \subset \mathcal{C}$$

   defines an empty chamber because within this set, to satisfy $W_i \cdot W_{i+1} > 0, W_{i+1} \cdot W_{i+2} > 0, W_i \cdot W_{i+2} \leq 0$ we must have that $W_{i+1}$ is between $W_i$ and $W_{i+1}$ on the circle. Therefore we require that the sum of angles between $W_i$ and $W_{i+1} = 2\pi$, but each of these 4 angles must be more than $\pi/2$ so the configuration isn't possible.

2. if $C \subset \mathcal{C}$ defines a nonempty chamber $W_C$ containing a $W$ with
$$W_i \cdot W_j > 0 \; \forall \{i, j\} \in C \text{ and } W_u \cdot W_v < 0 \; \forall \{u, v\} \notin C,$$
then $W_C$ contains the open subset
$$\{W \in M_{r,c}(\mathbb{R}) \,|\, W_i \cdot W_j > 0 \; \forall \{i, j\} \in C \text{ and } W_u \cdot W_v < 0 \; \forall \{u, v\} \notin C\}$$
of $M_{r,c}(\mathbb{R})$;

3. if $C \subset \mathcal{C}$ defines a nonempty chamber $W_C$, then for any $Q \subset C$,
$$\{W \in M_{r,c}(\mathbb{R}) \,|\, W_i \cdot W_j = 0 \; \forall \{i, j\} \in Q \text{ and } W_u \cdot W_v > 0 \Leftrightarrow \{u, v\} \in C \backslash Q\}$$
defines a boundary of $W_C$;

4. for any distinct subsets $C, Q$ of $\mathcal{C}$, $W_C \cap W_Q = \emptyset$;

5. the union of all chambers cover $M_{r,c}(\mathbb{R})$.

Consider a nonempty chamber $W_C$ for some $C \subset \mathcal{C}$. Suppose that $W_C$ contains an open subset of $M_{r,c}(\mathbb{R})$. Let $i \neq j \in \{1, 2, \ldots, c\}$. If $\{j, i\} \notin C$, then for all $W \in W_C$,
$$\delta(P_{j,i}) = 0.$$
Thus, in $W_C \times \mathbb{R}^c$, for each $i \in \{1, 2, \ldots, c\}$, $H_i^-(W, b)$ (see Lemma 3.1) is given by

$$H_i^-(W, b) = \sum_{j \neq i : \{j, i\} \in C} \delta(P_{i,j}) \left[ \frac{1}{W_i \cdot W_j} (W_i \cdot W_j + b_i)^3 \right]$$
$$+ \delta(P_i) \left[ \frac{b_i^3}{\|W_i\|^4} + \frac{b_i^3}{\|W_i\|^2} \right] + (1 - \delta(P_i)) + \delta(P_i) N_i.$$

Now we focus on the case $r = 2$. Let $W \in M_{2,c}(\mathbb{R})$. Then $W$ is contained in some chamber $W_C$. In the new parametrization $(l, \theta)$, we can describe the chamber $W_C$ in a different way. Let $(l_1, \ldots, l_c, \theta_1, \ldots, \theta_c)$ be the coordinate of $W$. For each $i = 1, \ldots, c$, a **wedge** $M_{ij}$ is defined by

$$M_{ij} = \begin{cases} (i, i+1, \ldots, i+j-1), & \text{if } j > 0 \text{ and } \theta_i + \cdots + \theta_{i+j-1} < \frac{\pi}{2}; \\ \emptyset, & \text{if } j = 0 \text{ or } \theta_i + \cdots + \theta_{i+j-1} \geq \frac{\pi}{2}, \end{cases}$$

where addtions are computed cyclically. For each $i = 1, \ldots, c$, let $t(i)$ denote the integer such that

$$\theta_i + \cdots + \theta_{i+t(i)-1} < \frac{\pi}{2} \text{ and } \theta_i + \cdots + \theta_{i+t(i)} \geq \frac{\pi}{2}.$$

We use the convention where $t(i) = 0$ if $\theta_i \geq \pi/2$. Then $M_{ij} = \emptyset$ for all $j \geq t(i)$. So we can list the set of all wedges $\mathcal{M} = \{M_{ij}\}$ into a table:

| $M_{11}$ | $M_{12}$ | $\cdots$ | $\cdots$ | $M_{1t(1)}$ |
|---|---|---|---|---|
| $M_{21}$ | $M_{21}$ | $\cdots$ | $\cdots$ | $M_{2t(2)}$ |
| $\vdots$ | | | | |
| $\vdots$ | | | | |
| $M_{(c-1)1}$ | $M_{(c-1)2}$ | $\cdots$ | $\cdots$ | $M_{(c-1)t(c-1)}$ |
| $M_{c1}$ | $M_{c2}$ | $\cdots$ | $\cdots$ | $M_{ct(c)}$ |

This set of wedges $\mathcal{M}$ describes a chamber $W_C$ containing $W$, where
$$C = \big\{ \{1, 2\}, \{1, 3\}, \ldots, \{1, t(1)+1\}, \{2, 3\}, \{2, 4\}, \ldots, \{2, t(2)+2\}, \ldots,$$
$$\ldots, \{c-1, c\}, \ldots, \{c-1, t(c-1)+c-1\}, \{c, 1\}, \ldots, \{c, t(c)+c\} \big\},$$
and the addtions are computed cyclically. For example, if $c = 5$, then a 5-gon has coordinate
$$l_1 = l_2 = l_3 = l_4 = l_5;$$
$$\theta_1 = \theta_2 = \theta_3 = \theta_4 = \frac{2\pi}{5}.$$
It is contained in the interior of the chamber $W_C$, where
$$C = \{\{1, 2\}, \{2, 3\}, \{3, 4\}, \{4, 5\}, \{5, 1\}\}.$$
This chamber is described by the set of wedges:

$$\begin{array}{|c|}\hline (1) \\\hline (2) \\\hline (3) \\\hline (4) \\\hline (5) \\\hline\end{array}.$$

Now let $W_C \subset M_{2,c}(\mathbb{R})$ be a nonempty chamber. Suppose that $W_C$ contains an open subset of $M_{2,c}(\mathbb{R})$. Let $\mathcal{M}$ be the set of wedges describing $W_C$. Set

1. $T_{M_{ij}}^{(1)} = \left\{ (l,\theta,b) \,|\, -l_i l_{i+j} \cos\left(\sum_{k \in M_{ij}} \theta_k\right) \le b_{i+j} \right\}$;

2. $T_{M_{ij}}^{(2)} = \left\{ (l,\theta,b) \,|\, -l_i l_{i+j} \cos\left(\sum_{k \in M_{ij}} \theta_k\right) \le b_i \right\}$;

3. $T_i = \{ (l,\theta,b) \,|\, -l_i^2 \le b_i \}$;

4. $S_{ij} = \{ (l,\theta,b) \,|\, -l_i l_j \cos(\theta_i + \cdots \theta_{j-1}) > b_i > 0 \}$.

Then in $W_C \times \mathbb{R}^c$, the TMS potential $H(l,\theta,b)$ in the new parametrization is

$$H(l,\theta,b) = \sum_{i=1}^{c} \delta(b_i \le 0) H_i^-(l,\theta,b) + \delta(b_i > 0) H_i^+(l,\theta,b), \tag{17}$$

where

$$
\begin{aligned}
H_i^-(l,\theta,b) =& \sum_{j: M_{(i-j)j} \in \mathcal{M}} \delta(T_{M_{(i-j)j}}^{(1)}) \frac{\left[ l_{i-j} l_i \cos\left(\sum_{k \in M_{(i-j)j}} \theta_k\right) + b_i \right]^3}{l_{i-j} l_i \cos\left(\sum_{k \in M_{(i-j)j}} \theta_k\right)} \\
&+ \sum_{j: M_{ij} \in \mathcal{M}} \delta(T_{M_{ij}}^{(2)}) \frac{\left[ l_i l_{i+j} \cos\left(\sum_{k \in M_{ij}} \theta_k\right) + b_i \right]^3}{l_i l_{i+j} \cos\left(\sum_{k \in M_{ij}} \theta_k\right)} \\
&+ \delta(T_i) \left[ (1 - l_i^2)^2 - 3(1 - l_i^2) b_i + 3 b_i^2 + \frac{b_i^3}{l_i^4} + \frac{b_i^3}{l_i^2} \right] + \left(1 - \delta(T_i)\right)
\end{aligned}
$$

and

$$
\begin{aligned}
H_i^+(l,\theta,b) =& \sum_{j \ne i} \delta(S_{ij}) \left[ \frac{-b_i^3}{l_i l_j \cos(\theta_i + \theta_{i+1} + \cdots \theta_{j-1})} \right] \\
&+ \left(1 - \delta(S_{ij})\right) \left[ \left( l_i l_j \cos(\theta_{ij}) \right)^2 + 3\left( l_i l_j \cos(\theta_{ij}) \right) b_i + 3 b_i^2 \right] \\
&+ \left[ (1 - l_i^2)^2 - 3(1 - l_i^2) b_i + 3 b_i^2 \right],
\end{aligned}
$$

where $\theta_{ij} = \theta_i + \theta_{i+1} + \cdots + \theta_{j-1}$ is the angle between $W_i$ and $W_j$.

**Remark G.1.** *If a critical point of $H$ is contained in the interior of a chamber (see Appendix H.1), then there is an open neighbourhood of the critical point in which $H$ is of the above form. However, critical points are not always contained in the interior of some chamber. If a critical point is contained in the boundary of different chambers (see Appendix J and Appendix H.3), then extra efforts are required for analysing the local geometry around the critical point.*

# H    DERIVATION OF LOCAL LEARNING COEFFICIENTS

## H.1    $k$-GONS WITH NEGTIVE BIAS FOR $k = c \notin 4\mathbb{Z}$

In this section, we show that for $c \in \{5, 6, 7\}$, the $c$-gon with coordinate

$$l_1 = \cdots = l_c = x^*, \quad \theta_1 = \cdots = \theta_c = \alpha = \frac{2\pi}{c}, \quad b_1 = \cdots = b_c = y^*,$$

where the values of $x^*$ and $y^*$ is given in Table H.1, is a non-degenerate critical point of the TMS potential. Therefore, the local learning coefficient of $c$-gon is $(3c - 1)/2$.

| $c$ | $x^*$ | $y^*$ |
|---|---|---|
| 5 | 1.17046 | $-0.28230$ |
| 6 | 1.32053 | $-0.61814$ |
| 7 | 1.44839 | $-0.96691$ |

Table H.1: Parameters of $c$-gons.

Let $c$ be an integer greater than or equal to $4$. We consider the case where $c$ is not a multiple of $4$. Consider the $c$-gon with coordinate $(l^*, \theta^*, b^*)$

$$l^* : l_1 = \cdots = l_c = x, \quad \theta^* : \theta_1 = \cdots = \theta_c = \alpha = \frac{2\pi}{c}, \quad b^* : b_1 = \cdots = b_c = y,$$

for some $x > 0$ and $y < 0$. The chamber containing $c$-gons is described by the following wedges (see Appendix G):

| $(1)$ | $(1, 2)$ | $\cdots$ | $\cdots$ | $(1, 2, \ldots, s)$ |
|---|---|---|---|---|
| $(2)$ | $(2, 3)$ | $\cdots$ | $\cdots$ | $(2, 3, \ldots, s + 1)$ |
| $\vdots$ | $\vdots$ | $\cdots$ | $\cdots$ | $\vdots$ |
| $(c + 1 - s)$ | $(c + 1 - s, c + 2 - s)$ | $\cdots$ | $\cdots$ | $(c + 1 - s, c + 2 - s, \ldots, c)$ |
| $(c + 2 - s)$ | $(c + 2 - s, c + 3 - s)$ | $\cdots$ | $\cdots$ | $(c + 2 - s, c + 3 - s, \ldots, c, 1)$ |
| $\vdots$ | $\vdots$ | $\cdots$ | $\cdots$ | $\vdots$ |
| $(c - 1)$ | $(c - 1, c)$ | $\cdots$ | $\cdots$ | $(c - 1, c, 1, \ldots, s - 2)$ |
| $(c)$ | $(c, 1)$ | $\cdots$ | $\cdots$ | $(c, 1, 2, \ldots, s - 1)$ |

where $s$ is the unique integer in the interval $\left[\frac{c}{4} - 1, \frac{c}{4}\right)$. Let $M_{ij}$ be the wedge in the $(i, j)$-position in the above table. Then $M_{ij} = (i, i + 1, \ldots, i + j - 1)$, where additions are computed cyclically. Then the local TMS potential (see Appendix G) is

$$H(l, \theta, b) = \sum_{M_{ij} \in \mathcal{M}} \delta(T^{(1)}_{M_{ij}}) \frac{1}{l_i l_{i+j} \cos\left(\sum_{k \in M_{ij}} \theta_k\right)} \left(l_i l_{i+j} \cos\left(\sum_{k \in M_{ij}} \theta_k\right) + b_{i+j}\right)^3$$

$$+ \sum_{M_{ij} \in \mathcal{M}} \delta(T^{(2)}_{M_{ij}}) \frac{1}{l_i l_{i+j} \cos\left(\sum_{k \in M_{ij}} \theta_k\right)} \left(l_i l_{i+j} \cos\left(\sum_{k \in M_{ij}} \theta_k\right) + b_i\right)^3$$

$$+ \sum_{i=1}^{c} \delta(T_i) \left[(1 - l_i^2)^2 - 3(1 - l_i^2) b_i + 3 b_i^2 + \frac{b_i^3}{l_i^4} + \frac{b_i^3}{l_i^2}\right] + (1 - \delta(T_i)),$$

where

1. $T^{(1)}_{M_{ij}} = \left\{(l, \theta, b) \mid -l_i l_{i+j} \cos\left(\sum_{k \in M_{ij}} \theta_k\right) \leq b_{i+j}\right\}$;

2. $T^{(2)}_{M_{ij}} = \left\{(l, \theta, b) \mid -l_i l_{i+j} \cos\left(\sum_{k \in M_{ij}} \theta_k\right) \leq b_i\right\}$;

3. $T_i = \{(l, \theta, b) \mid -l_i^2 \leq b_i\}$;

4. $\theta_1 + \theta_2 + \cdots + \theta_c = 2\pi$.

Consider the open subset of the parameter space defined by

$$-l_i l_{i+j} \cos\left(\sum_{k \in M_{ij}} \theta_k\right) < b_{i+j}, \quad -l_i l_{i+j} \cos\left(\sum_{k \in M_{ij}} \theta_k\right) < b_i$$

for all $M_{ij}$ and

$$-l_i^2 < b_i$$

for all $i = 1, \ldots, c$. In this open subset, we have

$$H(l, \theta, b) = \sum_{M_{ij}} \frac{1}{l_i l_{i+j} \cos\left(\sum_{k \in M_{ij}} \theta_k\right)} \left(l_i l_{i+j} \cos\left(\sum_{k \in M_{ij}} \theta_k\right) + b_{i+j}\right)^3$$

$$+ \sum_{M_{ij}} \frac{1}{l_i l_{i+j} \cos\left(\sum_{k \in M_{ij}} \theta_k\right)} \left(l_i l_{i+j} \cos\left(\sum_{k \in M_{ij}} \theta_k\right) + b_i\right)^3$$

$$+ \sum_{i=1}^{c} \left[(1 - l_i^2)^2 - 3(1 - l_i^2)b_i + 3b_i^2 + \frac{b_i^3}{l_i^4} + \frac{b_i^3}{l_i^2}\right],$$

with constraint $\theta_1 + \theta_2 + \cdots + \theta_c = 2\pi$. It follows from the Lagrangian multiplier method that a point $(l^*, \theta^*, b^*)$ is a critical point of $H(l, \theta, b)$ with constraint $\theta_1 + \cdots + \theta_c = 2\pi$ if and only if there exists $\lambda \in \mathbb{R}$ such that for all $a = 1, 2, \ldots, c$,

1. $\left.\frac{\partial}{\partial \theta_a}\right|_{l=l^*, \theta=\theta^*, b=b^*} H(l, \theta, b) = \lambda$;

2. $\left.\frac{\partial}{\partial l_a}\right|_{l=l^*, \theta=\theta^*, b=b^*} H(l, \theta, b) = 0$;

3. $\left.\frac{\partial}{\partial b_a}\right|_{l=l^*, \theta=\theta^*, b=b^*} H(l, \theta, b) = 0$.

We compute the partial derivative of $H(l, \theta, b)$ with respect to $b_a$:

$$\frac{\partial}{\partial b_a} H(l, \theta, b) = \sum_{M_{ij}: i+j=a} \frac{3}{l_i l_{i+j} \cos\left(\sum_{k \in M_{ij}} \theta_k\right)} \left(l_i l_{i+j} \cos\left(\sum_{k \in M_{ij}} \theta_k\right) + b_{i+j}\right)^2$$

$$+ \sum_{M_{ij}: i=a} \frac{3}{l_i l_{i+j} \cos\left(\sum_{k \in M_{ij}} \theta_k\right)} \left(l_i l_{i+j} \cos\left(\sum_{k \in M_{ij}} \theta_k\right) + b_i\right)^2$$

$$- 3(1 - l_a^2) + 6b_a + \frac{3}{l_a^4} b_a^2 + \frac{3}{l_a^2} b_a^2.$$

We list all $M_{ij}$ with $i = a$ and all $M_{ij}$ with $i + j = a$:

| $i = a:$ | $(a)$ | $(a, a+1)$ | $\cdots$ | $\cdots$ | $(a, a+1, \ldots, a+s-1)$ |
|---|---|---|---|---|---|
| $i+j = a:$ | $(a-1)$ | $(a-2, a-1)$ | $\cdots$ | $\cdots$ | $(a-s, \ldots, a-2, a-1)$ |

Then

$$\left.\frac{\partial}{\partial b_a}\right|_{l=l^*, \theta=\theta^*, b=b^*} H(l, \theta, b) = x^2 \left(3 + 6\sum_{j=1}^{s} \cos(j\alpha)\right) + \frac{y^2}{x^2}\left(3 + 6\sum_{j=1}^{s} \frac{1}{\cos(j\alpha)}\right)$$

$$+ y(12s + 6) + 3\frac{y^2}{x^4} - 3.$$

Multiplying both sides of the equation

$$x^2 \left( 3 + 6 \sum_{j=1}^{s} \cos(j\alpha) \right) + \frac{y^2}{x^2} \left( 3 + 6 \sum_{j=1}^{s} \frac{1}{\cos(j\alpha)} \right) + y(12s + 6) + 3\frac{y^2}{x^4} - 3 = 0$$

by $\frac{1}{3}x^4$, we have

$$x^6 \left( 1 + 2 \sum_{j=1}^{s} \cos(j\alpha) \right) + x^2 y^2 \left( 1 + 2 \sum_{j=1}^{s} \frac{1}{\cos(j\alpha)} \right) + 2x^4 y(2s+1) + y^2 - x^4 = 0.$$

Let $G(s) = 1 + 2\sum_{j=1}^{s} \cos(j\alpha)$, $H(s) = 1 + 2\sum_{j=1}^{s} \frac{1}{\cos(j\alpha)}$, $M(s) = 1 + 2s$. Then we obtain a parametrized polynomial equation in two variables:

$$x^6 G(s) + x^2 y^2 H(s) + 2x^4 y M(s) + y^2 - x^4 = 0.$$

Now we compute the partial derivative of $H(l, \theta, b)$ with respect to $l_a$:

$$\frac{\partial}{\partial l_a} H(l, \theta, b) = \frac{\partial}{\partial l_a} \sum_{M_{ij}} \frac{1}{l_i l_{i+j} \cos\left(\sum_{k \in M_{ij}} \theta_k\right)} \left( l_i l_{i+j} \cos\left(\sum_{k \in M_{ij}} \theta_k\right) + b_{i+j} \right)^3$$

$$+ \frac{\partial}{\partial l_a} \sum_{M_{ij}} \frac{1}{l_i l_{i+j} \cos\left(\sum_{k \in M_{ij}} \theta_k\right)} \left( l_i l_{i+j} \cos\left(\sum_{k \in M_{ij}} \theta_k\right) + b_i \right)^3$$

$$- 4(1 - l_a^2) l_a + 6 l_a b_a - 4\frac{b_a^3}{l_a^5} - 2\frac{b_a^3}{l_a^3}.$$

From the list of all $M_{ij}$ with $i = a$ and all $M_{ij}$ with $i + j = a$, we have

$$\frac{\partial}{\partial l_a} \sum_{M_{ij}} \frac{1}{l_i l_{i+j} \cos\left(\sum_{k \in M_{ij}} \theta_k\right)} \left( l_i l_{i+j} \cos\left(\sum_{k \in M_{ij}} \theta_k\right) + b_{i+j} \right)^3$$

$$= \sum_{j=1}^{s} \frac{3 \left( l_a l_{a+j} \cos\left(\sum_{k \in M_{aj}} \theta_k\right) + b_{a+j} \right)^2}{l_a} - \frac{\left( l_a l_{a+j} \cos\left(\sum_{k \in M_{aj}} \theta_k\right) + b_{a+j} \right)^3}{l_a^2 l_{a+j} \cos\left(\sum_{k \in M_{aj}} \theta_k\right)}$$

$$+ \sum_{j=1}^{s} \frac{3 \left( l_{a-j} l_a \cos\left(\sum_{k \in M_{(a-j)j}} \theta_k\right) + b_a \right)^2}{l_a} - \frac{\left( l_{a-j} l_a \cos\left(\sum_{k \in M_{(a-j)j}} \theta_k\right) + b_a \right)^3}{l_{a-j} l_a^2 \cos\left(\sum_{k \in M_{(a-j)j}} \theta_k\right)}.$$

So

$$\left. \frac{\partial}{\partial l_a} \right|_{l=l^*, \theta=\theta^*, b=b^*} \sum_{M_{ij}} \frac{1}{l_i l_{i+j} \cos\left(\sum_{k \in M_{ij}} \theta_k\right)} \left( l_i l_{i+j} \cos\left(\sum_{k \in M_{ij}} \theta_k\right) + b_{i+j} \right)^3$$

$$= 2 \sum_{j=1}^{s} \left( \frac{3 \left(x^2 \cos(j\alpha) + y\right)^2}{x} - \frac{\left(x^2 \cos(j\alpha) + y\right)^3}{x^3 \cos(j\alpha)} \right)$$

Similarly,

$$\left. \frac{\partial}{\partial l_a} \right|_{l=l^*, \theta=\theta^*, b=b^*} \sum_{M_{ij}} \frac{1}{l_i l_{i+j} \cos\left(\sum_{k \in M_{ij}} \theta_k\right)} \left( l_i l_{i+j} \cos\left(\sum_{k \in M_{ij}} \theta_k\right) + b_i \right)^3$$

$$= 2 \sum_{j=1}^{s} \left( \frac{3 \left(x^2 \cos(j\alpha) + y\right)^2}{x} - \frac{\left(x^2 \cos(j\alpha) + y\right)^3}{x^3 \cos(j\alpha)} \right)$$

Thus,

$$\frac{\partial}{\partial l_a}\Big|_{l=l^*,\theta=\theta^*,b=b^*} H(l,\theta,b) = x^3\left(4 + 8\sum_{j=1}^{s}\cos^2(j\alpha)\right) + xy\left(6 + 12\sum_{j=1}^{s}\cos(j\alpha)\right)$$

$$- \frac{y^3}{x^3}\left(2 + 4\sum_{j=1}^{s}\frac{1}{\cos(j\alpha)}\right) - 4x - 4\frac{y^3}{x^5}.$$

Multiplying both sides of the equation

$$x^3\left(4 + 8\sum_{j=1}^{s}\cos^2(j\alpha)\right) + xy\left(6 + 12\sum_{j=1}^{s}\cos(j\alpha)\right) - \frac{y^3}{x^3}\left(2 + 4\sum_{j=1}^{s}\frac{1}{\cos(j\alpha)}\right)$$

$$- 4x - 4\frac{y^3}{x^5} = 0$$

by $\frac{1}{2}x^5$, we have

$$2x^8\left(1 + 2\sum_{j=1}^{s}\cos^2(j\alpha)\right) + 3x^6 y\left(1 + 2\sum_{j=1}^{s}\cos(j\alpha)\right) - x^2 y^3\left(1 + 2\sum_{j=1}^{s}\frac{1}{\cos(j\alpha)}\right)$$

$$- 2x^6 - 2y^3 = 0.$$

Let $F(s) = 1 + 2\sum_{j=1}^{s}\cos^2(j\alpha)$. Then we obtain a parametrized polynomial equation in two variables:

$$2x^8 F(s) + 3x^6 y G(s) - x^2 y^3 H(s) - 2x^6 - 2y^3 = 0.$$

Therefore, we need to determine whether the system of two parametrized polynomial equations in two variables

$$x^6 G(s) + x^2 y^2 H(s) + 2x^4 y M(s) + y^2 - x^4 = 0$$

$$2x^8 F(s) + 3x^6 y G(s) - zy^3 H(s) - 2x^6 - 2y^3 = 0$$

where

1. $F(s) = 1 + 2\sum_{j=1}^{s}\cos^2(j\alpha)$;
2. $G(s) = 1 + 2\sum_{j=1}^{s}\cos(j\alpha)$;
3. $H(s) = 1 + 2\sum_{j=1}^{s}\frac{1}{\cos(j\alpha)}$;
4. $M(s) = 1 + 2s$.

have common solutions in $\mathbb{R}_{>0} \times \mathbb{R}_{<0}$ with $-x^2\cos(s\alpha) < y$ or not . Now we compute the partial derivative of $H(l,\theta,b)$ with respect to $\theta_a$.

$$\frac{\partial}{\partial\theta_a} H(l,\theta,b) = \frac{\partial}{\partial\theta_a}\sum_{M_{ij}}\frac{1}{l_i l_{i+j}\cos\left(\sum_{k\in M_{ij}}\theta_k\right)}\left(l_i l_{i+j}\cos\left(\sum_{k\in M_{ij}}\theta_k\right) + b_{i+j}\right)^3$$

$$+ \frac{\partial}{\partial\theta_a}\sum_{M_{ij}}\frac{1}{l_i l_{i+j}\cos\left(\sum_{k\in M_{ij}}\theta_k\right)}\left(l_i l_{i+j}\cos\left(\sum_{k\in M_{ij}}\theta_k\right) + b_i\right)^3.$$

We list all $M_{ij}$ containing $a$:

| $M_{a1}$ | $M_{a2}$ | $\cdots$ | $\cdots$ | $\cdots$ | $M_{as}$ |
|---|---|---|---|---|---|
| $M_{(a-s+1)s}$ | | | | | |
| $M_{(a-s+2)(s-1)}$ | $M_{(a-s+2)s}$ | | | | |
| $\vdots$ | | | | | |
| $M_{(a-2)3}$ | $\cdots$ | $\cdots$ | $M_{(a-2)s}$ | | |
| $M_{(a-1)2}$ | $M_{(a-1)3}$ | $\cdots$ | $\cdots$ | $M_{(a-1)s}$ | |

Rearrange these wedges in terms of the length:

| length 1: | $M_{a1}$ | | | | |
|---|---|---|---|---|---|
| length 2: | $M_{a2}$ | $M_{(a-1)2}$ | | | |
| length 3 | $M_{a3}$ | $M_{(a-1)3}$ | $M_{(a-2)3}$ | | |
| $\vdots$ | | | | | |
| length $s-1$: | $M_{a(s-1)}$ | $M_{(a-1)(s-1)}$ | $\cdots$ | $M_{(a-s+2)(s-1)}$ | |
| length $s$: | $M_{as}$ | $M_{(a-1)s}$ | $\cdots$ | $M_{(a-s+2)s}$ | $M_{(a-s+1)s}$ |

Note that for $j = 1, \ldots, s$, there exactly $j$ wedges of length $j$ containing $a$. Then

$$\frac{\partial}{\partial \theta_a} \sum_{M_{ij}} \frac{1}{l_i l_{i+j} \cos \left( \sum_{k \in M_{ij}} \theta_k \right)} \left( l_i l_{i+j} \cos \left( \sum_{k \in M_{ij}} \theta_k \right) + b_{i+j} \right)^3$$

$$= \sum_{M_{ij}: a \in M_{ij}} \frac{-3 \left[ l_i l_{i+j} \cos \left( \sum_{k \in M_{ij}} \theta_k \right) + b_{i+j} \right]^2 \sin \left( \sum_{k \in M_{ij}} \theta_k \right)}{\cos \left( \sum_{k \in M_{ij}} \theta_k \right)}$$

$$+ \sum_{M_{ij}: a \in M_{ij}} \frac{\sin \left( \sum_{k \in M_{ij}} \theta_k \right) \left[ l_i l_{i+j} \cos \left( \sum_{k \in M_{ij}} \theta_k \right) + b_{i+j} \right]^3}{l_i l_{i+j} \cos^2 \left( \sum_{k \in M_{ij}} \theta_k \right)}.$$

Then

$$\frac{\partial}{\partial \theta_a} \bigg|_{l=l^*, \theta=\theta^*, b=b^*} \sum_{M_{ij}} \frac{1}{l_i l_{i+j} \cos \left( \sum_{k \in M_{ij}} \theta_k \right)} \left( l_i l_{i+j} \cos \left( \sum_{k \in M_{ij}} \theta_k \right) + b_{i+j} \right)^3$$

$$= \sum_{j=1}^{s} j \cdot \left( \frac{-3 \left[ x^2 \cos (j\alpha) + y \right]^2 \sin (j\alpha)}{\cos (j\alpha)} + \frac{\sin (j\alpha) \left[ x^2 \cos (j\alpha) + y \right]^3}{x^2 \cos^2 (j\alpha)} \right).$$

Similarly,

$$\frac{\partial}{\partial \theta_a} \bigg|_{l=l^*, \theta=\theta^*, b=b^*} \sum_{M_{ij}} \frac{1}{l_i l_{i+j} \cos \left( \sum_{k \in M_{ij}} \theta_k \right)} \left( l_i l_{i+j} \cos \left( \sum_{k \in M_{ij}} \theta_k \right) + b_i \right)^3$$

$$= \sum_{j=1}^{s} j \cdot \left( \frac{-3 \left[ x^2 \cos (j\alpha) + y \right]^2 \sin (j\alpha)}{\cos (j\alpha)} + \frac{\sin (j\alpha) \left[ x^2 \cos (j\alpha) + y \right]^3}{x^2 \cos^2 (j\alpha)} \right).$$

Thus,

$$\frac{\partial}{\partial \theta_a} \bigg|_{l=l^*, \theta=\theta^*, b=b^*} H(l, \theta, b) = 2 \sum_{j=1}^{s} j \cdot \left( \frac{-3 \left[ x^2 \cos (j\alpha) + y \right]^2 \sin (j\alpha)}{\cos (j\alpha)} \right.$$

$$\left. + \frac{\sin (j\alpha) \left[ x^2 \cos (j\alpha) + y \right]^3}{x^2 \cos^2 (j\alpha)} \right),$$

which is independent of $a$. So if the system of polynomial equations

$$x^6 G(s) + x^2 y^2 H(s) + 2x^4 y M(s) + y^2 - x^4 = 0$$

$$2x^8 F(s) + 3x^6 y G(s) - x^2 y^3 H(s) - 2x^6 - 2y^3 = 0$$

has a common solution $(x^*, y^*) \in \mathbb{R}_{>0} \times \mathbb{R}_{<0}$ with $-x^2 \cos(s\alpha) < y$, then the Lagrangian multiplier is

$$2 \sum_{j=1}^{s} j \cdot \left( \frac{-3 \left[ (x^*)^2 \cos (j\alpha) + y^* \right]^2 \sin (j\alpha)}{\cos (j\alpha)} + \frac{\sin (j\alpha) \left[ (x^*)^2 \cos (j\alpha) + y^* \right]^3}{(x^*)^2 \cos^2 (j\alpha)} \right)$$

and the Lagrangian multipler method implies that the $c$-gon with coordinate

$$l_1 = \cdots = l_c = x^*, \quad b_1 = \cdots = b_c = y^*, \quad \theta_1 = \cdots = \theta_c = \alpha = \frac{2\pi}{c},$$

is a critical point of $H(l, \theta, b)$ with constraint $\theta_1 + \cdots + \theta_c = 2\pi$. So it remains to determine whether the system of two parametrized polynomial equations in two variables

$$x^6 G(s) + x^2 y^2 H(s) + 2x^4 y M(s) + y^2 - x^4 = 0$$
$$2x^8 F(s) + 3x^6 y G(s) - x^2 y^3 H(s) - 2x^6 - 2y^3 = 0$$

have common solutions $(x^*, y^*) \in \mathbb{R}_{>0} \times \mathbb{R}_{\leq 0}$ with $-x^2 \cos(s\alpha) < y$ or not. We compute the common solution using Mathematica.

1. $c = 5$: in this case, $s = 1$. Then $(x^*, y^*) \approx (1.17046, -0.28230)$ is the unique common solution such that $-x^2 \cos(2\pi/5) < y$. Moreover, the Hessian of $H$ at this 5-gon is non-degenerate. So the 5-gon with coordinate:

$$l_1 = l_2 = \cdots = l_5 = x \approx 1.17046;$$
$$\theta_1 = \theta_2 = \cdots = \theta_4 = \frac{2\pi}{5};$$
$$b_1 = b_2 = \cdots = b_5 = y \approx -0.28230,$$

   is a non-degenerate critical point. So its local learning coefficient is $(3c - 1)/2 = 7$.

2. $c = 6$: in this case $s = 1$. Then $(x^*, y^*) \approx (1.32053, -0.61814)$ is the unique common solution such that $-x^2 \cos(\pi/3) < y$. Moreover, the Hessian of $H$ at this 6-gon is non-degenerate. So the 6-gon with coordinate:

$$l_1 = l_2 = \cdots = l_6 \approx 1.32053;$$
$$\theta_1 = \theta_2 = \cdots = \theta_5 = \frac{\pi}{3};$$
$$b_1 = b_2 = \cdots = b_6 \approx -0.61814,$$

   is a non-degenerate critical point. So its learning coefficient is $(3c - 1)/2 = 8.5$.

3. $c = 7$: in this case $s = 1$. Then $(x^*, y^*) \approx (1.44839, -0.96691)$ is the unique common solution such that $-x^2 \cos(2\pi/7) < y$. Moreover, the Hessian of $H$ at this 7-gon is non-degenerate. So the 5-gon with coordinate:

$$l_1 = l_2 = \cdots = l_7 \approx 1.44839;$$
$$\theta_1 = \theta_2 = \cdots = \theta_6 = \frac{2\pi}{7};$$
$$b_1 = b_2 = \cdots = b_7 \approx -0.96691,$$

   is a non-degenerate critical point. So itslocal learning coefficient is $(3c - 1)/2 = 10$.

4. $c = 9$: in this case $s = 2$. There is no common solution.

5. $c = 10$: in this case $s = 2$. There is no common solution.

6. $c = 11$: in this case $s = 2$. There is no common solution.

7. $c = 13$: in this case $s = 3$. There is no common solution.

8. We checked that for any $9 \leq c \leq 203$ which is not a multiple of 4, there is no common solution.

## H.2  $k$-GONS WITH NEGATIVE BIAS FOR $k = c \in 4\mathbb{Z}$

In this section, we show that for $c = 8$, the 8-gon with coordinate

$$l_1 = l_2 = \cdots = l_8 \approx 1.55041, \quad \theta_1 = \theta_2 = \cdots = \theta_8 = \frac{\pi}{4}, \quad b_1 = b_2 = \cdots = b_8 \approx -1.29122,$$

is a non-degenerate critical point of the TMS potential. So the local learning coefficient of 8-gon is 11.5. Let $c > 4$ being a multiple of 4. A $c$-gon has $\theta$-coordinate:

$$\theta_1 = \theta_2 = \cdots = \theta_c = \frac{2\pi}{c}.$$

Let $s = c/4 - 1 \in \mathbb{Z}$. Then $(s+1) \cdot (2\pi/c) = \pi/2$. So

$$W_i \cdot W_{i+s} = l_i l_{i+s} \cos\left((s+1) \cdot \frac{2\pi}{c}\right) = l_i l_{i+s} \cos\left((s+1) \cdot \frac{\pi}{2}\right) = 0$$

for all $i = 1, \ldots, c$. So $c$-gons are on the boundaries of some chambers.

Since $c > 4$, $s \geq 1$. Let $\mathcal{M} = \{M_{ij}\}$ be wedges describing a chamber whose boundary contains $c$-gons. If $b_i < 0$ for all $i = 1, \ldots, c$, then the TMS potential (see Appendix G) is

$$H(l, \theta, b) = \sum_{M_{ij} \in \mathcal{M}} \delta(T_{M_{ij}}^{(1)}) \frac{1}{l_i l_{i+j} \cos\left(\sum_{k \in M_{ij}} \theta_k\right)} \left(l_i l_{i+j} \cos\left(\sum_{k \in M_{ij}} \theta_k\right) + b_{i+j}\right)^3$$

$$+ \sum_{M_{ij} \in \mathcal{M}} \delta(T_{M_{ij}}^{(2)}) \frac{1}{l_i l_{i+j} \cos\left(\sum_{k \in M_{ij}} \theta_k\right)} \left(l_i l_{i+j} \cos\left(\sum_{k \in M_{ij}} \theta_k\right) + b_i\right)^3$$

$$+ \sum_{i=1}^{c} \delta(T_i)\left[(1 - l_i^2)^2 - 3(1 - l_i^2)b_i + 3b_i^2 + \frac{b_i^3}{l_i^4} + \frac{b_i^3}{l_i^2}\right] + (1 - \delta(T_i)),$$

where

1. $T_{M_{ij}}^{(1)} = \left\{(l, \theta, b) \mid -l_i l_{i+j} \cos\left(\sum_{k \in M_{ij}} \theta_k\right) \leq b_{i+j}\right\}$;

2. $T_{M_{ij}}^{(2)} = \left\{(l, \theta, b) \mid -l_i l_{i+j} \cos\left(\sum_{k \in M_{ij}} \theta_k\right) \leq b_i\right\}$;

3. $T_i = \{(l, \theta, b) \mid -l_i^2 \leq b_i\}$;

4. $\theta_1 + \theta_2 + \cdots + \theta_c = 2\pi$.

Consider the $c$-gon with coordinate

$$l^*: l_1 = \cdots = l_c = x, \quad \theta^*: \theta_1 = \cdots = \theta_c = \alpha = \frac{2\pi}{c}, \quad b^*: b_1 = \cdots = b_c = y$$

for some $x > 0$ and $y < 0$. Suppose that $-x^2 < -x^2 \cos(\alpha) < \cdots < -x^2 \cos(s \cdot \alpha) < y$.

**Lemma H.1.** *The $c$-gon $(l^*, \theta^*, b^*)$ has an open neighbourhood in which the TMS potential is*

$$H(l, \theta, b) = \sum_{M_{ij}} \frac{1}{l_i l_{i+j} \cos\left(\sum_{k \in M_{ij}} \theta_k\right)} \left(l_i l_{i+j} \cos\left(\sum_{k \in M_{ij}} \theta_k\right) + b_{i+j}\right)^3$$

$$+ \sum_{M_{ij}} \frac{1}{l_i l_{i+j} \cos\left(\sum_{k \in M_{ij}} \theta_k\right)} \left(l_i l_{i+j} \cos\left(\sum_{k \in M_{ij}} \theta_k\right) + b_i\right)^3$$

$$+ \sum_{i=1}^{c}\left[(1 - l_i^2)^2 - 3(1 - l_i^2)b_i + 3b_i^2 + \frac{b_i^3}{l_i^4} + \frac{b_i^3}{l_i^2}\right],$$

*where $\theta_1 + \theta_2 + \cdots + \theta_c = 2\pi$, and $\mathcal{M} = \{M_{ij}\}$ is*

| $(1)$ | $(1,2)$ | $\cdots$ | $\cdots$ | $(1, 2, \ldots, s)$ |
|---|---|---|---|---|
| $(2)$ | $(2,3)$ | $\cdots$ | $\cdots$ | $(2, 3, \ldots, s+1)$ |
| $\vdots$ | $\vdots$ | $\cdots$ | $\cdots$ | $\vdots$ |
| $(c+1-s)$ | $(c+1-s, c+2-s)$ | $\cdots$ | $\cdots$ | $(c+1-s, c+2-s, \ldots, c)$ |
| $(c+2-s)$ | $(c+2-s, c+3-s)$ | $\cdots$ | $\cdots$ | $(c+2-s, c+3-s, \ldots, c, 1)$ |
| $\vdots$ | $\vdots$ | $\cdots$ | $\cdots$ | $\vdots$ |
| $(c-1)$ | $(c-1, c)$ | $\cdots$ | $\cdots$ | $(c-1, c, 1, \ldots, s-2)$ |
| $(c)$ | $(c, 1)$ | $\cdots$ | $\cdots$ | $(c, 1, 2, \ldots, s-1)$ |

*Proof.* Since $x > 0$ and $y < 0$, there are positive small real numbers $\epsilon_x$, $\epsilon_y$ such that $x - \epsilon_x > 0$ and $y + \epsilon_b < 0$. Then

$$\frac{y + \epsilon_b}{-(x + \epsilon_x)^2}$$

is a positive number. Then there exists $\gamma > 0$ such that

$$\cos\left(\frac{\pi}{2} - \gamma\right) < \frac{y + \epsilon_b}{-(x + \epsilon_x)^2}.$$

Let $\epsilon_\theta = \gamma/(s + 1)$. Consider the open neighbourhood of $(l^*, \theta^*, b^*)$ given by

$$(x - \epsilon_x, x + \epsilon_x) \times (\alpha - \epsilon_\theta, \alpha + \epsilon_\theta) \times (y - \epsilon_b, y + \epsilon_b).$$

Let $(l, \theta, b)$ be a point in this open neighbourhood. Consider for any wedge $M_{ij}$ containing more than $s$ numbers. Without loss of generality, assume that $M_{ij}$ contain $s+1$ elements. If $\sum_{k \in M_{ij}} \theta_k > \pi/2$, then

$$-l_i l_{i+j} \cos\left(\sum_{k \in M_{ij}} \theta_k\right) > 0 > y + \epsilon_b.$$

Otherwise, suppose $\sum_{k \in M_{ij}} \theta_k \leq \pi/2$. Then

$$\begin{aligned}
-l_i l_{i+j} \cos\left(\sum_{k \in M_{ij}} \theta_k\right) &> -(x + \epsilon_x)^2 \cos\left(\sum_{k \in M_{ij}} \theta_k\right) \\
&> -(x + \epsilon_x)^2 \cos\left((s+1) \times \left(\frac{2\pi}{n} - \epsilon_\theta\right)\right) \\
&= -(x + \epsilon_x)^2 \cos\left(\frac{\pi}{2} - \gamma\right) \\
&> y + \epsilon_b \\
&> b_i \text{ or } b_{i+j}.
\end{aligned}$$

Thus $(l, \theta, b) \notin T_{M_{ij}}^{(1)}$ and $(l, \theta, b) \notin T_{M_{ij}}^{(2)}$. Thus, the term in $H(l, \theta, b)$ indexed by this $M_{ij}$ disappears. So only $M_{ij}$ containing less than $(s + 1)$ numbers remain in the sum. $\qquad\square$

It follows from the calculations in (Appendix H.1) that

$$\frac{\partial}{\partial \theta_a}\bigg|_{l=l^*, \theta=\theta^*, b=b^*} H(l, \theta, b)$$

is independent of $a$. Moreover, the same calculations show that

$$\frac{\partial}{\partial b_a}\bigg|_{l=l^*, \theta=\theta^*, b=b^*} H(l, \theta, b) = 0$$

and

$$\frac{\partial}{\partial l_a}\bigg|_{l=l^*, \theta=\theta^*, b=b^*} H(l, \theta, b) = 0$$

give rise to a system of parametrized polynomial equations in $x$ and $y$:

$$x^6 G(s) + x^2 y^2 H(s) + 2x^4 y M(s) + y^2 - x^4 = 0$$
$$2x^8 F(s) + 3x^6 y G(s) - x^2 y^3 H(s) - 2x^6 - 2y^3 = 0$$

where

1. $F(s) = 1 + 2\sum_{j=1}^{s} \cos^2(j\alpha)$;
2. $G(s) = 1 + 2\sum_{j=1}^{s} \cos(j\alpha)$;
3. $H(s) = 1 + 2\sum_{j=1}^{s} \frac{1}{\cos(j\alpha)}$;

4. $M(s) = 1 + 2s$.

We compute the common solution using Mathematica.

1. $c = 8$: in this case, $s = 1$. Then $(x^*, y^*) \approx (1.55045, -1.29119)$ is a common solution such that $-x^2 \cos(\pi/4) < y$. Moreover, the Hessian of $H(l, \theta, b)$ at this 8-gon is non-degenerate. So the 8-gon with coordinate:

$$l_1 = l_2 = \cdots = l_8 \approx 1.55045;$$
$$\theta_1 = \theta_2 = \cdots = \theta_8 = \frac{\pi}{4};$$
$$b_1 = b_2 = \cdots = b_8 \approx -1.29119,$$

is a non-degenerate critical point. So its local learning coefficient is $(3c - 1)/2 = 11.5$

2. $n = 12$: in this case, $s = 2$. Then $(x^*, y^*) \approx (1.03322, -0.46654)$ and $(x^*, y^*) \approx (1.24975, -0.85483)$ are common solutions such that $-x^2 \cos(\pi/6) < y$.

3. We plot the level sets of two polynomial equations for $1 \le s \le 50$. There is always a common solution.

## H.3 $k$-GONS WITH NEGATIVE BIAS FOR $k < c$

Now for a fixed $c$, we discuss arbitrary $k$-gons where $k \le c$. A $k$-gon is obtained by shrinking $c - k$ $W_i$'s to zero. Without loss of generality, we assume that $W_c, W_{c-1}, \ldots, W_{k+1}$ are shrinking to zero. So a $k$-gon is on the boundary of some chamber. Note that different angles between $W_i$ and $W_j$ for $i, j \in \{c - k + 1, \ldots, c\}$ might give different chambers whose boundary contains the $k$-gon. The following example illustrates the idea. Let $c = 6$ and $k = 5$. Then there are three different chambers whose boundary contains the 5-gon.

1. Consider a family of 6-gons with coordinate $l_1 = l_2 = l_3 = l_4 = l_5 = l$, $l_6 = u$, $\theta_1 = \theta_2 = \theta_3 = \theta_4 = \frac{2\pi}{5}$, $\theta_5 = \alpha$ where $l, u > 0$ and $\alpha \in \left[0, \frac{\pi}{10}\right)$. This family of 6-gons is contained in the chamber described by the following wedges:

| (1) | |
|-----|-----|
| (2) | |
| (3) | |
| (4) | $(4, 5)$ |
| (5) | $(5, 6)$ |
| (6) | |

The 5-gon is obtained from this family of 6-gons by taking the limit as $u \to 0$. So the 5-gon is on the boundary of this chamber.

2. Consider another family of 6-gons with coordinate $l_1 = l_2 = l_3 = l_4 = l_5 = l$, $l_6 = u$, $\theta_1 = \theta_2 = \theta_3 = \theta_4 = \frac{2\pi}{5}$, $\theta_5 = \alpha$ where $l, u > 0$ and $\alpha \in \left[\frac{\pi}{10}, \frac{3\pi}{10}\right]$. This family of 6-gons is contained in the chamber described by the following wedges:

| (1) | |
|-----|-----|
| (2) | |
| (3) | |
| (4) | |
| (5) | $(5, 6)$ |
| (6) | |

The 5-gon is obtained from this family of 6-gons by taking the limit as $u \to 0$. So the 5-gon is on the boundary of this chamber.

3. Finally, consider the family of 6-gons with coordinate $l_1 = l_2 = l_3 = l_4 = l_5 = l$, $l_6 = u$, $\theta_1 = \theta_2 = \theta_3 = \theta_4 = \frac{2\pi}{5}$, $\theta_5 = \alpha$ where $l, u > 0$ and $\alpha \in \left(\frac{3\pi}{10}, \frac{2\pi}{5}\right]$. This family of 6-gons is contained in the chamber described by the following wedges:

| | |
|---|---|
| (1) | |
| (2) | |
| (3) | |
| (4) | |
| (5) | (5, 6) |
| (6) | (6, 1) |

The 5-gon is obtained from this family of 6-gons by taking the limit as $u \to 0$. So the 5-gon is on the boundary of this chamber.

**Theorem H.1.** *Let $k \in \mathbb{Z}_{>4}$ and $s$ be the unique integer in the interval $[\frac{k}{4} - 1, \frac{k}{4})$. If a $k$-gon with coordinate*

$$l_1 = \cdots = l_k = x; \quad \theta_1 = \cdots = \theta_k = \frac{2\pi}{k}; \quad b_1 = \cdots = b_k = y$$

*for some $x > 0$ and $y < 0$ satisfying*

$$-x^2 \cos\left(s \cdot \frac{2\pi}{k}\right) \le y$$

*is a critical point of $H(l, \theta, b)$ with constraint $\theta_1 + \cdots + \theta_k = 2\pi$ for $c = k$, then the $k$-gons with coordinate*

$$l_1 = \cdots = l_k = x, \quad l_{k+1} = \cdots = l_c = 0;$$

$$\theta_1 = \cdots = \theta_{k-1} = \frac{2\pi}{k}, \quad \theta_k + \cdots + \theta_c = \frac{2\pi}{k};$$

$$b_1 = \cdots = b_k = y, \quad b_{k+1}, \ldots, b_c < 0,$$

*are critical points of $H(l, \theta, b)$ with constraint $\theta_1 + \cdots + \theta_c = 2\pi$ for any $c > k$.*

*Proof.* We first show that for any $b_{k+1}, \ldots, b_c < 0$ and $\theta_k, \ldots, \theta_c \in [0, 2\pi)$ with $\theta_k + \cdots + \theta_c = 2\pi/k$, the $k$-gon $(l^*, \theta^*, b^*)$ with coordinate

$$l_1 = \cdots = l_k = x, \quad l_{k+1} = \cdots = l_c = 0;$$

$$\theta_1 = \cdots = \theta_{k-1} = \alpha = \frac{2\pi}{k}, \quad \theta_k, \ldots, \theta_c \in [0, 2\pi);$$

$$b_1 = \cdots = b_k = y, \quad b_{k+1}, \ldots, b_c < 0$$

has an open neighbourhood in which only the following types of wedges showing up in $H(l, \theta, b)$:

1. $M_{ij}$ does not contain any of $\{k, k+1, \ldots, c\}$ and has length at most $s$;

2. $M_{ij}$ contains $(k, k+1, \ldots, c)$ and has length at most $s + c - k$.

Since each coordinate in $b^*$ is less than zero, there is an open neighbourhood $B$ of $b^*$ contained in $\mathbb{R}_{<0}^c$. Since $l_1 = \cdots = l_k = x > 0$ and $l_{k+1} = \cdots = l_c = 0$, there exists an open neighbourhood $L$ of $l^*$ contained in $\mathbb{R}_{>0}^k \times \mathbb{R}_{\ge 0}^{c-k}$. If $k$ is not a multiple of 4, then $s \cdot \alpha < \pi/2$ and $(s+1) \cdot \alpha > \pi/2$. So we can perturb each $\theta_i$ to obtain an open neighbourhood $\Theta$ of $\theta^*$ in which

- $\theta_i + \cdots + \theta_{i+s-1} < \pi/2$ and $\theta_i + \cdots \theta_{i+s} > \pi/2$ for $\{i, i+1, \ldots, i+s\} \subset \{1, \ldots, k-1\}$;

- $\theta_i + \cdots + \theta_{k-1} + \theta_k + \cdots + \theta_c + \theta_1 + \cdots + \theta_{s-1-k+i} < \pi/2, \theta_i + \cdots + \theta_{k-1} + \theta_k + \cdots + \theta_c + \theta_1 + \cdots + \theta_{s-k+i} > \pi/2$, and $\theta_{i-1} + \theta_i + \cdots + \theta_{k-1} + \theta_k + \cdots + \theta_c + \theta_1 + \cdots + \theta_{s-1-k+i} < \pi/2$.

If $k$ is a multiple of 4, then $s \cdot \alpha < \pi/2$ and $(s+1) \cdot \alpha = \pi/2$. So we can perturb each $\theta_i$ to obtain an open neighbourhood $\Theta$ of $\theta^*$ in which

- $\theta_i + \cdots + \theta_{i+s-1} < \pi/2$ and $\theta_i + \cdots \theta_{i+s}$ is closed to $\pi/2$ for $\{i, i+1, \ldots, i+s\} \subset \{1, \ldots, k-1\}$;

- $\theta_i + \cdots + \theta_{k-1} + \theta_k + \cdots + \theta_c + \theta_1 + \cdots + \theta_{s-1-k+i} < \pi/2, \theta_i + \cdots + \theta_{k-1} + \theta_k + \cdots + \theta_c + \theta_1 + \cdots + \theta_{s-k+i}$ is closed to $\pi/2$, and $\theta_{i-1} + \theta_i + \cdots + \theta_{k-1} + \theta_k + \cdots + \theta_c + \theta_1 + \cdots + \theta_{s-1-k+i} < \pi/2$.

Then $L \times \Theta \times B$ is an open neighbourhood of $(l^*, \theta^*, b^*)$. Since $l_{k+1} = \cdots = l_c = 0$, we can shrink $L$ so that for every $(l, \theta, b) \in L \times \Theta \times B$,

$$-l_i l_{i+j} \cos\left(\sum_{k \in M_{ij}} \theta_k\right) > b_i, \quad -l_i l_{i+j} \cos\left(\sum_{k \in M_{ij}} \theta_k\right) > b_{i+j}$$

for all $M_{ij}$ with either $i \in \{k+1, \ldots, c\}$ or $i + j - 1 \in \{k, \ldots, c-1\}$, and

$$-l_i^2 > b_i$$

for all $i = k+1, \ldots, c$. Since

$$-x^2 < \cdots < -x^2 \cos\left((s-1) \cdot \frac{2\pi}{k}\right) < -x^2 \cos\left(s \cdot \frac{2\pi}{k}\right) < y,$$

we can shrink $L \times \Theta \times B$ so that for every $(l, \theta, b) \in L \times \Theta \times B$,

$$-l_i l_{i+j} \cos\left(\sum_{k \in M_{ij}} \theta_k\right) < b_i, \quad -l_i l_{i+j} \cos\left(\sum_{k \in M_{ij}} \theta_k\right) < b_{i+j}$$

for all $i, j$ with $i, i+j \in \{1, \ldots, k\}$ and

$$-l_i^2 < b_i$$

for all $i = 1, \ldots, k$. Since any $M_{ij}$ containing only some of $\{k, k+1, \ldots, c\}$ has either $i \in \{k+1, \ldots, c\}$ or $i + j - 1 \in \{k, \ldots, c-1\}$, we know that for $M_{ij}$ containing only some of $\{k, k+1, \ldots, c\}$, we have

$$(l, \theta, b) \notin T_{M_{ij}}^{(1)} \text{ and } (l, \theta, b) \notin T_{M_{ij}}^{(2)}$$

for all $(l, \theta, b) \in L \times \Theta \times B$. We have shown that if $M_{ij}$ contains some of $\{k, k+1, \ldots, c\}$, then it must contain $(k, k+1, \ldots, c)$. Moreover, for $k$ not being a multiple of $4$, since

- $\theta_i + \cdots + \theta_{k-1} + \theta_k + \cdots + \theta_c + \theta_1 + \cdots + \theta_{s-1-k+i} < \pi/2$, $\theta_i + \cdots + \theta_{k-1} + \theta_k + \cdots + \theta_c + \theta_1 + \cdots + \theta_{s-k+i} > \pi/2$, and $\theta_{i-1} + \theta_i + \cdots + \theta_{k-1} + \theta_k + \cdots + \theta_c + \theta_1 + \cdots + \theta_{s-1-k+i} < \pi/2$,

we know that if $M_{ij}$ contains $(k, k+1, \ldots, c)$, then it has length at most $s + c - k$. For $k$ being a multiple of $4$, since

- $\theta_i + \cdots + \theta_{k-1} + \theta_k + \cdots + \theta_c + \theta_1 + \cdots + \theta_{s-1-k+i} < \pi/2$, $\theta_i + \cdots + \theta_{k-1} + \theta_k + \cdots + \theta_c + \theta_1 + \cdots + \theta_{s-k+i}$ is closed to $\pi/2$, and $\theta_{i-1} + \theta_i + \cdots + \theta_{k-1} + \theta_k + \cdots + \theta_c + \theta_1 + \cdots + \theta_{s-1-k+i} < \pi/2$,

by the same argument we use in the proof of Lemma H.1, we know that if $M_{ij}$ contains $(k, k+1, \ldots, c)$, then it has length at most $s + c - k$. Now for $M_{ij}$ not containing any of $\{k, k+1, \ldots, c\}$, if $k$ is not a multiple of $4$, it follows from

- $\theta_i + \cdots + \theta_{i+s-1} < \pi/2$ and $\theta_i + \cdots \theta_{i+s} > \pi/2$ for $\{i, i+1, \ldots, i+s\} \subset \{1, \ldots, k-1\}$

that $M_{ij}$ has length at most $s$. If $k$ is a multiple of $4$, it follows from

- $\theta_i + \cdots + \theta_{i+s-1} < \pi/2$ and $\theta_i + \cdots \theta_{i+s}$ is closed to $\pi/2$ for $\{i, i+1, \ldots, i+s\} \subset \{1, \ldots, k-1\}$

and the same argument in Lemma H.1 that $M_{ij}$ has length at most $s$. Therefore in the open neighbourhood $L \times \Theta \times B$ of $(l^*, \theta^*, b^*)$, only the following types of wedges showing up in $H$:

1. $M_{ij}$ does not contain any of $\{k, k+1, \ldots, c\}$ and has length at most $s$;

2. $M_{ij}$ contains $(k, k+1, \ldots, c)$ and has length at most $s + c - k$.

For $a = k+1, \ldots, c$, since $(L \times \Theta \times B) \cap T_a = \emptyset$, and $(L \times \Theta \times B) \cap T_{M_{ij}}^{(1)} = (L \times \Theta \times B) \cap T_{M_{ij}}^{(2)} = \emptyset$ for $M_{ij}$ with either $i \in \{k+1, \ldots, c\}$ or $i + j - 1 \in \{k, \ldots, c-1\}$, we know that

$$\frac{\partial}{\partial b_a}\bigg|_{l=l^*, \theta=\theta^*, b=b^*} H(l, \theta, b) = 0$$

and

$$\frac{\partial}{\partial l_a}\bigg|_{l=l^*, \theta=\theta^*, b=b^*} H(l, \theta, b) = 0.$$

For $a = 1, \ldots, k$, since

$$l_1 = \cdots = l_k = x; \quad \theta_1 = \cdots = \theta_k = \frac{2\pi}{k}; \quad b_1 = \cdots = b_k = y$$

is a critical point of the TMS potential for $c = k$, we know that

$$\frac{\partial}{\partial b_a}\bigg|_{l=l^*, \theta=\theta^*, b=b^*} H(l, \theta, b) = 0$$

and

$$\frac{\partial}{\partial l_a}\bigg|_{l=l^*, \theta=\theta^*, b=b^*} H(l, \theta, b) = 0.$$

For $a = 1, \ldots, c$, we have

$$\frac{\partial}{\partial \theta_a}\bigg|_{l=l^*, \theta=\theta^*, b=b^*} H(l, \theta, b)$$

$$= 2 \sum_{j=1}^{s} j \cdot \left( \frac{-3 \left[ x^2 \cos(j\alpha) + y \right]^2 \sin(j\alpha)}{\cos(j\alpha)} + \frac{\sin(j\alpha) \left[ x^2 \cos(j\alpha) + y \right]^3}{x^2 \cos^2(j\alpha)} \right)$$

which is independent of $a$. Therefore, by Lagrangian multiplier method, the $k$-gon $(l^*, \theta^*, b^*)$ is a critical point of $H(l, \theta, b)$ with constraint $\theta_1 + \cdots + \theta_c = 2\pi$. □

In the previous example, we see that for $n = 6$, there are three different chambers whose boundary contains the 5-gon. As different chambers give different explicit form of $H$, one might think $H$ is not differentiable at the 5-gon. However, in our proof, we show that the 5-gon has an open neighbourhood in which $H$ has the explicit form

$$\sum_{M_{ij} \in \mathcal{M}} \frac{1}{l_i l_{i+j} \cos\left(\sum_{k \in M_{ij}} \theta_k\right)} \left( l_i l_{i+j} \cos\left(\sum_{k \in M_{ij}} \theta_k\right) + b_{i+j} \right)^3$$

$$+ \sum_{M_{ij} \in \mathcal{M}} \frac{1}{l_i l_{i+j} \cos\left(\sum_{k \in M_{ij}} \theta_k\right)} \left( l_i l_{i+j} \cos\left(\sum_{k \in M_{ij}} \theta_k\right) + b_i \right)^3$$

$$+ \sum_{i=1}^{5} \left[ (1 - l_i^2)^2 - 3(1 - l_i^2)b_i + 3b_i^2 + \frac{b_i^3}{l_i^4} + \frac{b_i^3}{l_i^2} \right] + 1,$$

where $\theta_1 + \cdots + \theta_6 = 2\pi$ and $\mathcal{M}$ consists of the following wedges:

| (1) |
|-----|
| (2) |
| (3) |
| (4) |
| (5, 6) |

.

So $H$ is actually analytic at the 5-gon.

**Corollary H.1.** *Let $k \in \mathbb{Z}_{>4}$ and $s$ be the unique integer in the interval $[\frac{k}{4} - 1, \frac{k}{4})$. Let $H^{(k)}(l, \theta, b)$ denote the TMS potential for $c = k$. Suppose that a $k$-gon with coordinate*

$$l_1 = \cdots = l_k = x; \quad \theta_1 = \cdots = \theta_k = \frac{2\pi}{k}; \quad b_1 = \cdots = b_k = y$$

*for some $x > 0$ and $y < 0$ satisfying*

$$-x^2 \cos\left(s \cdot \frac{2\pi}{k}\right) \le y$$

*is a critical point of $H^{(k)}(l, \theta, b)$. Let $\Pi_{c,k}$ be the projection defined by*

$$\Pi_{c,k} \colon (l_1, \ldots, l_c, \theta_1, \ldots, \theta_{c-1}, b_1, \ldots, b_c) \mapsto (l_1, \ldots, l_k, \theta_1, \ldots, \theta_{k-1}, b_1, \ldots, b_k).$$

*Then for $c > k$, the $k$-gon with coordinate*

$$l_1 = \cdots = l_k = x, \quad l_{k+1} = \cdots = l_c = 0;$$
$$\theta_1 = \cdots = \theta_{k-1} = \frac{2\pi}{k}, \quad \theta_k + \cdots + \theta_c = \frac{2\pi}{k};$$
$$b_1 = \cdots = b_k = y, \quad b_{k+1}, \ldots, b_c < 0,$$

*has an open neighbourhood in which*

$$H(l, \theta, b) = H^{(k)} \circ \Pi_{c,k}(l, \theta, b) + (c - k).$$

*Proof.* Let $\tau = (k, k+1, \ldots, c)$. From the proof of the theorem, we know that for $c > k$, the $k$-gon

$$l_1 = \cdots = l_k = x, \quad l_{k+1} = \cdots = l_c = 0;$$
$$\theta_1 = \cdots = \theta_{k-1} = \frac{2\pi}{k}, \quad \theta_k + \cdots + \theta_c = \frac{2\pi}{k};$$
$$b_1 = \cdots = b_k = y, \quad b_{k+1}, \ldots, b_c < 0,$$

has an open neighbourhood in which

$$
H(l, \theta, b) = \sum_{M_{ij} \in \mathcal{M}} \frac{1}{l_i l_{i+j} \cos\left(\sum_{k \in M_{ij}} \theta_k\right)} \left( l_i l_{i+j} \cos\left(\sum_{k \in M_{ij}} \theta_k\right) + b_{i+j} \right)^3
$$
$$
+ \sum_{M_{ij} \in \mathcal{M}} \frac{1}{l_i l_{i+j} \cos\left(\sum_{k \in M_{ij}} \theta_k\right)} \left( l_i l_{i+j} \cos\left(\sum_{k \in M_{ij}} \theta_k\right) + b_i \right)^3
$$
$$
+ \sum_{i=1}^{k} \left[ (1 - l_i^2)^2 - 3(1 - l_i^2)b_i + 3b_i^2 + \frac{b_i^3}{l_i^4} + \frac{b_i^3}{l_i^2} \right] + (c - k),
$$

where $\theta_1 + \cdots + \theta_c = 2\pi$ and $\mathcal{M}$ is

| $(1)$ | $(1,2)$ | $\cdots$ | $\cdots$ | $(1, 2, \ldots, s)$ |
|---|---|---|---|---|
| $(2)$ | $(2,3)$ | $\cdots$ | $\cdots$ | $(2, 3, \ldots, s+1)$ |
| $\vdots$ | $\vdots$ | $\cdots$ | $\cdots$ | $\vdots$ |
| $(k-s)$ | $(k-s, k-s+1)$ | $\cdots$ | $\cdots$ | $(k-s, \ldots, k-1)$ |
| $(k-s+1)$ | $(k-s+1, k-s+2)$ | $\cdots$ | $\cdots$ | $(k-s+1, \ldots, k-1, \tau)$ |
| $(k-s+2)$ | $(k-s+1, k-s+2)$ | $\cdots$ | $\cdots$ | $(k-s+1, \ldots, k-1, \tau, 1)$ |
| $\vdots$ | $\vdots$ | $\cdots$ | $\cdots$ | $\vdots$ |
| $(\tau)$ | $(\tau, 1)$ | $\cdots$ | $\cdots$ | $(\tau, 1, \ldots, s-1)$ |

Let $H^{(k)}(l_1, \ldots, l_k, \theta_1, \ldots, \theta_{k-1}, b_1, \ldots, b_k)$ denote the TMS potential for $c = k$. Consider the projection

$$\Pi_{c,k} \colon (l_1, \ldots, l_c, \theta_1, \ldots, \theta_{c-1}, b_1, \ldots, b_c) \mapsto (l_1, \ldots, l_k, \theta_1, \ldots, \theta_{k-1}, b_1, \ldots, b_k).$$

Since

$$\sum_{i \in \tau} \theta_i = 2\pi - (\theta_1 + \cdots + \theta_{k-1}),$$

we have

$$H(l, \theta, b) = H^{(k)} \circ \Pi_{c,k} + (c - k).$$

$\square$

Since the $c$-gons are non-degenerate (modulo $O(c)$-action) critical points for $c = 5, 6, 7, 8$, the corollary implies that for $c > k$ and $k \in \{5, 6, 7, 8\}$, the $k$-gon is a degenerate critical point but it is minimally singular in the sense of the potential $H$ being locally a sum of squares with all squares having positive coefficients around the $k$-gon. We can compute the local learning coefficient of each $k$-gon for $k = 5, 6, 7, 8$:

| Critical point | Local learning coefficient | $L$ |
|---|---|---|
| 5 | 7 | $(0.23738 + c - 5)/3c$ |
| 6 | 8.5 | $(0.86746 + c - 6)/3c$ |
| 7 | 10 | $(1.74870 + c - 7)/3c$ |
| 8 | 11.5 | $(2.77311 + c - 8)/3c$ |

Table H.2: Local learning coefficients and losses for $k$-gons

## I   INCLUDING POSITIVE BIASES

In this section, we discuss the $k^{\sigma+}$-gon defined in Section 3.2. We show that for $c = 6$, $5^+$-gon is a critical point and its local learning coefficient is 8. We also show that in general $k^{\sigma+}$-gons are critical points. Let $W_C$ be a nonempty chamber. Suppose that $W_C$ contains an open subset of $M_{2,c}(\mathbb{R})$. Let $\mathcal{M}$ be the set of wedges describing $W_C$. Recall that the local TMS potential is given by (17) in Appendix G. Let's first discuss $c = 6$. Consider the 5-gon $(l^*, \theta^*, b^*)$ with coordinate

$$l^* : l_1 = \cdots = l_5 = x \approx 1.17046, \ l_6 = 0;$$

$$\theta^* : \theta_1 = \cdots = \theta_4 = \frac{2\pi}{5}, \ \theta_5 + \theta_6 = \frac{2\pi}{5};$$

$$b^* : b_1 = \cdots = b_5 = y \approx -0.28230, \ b_6 = z \in \mathbb{R}_{>0}.$$

In Appendix H.3, we see that depending on the value of $\theta_5$, there are three different chambers whose boundary containing the 5-gon, but for the negative bias case ($b_i < 0$ for all $i = 1, \ldots, c$), there is an open neighbourhood in which the local TMS potential is smooth (Corollary H.1). Now we claim that this holds for the general case, i.e. there is an open neighbourhood of the 5-gon in which the local TMS potential is smooth. Since $l_6 = 0$ and $b_i < 0$ for all $i = 1, \ldots, 5$, we know that

$$- l_6 l_1 \cos(\theta_6) = 0 > b_1, \ l_6 l_2 \cos(\theta_6 + \theta_1) = 0 > b_2,$$
$$- l_5 l_6 \cos(\theta_5) = 0 > b_5, \ -l_4 l_6 \cos(\theta_4 + \theta_5) = 0 > b_4$$

So there is an open neighbourhood $U$ of the 5-gon in which these inequalities hold. Thus, in $U$, the wedges appearing in the local TMS potential are

$$\begin{array}{|c|}
\hline
(1) \\
\hline
(2) \\
\hline
(3) \\
\hline
(4) \\
\hline
(5,6) \\
\hline
\end{array}.$$

Moreover, since $b_6 > 0$, we have

$$-l_6 l_i \cos(\theta_6 + \cdots + \theta_{i-1}) = 0 < b_i,$$

for all $i = 1, \ldots, 5$. We can shrink $U$ so that these inequalities hold in $U$. Thus, in $U$, $\delta(S_{6j}) = 0$ for all $j = 1, \ldots, 5$. Therefore, the local TMS potential is

$$\sum_{i=1}^{4} \frac{1}{l_i l_{i+1} \cos(\theta_i)} \left( l_i l_{i+1} \cos(\theta_i) + b_i \right)^3 + \frac{1}{l_5 l_1 \cos(\theta_5 + \theta_6)} \left( l_5 l_1 \cos(\theta_5 + \theta_6) + b_5 \right)^3$$

$$+ \sum_{i=1}^{4} \frac{1}{l_i l_{i+1} \cos(\theta_i)} \left( l_i l_{i+1} \cos(\theta_i) + b_{i+1} \right)^3 + \frac{1}{l_5 l_1 \cos(\theta_5 + \theta_6)} \left( l_5 l_1 \cos(\theta_5 + \theta_6) + b_1 \right)^3$$

$$+ \sum_{i=1}^{5} \left[ (1 - l_i^2)^2 - 3(1 - l_i^2) b_i + 3b_i^2 + \frac{b_i^3}{l_i^4} + \frac{b_i^3}{l_i^2} \right]$$

$$+ \sum_{i=1}^{5} \left[ \left( l_6 l_i \cos(\theta_6 + \cdots + \theta_{i-1}) \right)^2 + 3 \left( l_6 l_i \cos(\theta_6 + \cdots + \theta_{i-1}) \right) b_6 + 3b_6^2 \right]$$

$$+ \left[ (1 - l_6^2)^2 - 3(1 - l_6^2) b_6 + 3b_6^2 \right].$$

Let

$$\Phi(l, \theta, b) = \sum_{i=1}^{4} \frac{\left( l_i l_{i+1} \cos(\theta_i) + b_i \right)^3}{l_i l_{i+1} \cos(\theta_i)} + \frac{\left( l_5 l_1 \cos(\theta_5 + \theta_6) + b_5 \right)^3}{l_5 l_1 \cos(\theta_5 + \theta_6)}$$

$$+ \sum_{i=1}^{4} \frac{\left( l_i l_{i+1} \cos(\theta_i) + b_{i+1} \right)^3}{l_i l_{i+1} \cos(\theta_i)} + \frac{\left( l_5 l_1 \cos(\theta_5 + \theta_6) + b_1 \right)^3}{l_5 l_1 \cos(\theta_5 + \theta_6)}$$

$$+ \sum_{i=1}^{5} \left[ (1 - l_i^2)^2 - 3(1 - l_i^2) b_i + 3b_i^2 + \frac{b_i^3}{l_i^4} + \frac{b_i^3}{l_i^2} \right],$$

and

$$\Psi(l, \theta, b) = \sum_{i=1}^{5} \left[ \big(l_6 l_i \cos(\theta_6 + \cdots + \theta_{i-1})\big)^2 + 3\big(l_6 l_i \cos(\theta_6 + \cdots + \theta_{i-1})\big)b_6 + 3b_6^2 \big) \right]$$
$$+ \left[ (1 - l_6^2)^2 - 3(1 - l_6^2)b_6 + 3b_6^2 \right],$$

Then the local TMS potential is

$$H(l, \theta, b) = \Phi(l, \theta, b) + \Psi(l, \theta, b).$$

Note that Corollary (H.1) implies that $\Phi(l, \theta, b) = H^{(5)} \circ \Pi_{6,5}(l, \theta, b)$, where $H^{(5)}$ is the TMS potential for $n = 5$ and $\Pi_{6,5}$ is the projection

$$\Pi_{6,5} \colon (l_1, \ldots, l_6, \theta_1, \ldots, \theta_5, b_1, \ldots, b_6) \mapsto (l_1, \ldots, l_5, \theta_1, \ldots, \theta_4, b_1, \ldots, b_5).$$

Thus, for all $a = 1, \ldots, 6$,

$$\left. \frac{\partial}{\partial l_a} \right|_{l=l^*, \theta=\theta^*, b=b^*} \Phi(l, \theta, b) = 0, \quad \left. \frac{\partial}{\partial b_a} \right|_{l=l^*, \theta=\theta^*, b=b^*} \Phi(l, \theta, b) = 0,$$

and

$$\left. \frac{\partial}{\partial \theta_a} \right|_{l=l^*, \theta=\theta^*, b=b^*} \Phi(l, \theta, b)$$

is independent of $i$. For $a = 1, \ldots, 5$,

$$\frac{\partial}{\partial l_a} \Psi(l, \theta, b) = 2\big(l_6 l_a \cos(\theta_6 + \cdots + \theta_{a-1})\big)l_6 \cos(\theta_6 + \cdots + \theta_{a-1})$$
$$+ 3l_6 \cos(\theta_6 + \cdots + \theta_{a-1})b_6.$$

Since $l_6 = 0$ for the 5-gon, for all $a = 1, \ldots, 5$,

$$\left. \frac{\partial}{\partial l_a} \right|_{l=l^*, \theta=\theta^*, b=b^*} \Psi(l, \theta, b) = 0.$$

Compute

$$\frac{\partial}{\partial l_6} \Psi(l, \theta, b) = \left\{ \sum_{i=1}^{5} 2\big(l_6 l_i \cos(\theta_6 + \cdots + \theta_{i-1})\big)l_i \cos(\theta_6 + \cdots + \theta_{i-1}) \right.$$
$$\left. + 3l_i \cos(\theta_6 + \cdots + \theta_{i-1})b_6 \right\} - 4(1 - l_6^2)l_6 + 6l_6 b_6.$$

Then

$$\left. \frac{\partial}{\partial l_6} \right|_{l=l^*, \theta=\theta^*, b=b^*} \Psi(l, \theta, b) = 3xz \sum_{i=1}^{5} \cos\left( \theta_6 + (i-1) \cdot \frac{2\pi}{5} \right)$$

$$= 3xz \sum_{i=1}^{5} \cos(\theta_6) \cos\left( (i-1) \cdot \frac{2\pi}{5} \right) - \sin(\theta_6) \sin\left( (i-1) \cdot \frac{2\pi}{5} \right)$$

$$= 3xz \left[ \cos(\theta_6) \sum_{i=1}^{5} \cos\left( (i-1) \cdot \frac{2\pi}{5} \right) \right.$$

$$\left. - \sin(\theta_6) \sum_{i=1}^{5} \sin\left( (i-1) \cdot \frac{2\pi}{5} \right) \right]$$

$$= 3xz \left[ \cos(\theta_6) \cdot 0 - \sin(\theta_6) \cdot 0 \right]$$

$$= 0.$$

For $a = 1, \ldots, 5$,

$$\frac{\partial}{\partial b_a} \Psi(l, \theta, b) = 0.$$

Compute

$$\frac{\partial}{\partial b_6} \Psi(l, \theta, b) = \sum_{i=1}^{5} \left[ 3\big(l_6 l_i \cos(\theta_6 + \cdots + \theta_{i-1})\big) + 6b_6 \right] - 3(1 - l_6^2) + 6b_6.$$

Then

$$\frac{\partial}{\partial b_6}\bigg|_{l=l^*, \theta=\theta^*, b=b^*} \Psi(l, \theta, b) = \left( \sum_{i=1}^{5} 6z \right) - 3 + 6z$$
$$= 36z - 3.$$

So

$$\frac{\partial}{\partial b_6}\bigg|_{l=l^*, \theta=\theta^*, b=b^*} \Psi(l, \theta, b) = 0$$

if and only if $z = \frac{1}{12}$. Note that

$$\frac{\partial}{\partial \theta_5} \Psi(l, \theta, b) = 0$$

as $\theta_5$ is not in $\Psi_{l,\theta,b}$. For $a = 1, \ldots, 4$,

$$\frac{\partial}{\partial \theta_a} \Psi(l, \theta, b) = \sum_{i=a+1}^{5} \left[ -2\big(l_6 l_i \cos(\theta_6 + \cdots + \theta_{i-1})\big) l_6 l_i \sin(\theta_6 + \cdots + \theta_{i-1}) \right.$$
$$\left. - 3l_6 l_i \sin(\theta_6 + \cdots + \theta_{i-1}) b_6 \right].$$

Since for the 5-gon, $l_6 = 0$, then for $a = 1, \ldots, 4$,

$$\frac{\partial}{\partial \theta_a}\bigg|_{l=l^*, \theta=\theta^*, b=b^*} \Psi(l, \theta, b) = 0.$$

Compute

$$\frac{\partial}{\partial \theta_6} \Psi(l, \theta, b) = \sum_{i=1}^{5} \left[ -2\big(l_6 l_i \cos(\theta_6 + \cdots + \theta_{i-1})\big) l_6 l_i \sin(\theta_6 + \cdots + \theta_{i-1}) \right.$$
$$\left. - 3l_6 l_i \sin(\theta_6 + \cdots + \theta_{i-1}) b_6 \right].$$

Since for the 5-gon, $l_6 = 0$, then

$$\frac{\partial}{\partial \theta_6}\bigg|_{l=l^*, \theta=\theta^*, b=b^*} \Psi(l, \theta, b) = 0.$$

Therefore, the 5-gon with coordinate

$$l^* : l_1 = \cdots = l_5 = x \approx 1.17046, \ l_6 = 0;$$
$$\theta^* : \theta_1 = \cdots = \theta_4 = \frac{2\pi}{5}, \ \theta_5 + \theta_6 = \frac{2\pi}{5};$$
$$b^* : b_1 = \cdots = b_5 = y \approx -0.28230, \ b_6 = \frac{1}{12};$$

is a critical point of $H(l, \theta, b)$. Note that this is the $5^+$-gon defined in Section 3.2. We claim that the TMS potential is minimally singular in some neighbourhood of $5^+$-gon and compute its local learning coefficient.

**Lemma I.1.** *For $c = 6$, the $5^+$-gon with coordinate $(l^*, \theta^*, b^*)$ has an open neighbourhood in which the TMS potential is minimally singular. Moreover, its local learning coefficient is $8$.*

*Proof.* Note that the Hessian of $H$ at the $5^+$-gon $(l^*, \theta^*, b^*)$ has all positive eigenvalues except one zero eigenvalue in the $\theta_5$-direction. Note that there is an open subset $U$ of the parameter space such that the connected 1-dimensional space:

$$C = \left\{ (l, \theta, b) \mid l = l^*, b = b^*, \theta_1 = \cdots \theta_4 = \frac{2\pi}{5}, 0 \leq \theta_5 \leq \frac{2\pi}{5} \right\}$$

is the set of critical points of $H$ in $U$. Moreover, since for all $(l, \theta, b) \in C$, the Hessian of $H$ at $(l, \theta, b)$ is non-degenerate in the direction normal to $T_{(l,\theta,b)}C$, $H|_U$ is a Morse-Bott function with index 0. It follows from the Morse-Bott lemma that $H$ is minimally singular in $U$ and the local learning coefficient is $(17 - 1)/2 = 8$. $\qquad\square$

Now we discuss the $k^{\sigma+}$-gons (in Section 3.2) for $k < c$. Let $B^+ \subset \{k+1, \ldots, c\}$. Let $\sigma = |B^+|$. Consider the $k$-gon $(l^*, \theta^*, b^*)$ with coordinate

$$l^* : l_1 = \cdots = l_k = x > 0, \ l_{k+1} = \cdots = l_c = 0;$$

$$\theta^* : \theta_1 = \cdots = \theta_{k-1} = \frac{2\pi}{c}, \ \theta_k + \cdots + \theta_c = \frac{2\pi}{c};$$

$$b^* : b_1 = \cdots = b_k = y < 0,$$

$$\text{for } i = k+1, \ldots, c, \ b_i < 0 \text{ if } i \notin B^+ \text{ and } b_i = z > 0 \text{ if } i \in B^+.$$

**Theorem I.1.** *There is an open neighbourhood of $(l^*, \theta^*, b^*)$ in which the TMS potential is*

$$H(l, \theta, b) = H^{(k)} \circ \Pi_{c,k}(l, \theta, b) + \big( c - (k + \sigma) \big) + \sum_{i \in B^+} H_i^+(l, \theta, b),$$

*where*

1. *$H^{(k)}$ is the TMS potential for $c = k$;*

2. *$\Pi_{c,k}$ is the projection:*
   $$\Pi_{c,k} : (l_1, \ldots, l_c, \theta_1, \ldots, \theta_{c-1}, b_1, \ldots, b_c) \mapsto (l_1, \ldots, l_k, \theta_1, \ldots, \theta_{k-1}, b_1, \ldots, b_k);$$

3. *for each $i \in B^+$,*
   $$H_i^+(l, \theta, b) = \sum_{j \neq i} \left[ \big( l_i l_j \cos(\theta_{ij}) \big)^2 + 3 \big( l_i l_j \cos(\theta_{ij}) \big) b_i + 3 b_i^2 \right]$$
   $$+ \left[ (1 - l_i^2)^2 - 3(1 - l_i^2) b_i + 3 b_i^2 \right],$$
   *where $\theta_{ij}$ denote the angle between $W_i$ and $W_j$.*

*Proof.* This follows from applying the same argument in the $c = 6$ case. $\qquad\square$

**Theorem I.2.** *Let $k \in \mathbb{Z}_{>4}$ and $s$ be the unique integer in the interval $[\frac{k}{4} - 1, \frac{k}{4})$. If a $k$-gon with coordinate*

$$l_1 = \cdots = l_k = x; \quad \theta_1 = \cdots = \theta_k = \frac{2\pi}{k}; \quad b_1 = \cdots = b_k = y$$

*for some $x > 0$ and $y < 0$ satisfying*

$$-x^2 \cos\left( s \cdot \frac{2\pi}{k} \right) \leq y$$

*is a critical point of $H(l, \theta, b)$ with constraint $\theta_1 + \cdots + \theta_k = 2\pi$ for $c = k$, then for any integer $0 \leq \sigma \leq c - k$, the $k^{\sigma+}$-gons defined in Section 3.2 are critical points of $H(l, \theta, b)$ with constraint $\theta_1 + \cdots + \theta_c = 2\pi$ for any $c > k$.*

*Proof.* This follows from applying the same argument in the $c = 6$ case. $\qquad\square$

**Remark I.1.** *In Lemma I.1, we show that the $5^+$-gon for $c = 6$ is minimally singular and compute the local learning coefficient. In general, we do not know whether the $k^{\sigma+}$-gons are minimally singular or not for $c > k$. However, given a $k^{\sigma+}$-gon, the method to check whether it is minimally singular or not is the same as the method used in Lemma I.1. We compute the Hessian of the TMS potential at each $k^{\sigma+}$-gon. Then check that the Hessian of the TMS potential is non-degenerate in the direction normal to the tangent space of $k^{\sigma+}$-gons. If this is the case, then we conclude that the $k^{\sigma+}$-gon is minimally singular by Morse-Bott lemma.*

## J  4-GONS

In this section we discuss 4-gons. The subtlety here is that the TMS potential is not smooth at 4-gons. In particular, the directional Hessian of the TMS potential at 4-gons depends on the direction.

### J.1  $c = 4$

#### J.1.1  STANDARD 4-GON

For $c = 4$, consider the (standard) 4-gon $(l^*, \theta^*, b^*)$, where

$$l^* = (1, 1, 1, 1), \theta^* = \left(\frac{\pi}{2}, \frac{\pi}{2}, \frac{\pi}{2}, \frac{\pi}{2}\right), b^* = (0, 0, 0, 0).$$

Since $H(l, \theta, b) \geq 0$ and $H(l^*, \theta^*, b^*) = 0$, we know the 4-gon is a global minimum. Let's work out the explicit form of $H$ around the 4-gon $(l^*, \theta^*, b^*)$. Since $\cos(\pi/2) = 0$, the 4-gon is at boundary of some chambers. Let $\mathcal{M} = \{M_{ij}\}$ be a chamber whose boundary contains the 4-gon $(l^*, \theta^*, b^*)$. We claim that each wedge $M_{ij}$ is either empty or contains one number. Assume, by contradiction, there is a $M_{ij}$ in $\mathcal{M}$ contains more than one number. Then we show that the 4-gon $(l^*, \theta^*, b^*)$ is not in the boundary of the chamber described by $\mathcal{M}$. Let $\epsilon_\theta \in \mathbb{R}_{>0}$ be such that

$$\pi - 2\epsilon_\theta > \frac{\pi}{2}.$$

Then for any $\theta, \theta' \in \left(\frac{\pi}{2} - \epsilon, \frac{\pi}{2} + \epsilon\right)$,

$$\theta + \theta' > \frac{\pi}{2} - \epsilon_\theta + \frac{\pi}{2} - \epsilon_\theta = \pi - 2\epsilon_\theta > \frac{\pi}{2}.$$

Let $\epsilon_l \in \mathbb{R}_{>0}$ be such that $1 - \epsilon_l > 0$. So $(l^*, \theta^*, b^*)$ has an open neighbourhood given by

$$(1 - \epsilon_l, 1 + \epsilon_l)^4 \times \left(\frac{\pi}{2} - \epsilon_\theta, \frac{\pi}{2} + \epsilon_\theta\right)^3 \times \mathbb{R}^4$$

which does not intersect the interior of the chamber described by $\mathcal{M}$. Thus, $(l^*, \theta^*, b^*)$ is not in the boundary of the chamber described by $\mathcal{M}$ when some of $M_{ij}$ in $\mathcal{M}$ contains more than one element. So each wedge $M_{ij}$ in $\mathcal{M}$ contains at most one element. Because of the permutation symmetry, there are three possible $\mathcal{M}$:

$$\mathcal{M}_1 = \boxed{\boxed{(1)}}, \quad \mathcal{M}_2 = \boxed{\begin{array}{c}\boxed{(1)}\\\boxed{(2)}\end{array}}, \quad \mathcal{M}_3 = \boxed{\begin{array}{c}\boxed{(1)}\\\boxed{(3)}\end{array}}, \quad \mathcal{M}_4 = \boxed{\begin{array}{c}\boxed{(1)}\\\boxed{(2)}\\\boxed{(3)}\end{array}}.$$

Since $-1^2 < 0$, $(l^*, \theta^*, b^*)$ has an open neighbourhood in which

$$-l_i^2 < b_i$$

for $i = 1, 2, 3, 4$. Since $\cos(\pi) = -1 < 0$, the 4-gon $(l^*, \theta^*, b^*)$ has an open neighbourhood in which for all $i = 1, 2, 3, 4$, if $b_i > 0$, then

$$l_i l_{i+2} \cos(\theta_i + \theta_{i+1}) < -b_i.$$

Recall the formula 17 for the local TMS potential in Section G. The 4-gon $(l^*, \theta^*, b^*)$ has an open neighbourhood in which the local TMS potential is

$$H(l, \theta, b) = \sum_{i=1}^{4} \delta(b_i \leq 0) H_i^-(l, \theta, b) + \delta(b_i > 0) H_i^+(l, \theta, b), \tag{18}$$

where

$$H_i^-(l, \theta, b) = \delta(T_{M_{(i-1)1}}^{(1)}) \frac{\left[l_{i-1} l_i \cos(\theta_{i-1}) + b_i\right]^3}{l_{i-1} l_i \cos(\theta_{i-1})} + \delta(T_{M_{i1}}^{(2)}) \frac{\left[l_i l_{i+1} \cos(\theta_i) + b_i\right]^3}{l_i l_{i+1} \cos(\theta_i)}$$
$$+ \left[(1 - l_i^2)^2 - 3(1 - l_i^2) b_i + 3 b_i^2 + \frac{b_i^3}{l_i^4} + \frac{b_i^3}{l_i^2}\right],$$

and

$$H_i^+(l, \theta, b) = \delta(S_{i(i+1)}) \left[ \frac{-b_i^3}{l_i l_{i+1} \cos(\theta_i)} \right]$$

$$+ \left(1 - \delta(S_{i(i+1)})\right) \left[ \left(l_i l_{i+1} \cos(\theta_i)\right)^2 + 3\left(l_i l_{i+1} \cos(\theta_i)\right)b_i + 3b_i^2 \right]$$

$$+ \left[ \frac{-b_i^3}{l_i l_{i+2} \cos(\theta_i + \theta_{i+1})} \right]$$

$$+ \delta(S_{i(i+3)}) \left[ \frac{-b_i^3}{l_i l_{i+3} \cos(\theta_{i+3})} \right]$$

$$+ \left(1 - \delta(S_{i(i+3)})\right) \left[ \left(l_i l_{i+3} \cos(\theta_{i+3})\right)^2 + 3\left(l_i l_{i+3} \cos(\theta_{i+3})\right)b_i + 3b_i^2 \right]$$

$$+ (1 - l_i^2)^2 - 3(1 - l_i^2)b_i + 3b_i^2$$

We checked that each term in $H_i^-(l, \theta, b)$ and $H_i^+(l, \theta, b)$ has gradient zero when approaching $(l^*, \theta^*, b^*)$ in the region specified by the indicator function associated with it. Thus, the TMS potential is differentiable at the $(l^*, \theta^*, b^*)$, and $(l^*, \theta^*, b^*)$ is a critical point. However, the TMS potential is not continuously differentiable twice, i.e. there are different directions in which directional Hessians are different. We checked that $(l^*, \theta^*, b^*)$ is minimally singular in each subspace with nonempty interior containing $(l^*, \theta^*, b^*)$ in the boundary. So we obtain a list $\{4, 4.5, 5, 5.5\}$ of local learning coefficient when approached from these different subspaces.

### J.1.2 $4^{\phi-}$-GON

Let $B^- \subset \{1, 2, 3, 4\}$ and $\phi = |B^-|$. Consider the $4^{\phi-}$-gon (Section 3.2) with coordinate

$$\theta^* : \theta_1 = \theta_2 = \theta_3 = \theta_4 = \frac{\pi}{2};$$

$$b^* : b_i < 0 \text{ if } i \in B^- \text{ and } b_i = 0 \text{ if } i \notin B^-;$$

$$l^* : 0 < l_i^2 < -b_i \text{ if } i \in B^- \text{ and } l_i = 1 \text{ if } i \notin B^-.$$

Since $\cos(\pi/2) = 0$, the $4^{\phi-}$-gon is on the boundary of some chambers. Let $\mathcal{M} = \{M_{ij}\}$ be a chamber whose boundary contains $4^{\phi-}$-gon. Using the same arguments in Section J.1.1, we know that each $M_{ij}$ is either empty or contains one number. Because of the $O(2)$-symmetry, there are four possible $\mathcal{M}$:

$$\mathcal{M}_1 = \boxed{\boxed{(1)}}, \quad \mathcal{M}_2 = \boxed{\begin{array}{c} (1) \\ \hline (2) \end{array}}, \quad \mathcal{M}_3 = \boxed{\begin{array}{c} (1) \\ \hline (3) \end{array}}, \quad \mathcal{M}_4 = \boxed{\begin{array}{c} (1) \\ \hline (2) \\ \hline (3) \end{array}}.$$

For $i \notin B^-$, we have $l_i^2 = 1 > 0 = b_i$. For $i \in B^-$, we have $-l_i^2 > b_i$, $-l_i l_{i+1} \cos(\theta_i) = 0 > b_i$, and $-l_{i-1} l_i \cos(\theta_{i-1}) = 0 > b_i$. Since $\cos(\pi) = -1 < 0$, the $4^{\phi-}$-gon has an open neighbourhood in which for all $i = 1, 2, 3, 4$, if $b_i > 0$, then

$$l_i l_{i_1} \cos(\theta_i + \theta_{i+1}) < b_i.$$

So the $4^{\phi-}$-gon has an open neighbourhood in which the local TMS potential is

$$H(l, \theta, b) = \sum_{i=1}^{4} \delta(b_i \leq 0) H_i^-(l, \theta, b) + \delta(b_i > 0) H_i^+(l, \theta, b), \tag{19}$$

where

1. if $i \notin B^-$, then

$$H_i^-(l, \theta, b) = \delta(T_{M_{(i-1)1}}^{(1)}) \frac{\left[l_{i-1} l_i \cos(\theta_{i-1}) + b_i\right]^3}{l_{i-1} l_i \cos(\theta_{i-1})} + \delta(T_{M_{i1}}^{(2)}) \frac{\left[l_i l_{i+1} \cos(\theta_i) + b_i\right]^3}{l_i l_{i+1} \cos(\theta_i)}$$

$$+ \left[(1 - l_i^2)^2 - 3(1 - l_i^2)b_i + 3b_i^2 + \frac{b_i^3}{l_i^4} + \frac{b_i^3}{l_i^2}\right],$$

and

$$H_i^+(l,\theta,b) = \delta(S_{i(i+1)})\left[\frac{-b_i^3}{l_i l_{i+1}\cos(\theta_i)}\right]$$

$$+ \left(1 - \delta(S_{i(i+1)})\right)\left[\left(l_i l_{i+1}\cos(\theta_i)\right)^2 + 3\left(l_i l_{i+1}\cos(\theta_i)\right)b_i + 3b_i^2\right]$$

$$+ \left[\frac{-b_i^3}{l_i l_{i+2}\cos(\theta_i+\theta_{i+1})}\right]$$

$$+ \delta(S_{i(i+3)})\left[\frac{-b_i^3}{l_i l_{i+3}\cos(\theta_{i+3})}\right]$$

$$+ \left(1 - \delta(S_{i(i+3)})\right)\left[\left(l_i l_{i+3}\cos(\theta_{i+3})\right)^2 + 3\left(l_i l_{i+3}\cos(\theta_{i+3})\right)b_i + 3b_i^2\right]$$

$$+ (1 - l_i^2)^2 - 3(1 - l_i^2)b_i + 3b_i^2;$$

2. if $i \in B^-$, then

$$H_i^-(l,\theta,b) = 1,$$

and

$$H_i^+(l,\theta,b) = 0.$$

If $\phi = 4$, then the $4^{\phi-}$-gon has an open neighbourhood in which the TMS potential is the zero function, hence it is a critical point with local learning coefficient 0. For $0 \le \phi \le 3$, we checked that each term in $H_i^-(l,\theta,b)$ and $H_i^+(l,\theta,b)$ has gradient zero when approaching the $4^{\phi-}$-gon in the region specified by the indicator function associated with it. Thus, the TMS potential is differentiable at the $4^{\phi-}$-gon, and the $4^{\phi-}$-gon is a critical point. However, the TMS potential is not continuously differentiable twice, i.e. there are different directions in which directional Hessians are different. We checked that the $4^{\phi-}$-gon is minimally singular in each subspace with nonempty interior containing the $4^{\phi-}$-gon in the boundary. So we obtain lists

$$\phi = 1 : \{3, 3.5, 4, 4.5\};$$
$$\phi = 2 : \{2, 2.5, 3, 3.5\} \text{ for } B^- = \{i, i+1\}, \text{ where } i = 1, 2, 3, 4,$$
$$\{2.5, 3, 3.5, 4\} \text{ for } B^- = \{i, i+2\}, \text{ where } i = 1, 2, 3, 4;$$
$$\phi = 3 : \{1, 1.5, 2, 2.5\};$$
$$\phi = 4 : \{0\}.$$

of local learning coefficient for each when approached from these different subspaces.

## J.2 $c > 4$

We analyse four typical 4-gons appearing as critical points of TMS potential in this section. In particular, we state their coordinates (hence computing their loss), and discuss their local learning coefficient. Because of the permutation symmetry, we may assume that 4-gons have $\theta$-coordinate

$$\theta_1 = \theta_2 = \theta_3 = \frac{\pi}{2}, \quad \theta_4 + \cdots + \theta_c = \frac{\pi}{2}.$$

So for given $l_1, \ldots, l_c$ and biases $b_1, \ldots, b_c$, there is a set of 4-gons given by $(\theta_4, \ldots, \theta_c)$ with $\theta_4 + \cdots + \theta_c = \pi/2$. As discussing in Appendix J.1, 4-gons are in the boundary of some chambers. Let $\mathcal{M} = \{M_{ij}\}$ be wedges describing a chamber containing 4-gons in its boundary. Let $\tau = (4, 5, \ldots, c)$. Applying the same arguments in Appendix H.3 and Appendix J.1, we know that $\mathcal{M}$ can only be a proper subset of $\{(1), (2), (3), \tau\}$.

Consider the standard 4-gons with coordinate

$$l_1 = l_2 = l_3 = l_4 = 1, \quad l_5 = \cdots = l_c = 0;$$
$$\theta_1 = \theta_2 = \theta_3 = \frac{\pi}{2}, \quad \theta_4 + \cdots + \theta_c = \frac{\pi}{2};$$
$$b_1 = b_2 = b_3 = b_4 = 0, \quad b_{k+1}, \ldots, b_c < 0.$$

**Theorem J.1.** *For a fixed $\mathcal{M}$, let $H^{(4)}$ denote the TMS potential in some neighbourhood of the standard 4-gon for $c = 4$ (Appendix J.1.1). Consider the projection*

$$\Pi_{c,4} : (l_1, \ldots, l_c, \theta_1, \ldots, \theta_{c-1}, b_1, \ldots, b_c) \mapsto (l_1, \ldots, l_4, \theta_1, \ldots, \theta_3, b_1, \ldots, b_4).$$

*The TMS potential in the chamber described by $\mathcal{M}$ is*

$$H(l, \theta, b) = H^{(k)} \circ \Pi_{c,4}(l, \theta, b) + (c - k).$$

*Proof.* This theorem follows from the same arguments used in Corollary H.1. $\square$

Thus, the standard 4-gons are critical points of the TMS potential. Moreover, we obtain a list $\{4, 4.5, 5, 5.5\}$ of local learning coefficient when approached from different chambers.

Let $B^- \subset \{1, 2, 3, 4\}$. Let $\phi = |B^-|$. Consider the $4^{\phi-}$-gons (Section 3.2) with coordinate

$$\theta^* : \quad \theta_1 = \theta_2 = \theta_3 = \frac{\pi}{2}, \quad \theta_4 + \cdots + \theta_c = \frac{\pi}{2}$$

$$b^* : \quad \text{for } i = 1, 2, 3, 4, \, b_i < 0 \text{ if } i \in B^- \text{ and } b_i = 0 \text{ if } i \notin B^-,$$
$$\text{for } j = 5, 6, \cdots, c, \, b_j < 0;$$

$$l^* : \quad \text{for } i = 1, 2, 3, 4, \, 0 < l_i^2 < -b_i \text{ if } i \in B^- \text{ and } l_i = 1 \text{ if } i \notin B^-,$$
$$\text{for } j = 5, 6, \ldots, c, \, l_j = 0.$$

**Theorem J.2.** *For a fixed $\mathcal{M}$, let $H^{(4,-)}$ denote the TMS potential in some neighbourhood of $4^{\phi-}$-gon for $c = 4$ (Appendix J.1.2). Consider the projection*

$$\Pi_{c,4} : (l_1, \ldots, l_c, \theta_1, \ldots, \theta_{c-1}, b_1, \ldots, b_c) \mapsto (l_1, \ldots, l_4, \theta_1, \ldots, \theta_3, b_1, \ldots, b_4).$$

*The TMS potential in the chamber described by $\mathcal{M}$ is*

$$H(l, \theta, b) = H^{(k,-)} \circ \Pi_{c,4}(l, \theta, b) + (c - k).$$

*Proof.* This theorem follows from the same arguments used in Corollary H.1. $\square$

Thus, the $4^{\phi-}$-gons are critical points of the TMS potential. Moreover, we obtain lists

$$\phi = 1 : \{3, 3.5, 4, 4.5\};$$
$$\phi = 2 : \{2, 2.5, 3, 3.5\} \text{ for } B^- = \{i, i+1\}, \text{ where } i = 1, 2, 3, 4,$$
$$\{2.5, 3, 3.5, 4\} \text{ for } B^- = \{i, i+2\}, \text{ where } i = 1, 2, 3, 4;$$
$$\phi = 3 : \{1, 1.5, 2, 2.5\};$$
$$\phi = 4 : \{0\}.$$

of local learning coefficient when approached from different chambers.

Now, we discuss when $\sigma > 0$. We show that all $4^{\sigma+,\phi-}$-gons are critical points for $\sigma = 1, 2$ and $0 \leq \phi \leq 4$. However, we do not know whether these critical points are minimally singular or not. Let $B^+ \subset \{5, 6, \ldots, c\}$. Let $\sigma = |B^+|$. Consider the $4^{\sigma+}$-gons (Section 3.2) with coordinate

$$l^* : \quad l_1 = l_2 = l_3 = l_4 = 1 > 0, \, l_{k+1} = \cdots = l_c = 0;$$

$$\theta^* : \quad \theta_1 = \theta_2 = \theta_3 = \frac{\pi}{2}, \, \theta_4 + \cdots + \theta_c = \frac{\pi}{2};$$

$$b^* : \quad b_1 = b_2 = b_3 = b_4 = 0,$$

$$\text{for } i = 5, 6, \ldots, c, \, b_i < 0 \text{ if } i \notin B^+ \text{ and } b_i = \frac{1}{2c} \text{ if } i \in B^+.$$

**Theorem J.3.** *For a fixed $\mathcal{M}$, let $H^{(4)}$ denote the TMS potential in some neighbourhood of the standard 4-gon for $c = 4$ (Appendix J.1.1). Consider the projection*

$$\Pi_{c,4} : (l_1, \ldots, l_c, \theta_1, \ldots, \theta_{c-1}, b_1, \ldots, b_c) \mapsto (l_1, \ldots, l_4, \theta_1, \ldots, \theta_3, b_1, \ldots, b_4).$$

*The TMS potential in the chamber described by $\mathcal{M}$ is*

$$H(l, \theta, b) = H^{(4)} \circ \Pi_{c,4}(l, \theta, b) + \big(c - (4 + \sigma)\big) + \sum_{i \in B^+} H_i^+(l, \theta, b),$$

*where for each $i \in B^+$,*

$$H_i^+(l, \theta, b) = \sum_{j \neq i} \left[ \big(l_i l_j \cos(\theta_{ij})\big)^2 + 3\big(l_i l_j \cos(\theta_{ij})\big)b_i + 3b_i^2 \right]$$
$$+ \left[ (1 - l_i^2)^2 - 3(1 - l_i^2)b_i + 3b_i^2 \right],$$

*where $\theta_{ij}$ denote the angle between $W_i$ and $W_j$.*

*Proof.* This follows from the same arguments in Theorem I.1. □

Thus, the $4^{\sigma+}$-gons are critical points of the TMS potential. However, we do not know whether the $4^{\sigma+}$-gons are minimally singular or not. In $c = 6$ case, we computed the directional Hessian at $4^{\sigma+}$-gon for $\sigma = 1, 2$ along various directions. For directions we have analysed, the eigenvectors of the Hessian with negative eigenvalues point in directions that escapes the set where the current form of the local TMS potential applies. By counting the number of zero eigenvalues, we obtain a list $\{6, 6.5, 7, 7.5\}$ of upper bounds for the local learning coefficients for $4^{2+}$-gons, and a list of $\{5, 5.5, 6, 6.5\}$ of upper bounds of local learning coefficients for $4^+$-gons.

Let $B^- \subset \{1, 2, 3, 4\}$ and $B^+ \subset \{5, 6, \cdots, c\}$. Consider the $4^{\sigma+,\phi-}$-gon (Section 3.2) with coordinate

$$\begin{aligned}
\theta^* : \quad & \theta_1 = \theta_2 = \theta_3 = \frac{\pi}{2}, \quad \theta_4 + \cdots + \theta_c = \frac{\pi}{2} \\
b^* : \quad & \text{for } i = 1, 2, 3, 4, \, b_i < 0 \text{ if } i \in B^- \text{ and } b_i = 0 \text{ if } i \notin B^-, \\
& \text{for } j = 5, 6, \cdots, c, \, b_j < 0 \text{ if } j \notin B^+ \text{ and } b_j = \frac{1}{2c} \text{ if } j \in B^+; \\
l^* : \quad & \text{for } i = 1, 2, 3, 4, \, 0 < l_i^2 < -b_i \text{ if } i \in B^- \text{ and } l_i = 1 \text{ if } i \notin B^-, \\
& \text{for } j = 5, 6, \ldots, c, \, l_j = 0.
\end{aligned}$$

**Theorem J.4.** *For a fixed $\mathcal{M}$, let $H^{(4,-)}$ denote the TMS potential in some neighbourhood of the $4^{\phi-}$-gon for $c = 4$ (Appendix J.1.2). Consider the projection*

$$\Pi_{c,4} : (l_1, \ldots, l_c, \theta_1, \ldots, \theta_{c-1}, b_1, \ldots, b_c) \mapsto (l_1, \ldots, l_4, \theta_1, \ldots, \theta_3, b_1, \ldots, b_4).$$

*The TMS potential in the chamber described by $\mathcal{M}$ is*

$$H(l, \theta, b) = H^{(4,-)} \circ \Pi_{c,4}(l, \theta, b) + \big(c - (4 + \sigma)\big) + \sum_{i \in B^+} H_i^+(l, \theta, b),$$

*where for each $i \in B^+$,*

$$H_i^+(l, \theta, b) = \sum_{j \neq i} \left[ \big(l_i l_j \cos(\theta_{ij})\big)^2 + 3\big(l_i l_j \cos(\theta_{ij})\big)b_i + 3b_i^2 \right]$$
$$+ \left[ (1 - l_i^2)^2 - 3(1 - l_i^2)b_i + 3b_i^2 \right],$$

*where $\theta_{ij}$ denote the angle between $W_i$ and $W_j$.*

*Proof.* This follows from the same arguments in Theorem I.1. □

Thus, the $4^{\sigma+,\phi-}$-gons are critical points of the TMS potential. However, we do not know whether the $4^{\sigma+,\phi-}$-gons are minimally singular or not.

**Remark J.1.** *For $\sigma > 0$, we do not have a precise value for the local learning coefficient. In Appendix K, we provide estimation of local learning coefficients for $4^{\sigma+,\phi-}$-gons.*

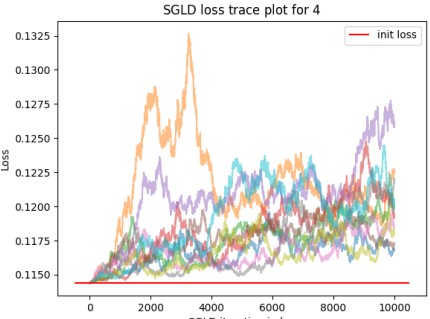 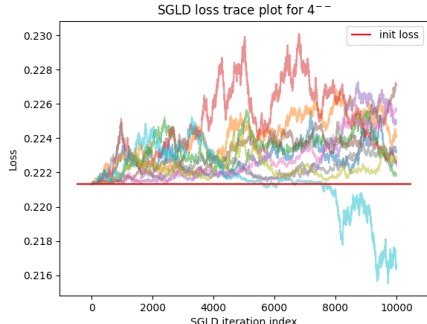

Figure K.1: Loss trace plots for samples used for $\hat{\lambda}$ produced via SGLD sampling. The left plot show SGLD chains that are all "healthy" and the right plot shows a trajectory that escapes the phase of the initialising critical point (loss indicated by red horizontal line) to another phase with lower loss. To obtain the estimates $\hat{\lambda}$ listed in Table K.1, such unhealthy chains are removed from consideration.

## K    DETAILS OF LOCAL LEARNING COEFFICIENT ESTIMATION

In this section, we discuss technical details and caveats about the values of the local learning coefficient estimates $\hat{\lambda}$ given throughout the paper. It was claimed in Lau et al. (2023) that the $\hat{\lambda}$ algorithm is valid for comparing or ordering critical points by their level of degeneracy. Obtaining the correct local learning coefficient can prove challenging. For TMS, we find that

- The ordering of $\hat{\lambda}$ for different critical points lines up with the theoretical prediction.

- For critical points with low loss such as 6, 5 and $5^+$-gon depicted in Figure A.2, the $\hat{\lambda}$ values are close to theoretically derived values.

- However, for critical points with higher loss, mis-configured SGLD step size used in the algorithm can causing the sample path itself to undergo a phase transition to a lower loss state. See the diagnostic trace plot on the right of Figure K.1 for an example where SGLD trajectory drop to a different phase. This is the reason for *negative* $\hat{\lambda}$ values shown in Figure 3, in which we opted to use a uniform set of SGLD hyperparameters since we cannot a priori predict which critical point an SGD trajectory will visit.

- Lowering SGLD step size can ameliorate this issue, at the cost of increasing the required number of sampling steps needed.

Table K.1 shows a set of $\hat{\lambda}$ values computed using bespoke SGLD step size (explained below) for each group of 3 critical points with similar loss (again c.f. Figure A.2. Specifically, we take the dataset size $n = 5000$, and SGLD hyperparameters $\gamma = 0.1$, number of steps $= 10000$.

Furthermore, for each critical point, we run 10 independent SGLD chains and discard any chain where more than 5% of the samples have loss values that are lower than the critical point itself. The $\hat{\lambda}$ estimate and the standard deviation are then calculated from the remaining chains. The SGLD step size is manually chosen so that the majority of chains in each group passes the test above.

| Critical point | Estimated $\hat{\lambda}$ (std) | SGLD step size |
|---|---|---|
| $4^{----}$ | 0.000 (0.00) | 0.0000005 |
| $4^{+----}$ | 0.540 (0.16) | 0.0000005 |
| $4^{++----}$ | 0.998 (0.54) | 0.0000005 |
| $4^{---}$ | 1.024 (0.74) | 0.000001 |
| $4^{+---}$ | 1.619 (0.89) | 0.000001 |
| $4^{++---}$ | 1.899 (0.76) | 0.000001 |
| $4^{--}$ | 1.689 (0.88) | 0.000001 |
| $4^{+--}$ | 2.096 (1.00) | 0.000001 |
| $4^{++--}$ | 2.597 (0.88) | 0.000001 |
| $4^{-}$ | 2.991 (0.35) | 0.000005 |
| $4^{+-}$ | 3.393 (0.65) | 0.000005 |
| $4^{++-}$ | 4.097 (0.65) | 0.000005 |
| 4 | 5.297 (0.04) | 0.00001 |
| $4^{+}$ | 5.761 (1.53) | 0.00001 |
| $4^{++}$ | 6.203 (0.99) | 0.00001 |
| 5 | 7.705 (0.85) | 0.00005 |
| $5^{+}$ | 9.906 (1.27) | 0.00005 |
| 6 | 9.027 (0.59) | 0.00005 |

Table K.1: $\hat{\lambda}$ for known critical points in $r = 2, c = 6$, their standard deviation across viable SGLD chains and the SGLD step size used.

