# OpenReview forum: "Dynamical versus Bayesian Phase Transitions in a Toy Model of Superposition"
_ICLR.cc/2024/Conference — Submitted to ICLR 2024_

### Official Review · Reviewer_K4kT · 2023-10-31

**Soundness:** 2 fair
**Presentation:** 1 poor
**Contribution:** 2 fair
**Rating:** 3
**Confidence:** 3

**Summary:**

The paper examines a simple auto-encoder model with just two hidden units, applied to extremely sparse data where only one element of the input vector is non-zero. The primary contribution lies in providing a complete analytical characterisation of the critical points of the population loss in this specific setup. This characterization is then leveraged to make arguments about the existence of distinct phases within the model's behavior as a function of the number of samples available. Furthermore, it serves as a guide for exploring the transitions between these different phases during the model's dynamics.

**Strengths:**

While the model under consideration has already been proposed in Elhage et al., 2022, the authors present a complete, explicit and exact characterisation of the critical points of the population loss, which is in quite rare. This is then used to compute a rough approximation of the free energy of the network near the critical points in order to give some qualitative indication on the phase of the model under study.

**Weaknesses:**

The presentation in this paper is extremely ineffective. The classification of critical points is crucial to understanding the paper, yet a complete and accurate description is relegated to the appendix. Similarly, there are many k-gon diagrams in the main text with no direct explanation on how to read them. I believe the paper would be much easier to read with a compact set of definitions clearly stated in the main text.

While I personally believe that studying simple models can be interesting, I struggle to see any practical interest in studying the model the authors propose, and I am not sure how feasible it can be to relax the many underlying assumptions: having more than 2 hidden units makes the structure of the critical points significantly more complicated, and changing the data model would make the whole theory collapse.

Even accepting all the limitations above, I don't think the authors provide an adequate quantitative study of the phase transitions of this model beyond some general remarks. I would argue that the analytical results on the loss play no role in the discussion of the dynamical transition picture, as the energy levels could have been identified from the experiments, and knowing the exact values has no importance in what is presented.

**Questions:**

1) Would it be possible to also study analytically whether the critical points are (local) minima? This is particularly interesting for energy levels $4^{+,3-}$, $4^{2+,3-}$, $4^{+,2-}$, $4^{2+,2-}$, $4^{+,-}$, $4^{2+,-}$, $4^{2+}$ in Figure 3, as it seems there are no plateaus associated to those levels.

2) In section 4.1 you define a transition as the number of samples for which the phase changes. The simulation in Figure 2 seems to suggest that there is also a number of samples for which a certain phase has proportion zero. Is it the case? If so can you characterise it?

3) In Figure 3 it would seem you are doing up to 1/4 million SGD steps. Is it a typo?

3) Also in Figure 3, I would expect that for large enough times SGD escapes the plateaus and jumps to a lower energy level. I think there is already an example of this in line 5 of the right panel. Is it possible to characterise this phenomenon?

4) What would Figure 3 look like if SGD was randomly initialised?

---

> ### Author Response · Authors · 2023-11-18
>
> It seems that the way we have written the introduction may have led to a misunderstanding. While it is listed as the first contribution of the paper, we do *not* believe that providing a classification of the critical points for the high-sparsity TMS potential is the *primary* contribution of the paper. Indeed, it's unclear how much generalisable insight such a classification would provide, since TMS is a very toy system and the classification is quite laborious even in this case.
>
> The primary contribution of the paper lies in proving empirically that Bayesian phase transitions occur, that they are well-described by SLT, and that there are interesting (but not yet completely established) links between these Bayesian transitions and the dynamical transitions that many in the deep learning community have noted.
>
> To respond to some of your individual points:
>
> - You highlighted the importance of detailing the classification of critical points within the main text rather than relegating it to the appendices. Please note the definition of $k$-gon critical points is already presented in Section 3.2 of the main text. Although Figures A.1 to A.3 in Appendix A enhance the visualization of these $k$-gons, we regret that space constraints prevent their inclusion in the main text.
> - It is true that changing the system would invalidate the current classification. However, as remarked above, the classification is *not* the primary contribution of the paper: it is one piece of the puzzle, that allows us to establish the other contributions listed above as the actually important contributions (about phase transitions). So the toy model we study is interesting not because the classification of critical points generalises, but because the Bayesian theory of phase transitions *may* generalise to larger models. This expectation is very reasonable theoretically, and the experiments in this paper are the first step towards verifying this theory empirically.
> - You state that "I don't think the authors provide an adequate quantitative study of the phase transitions of this model beyond some general remarks". This paper contains, by far, the most detailed study so far in the literature of phases and phase transitions in any neural network and goes far beyond general remarks, providing a classification (both theoretically and empirically backed) of phases and phase transitions.
> - Re: the role of the analytical results in the dynamical transition picture. There are ample evidence that properties of the critical points other than their energy level, play an important role. Analytical results about the geometrical properties like the learning coefficients can gives us a more detailed picture. For example, it leads to the curious observation that as loss drops during SGD training, there is a corresponding increase in complexity.
>
> Regarding your questions:
> - Re: whether other critical points are local minima. We can calculate Hessians at each of them and see that they contain no negative eigenvalues, but it is more difficult to prove they are local minima and this has not been done in all cases. As the paper is already 60 pages we do not include all these calculations.
> - Re: phases with zero proportion. The question seems to be about phases that aren't present in Figure 2, such as $4^{----}$. Theoretically, the posterior has non-zero mass at all phases for any $n$. The posterior mass of each phase $W_\alpha$ is roughly given by $e^{-F_n(W_\alpha)} = e^{-n L_n(W_\alpha) + \lambda_\alpha \log n + c_\alpha}$. Thus, it is clear that high energy phases are exponentially suppressed and won't be dominant in the regime of $n$ we are probing. Figure F.1 contains the occupancy plots for more of the classified critical points, these are omitted in the main text for simplicity.
> - Re: number of SGD steps. This is not a typo. We run 4500 epoch in our SGD experiments, each with 50 SGD steps (n / batch size = 1000 / 20) per epoch.
> - Re: SGD escaping plateaus. We have not studied this, it does not seem to be common in the trajectories we studied.
> - Re: Figure 3 and SGD initialisation. Different initialisations discover different phases with different probabilities. Studying this distribution was not the aim of the paper. Our aim was to find as many examples of phases and phase transitions as possible. The posterior distribution at low $n$ serves this purpose well. It has more probability mass near critical points of the energy function but is also diffused. When we initialise instead using a Gaussian at the origin, the plateaus in Figure 3 look roughly similar, but less phases are discovered (e.g. $5^+, 6, 4^{---}$ are not discovered). Our current view is that this is not very fundamental, and we would see similar results with either choice if enough SGD trajectories were sampled, but it would be interesting to study further.

---

> > ### Comment · Reviewer_K4kT · 2023-11-22
> >
> > Thanks for your clarification.
> > I still have some doubts on this work.
> >
> > As stated in the review, I do think that there is some interesting work in this paper, but I am not convinced the phase picture of the model at hand is adequately detailed.
> > In particular, showing which minimum dominates the posterior is typically not enough to ensure it will be algorithmically easy to reach such minimum. On the contrary, in many other models [1,2,3] it's possible to show that it is in fact a hard task to escape a sub-optimal minimum. Studying such effects is for me crucial to understanding phase transitions.
> >
> > For these reasons, as well as for the overall lack of clarity of the paper, I choose to not change my score.
> >
> > [1]: "Disordered systems insights on computational hardness", Gamarnik, David and Moore, Cristopher and Zdeborová, Lenka, Journal of Statistical Mechanics: Theory and Experiment 2022
> >
> > [2]: "Passed & Spurious: Descent Algorithms and Local Minima in Spiked Matrix-Tensor Models", Stefano Sarao Mannelli and Florent Krzakala and Pierfrancesco Urbani and Lenka Zdeborová, ICML 2019
> >
> > [3]: "The committee machine: computational to statistical gaps in learning a two-layers neural network", Aubin, Benjamin and Maillard, Antoine and Barbier, Jean and Krzakala, Florent and Macris, Nicolas and Zdeborová, Lenka, Journal of Statistical Mechanics: Theory and Experiment, 2019

---

### Official Review · Reviewer_ajyb · 2023-11-01

**Soundness:** 3 good
**Presentation:** 3 good
**Contribution:** 2 fair
**Rating:** 5
**Confidence:** 2

**Summary:**

This paper investigates phase transitions in a Toy Model of Superposition using both dynamical and Bayesian methods. It uncovers critical points in the form of regular k-gons and demonstrates their impact on learning processes. The findings support the idea of a sequential learning mechanism in Stochastic Gradient Descent. The study enhances our understanding of phase transitions in deep learning.

**Strengths:**

The paper demonstrates originality by focusing on the analysis of phase transitions in a small-scale learning task. This approach offers a fresh perspective on learning processes, especially in a Toy Model of Superposition (TMS). The study takes an innovative angle by investigating the role of critical points and sequential learning mechanisms in understanding phase transitions.

**Weaknesses:**

One significant weakness is the limited exploration of how the results obtained from this small-scale learning task can be applied to larger-scale and more practical problems. The paper does not clearly articulate how the insights gained from this Toy Model of Superposition (TMS) could be extended to real-world, complex learning scenarios. Providing a bridge between the toy model and practical applications is essential for enhancing the paper's impact.

**Questions:**

See Weaknesses

---

> ### Author Response · Authors · 2023-11-17
>
> Regarding the link between the small-scale learning task studied in our paper and larger-scale and more practical problems, our response is as follows. The paper makes several main claims, including:
>
> - (A) Bayesian phase transitions are theoretically predicted and actually occur in TMS
> - (B) Dynamical transitions occur in TMS, and
> - (C) There is some relation between Bayesian and dynamical transitions
> - (D) The local learning coefficient is a powerful tool for reasoning about phase transitions
>
> In both (A) and (B) we don't see the scope of our results as really being limited to small-scale systems. Since (A) is a general phenomena of Bayesian learning once the general principles are seen to apply in TMS, it is reasonable to expect such transitions to occur in larger systems, since the larger the system the *more* we expect its behaviour to be governed by general laws. Dynamical transitions have already been observed in larger scale systems, including language models with billions of parameters in the work on Induction Heads and In-Context Learning from Anthropic.
>
> We agree that in (C) the current evidence linking Bayesian and dynamical transitions is restricted to one toy system (TMS). It will require follow-up work to assess whether this can be extended to a general principle for working with larger-scale systems.
>
> However in the case of (D) we believe our work is *already* informing people's plans about how to think about phase transitions in real-world systems and so we feel confident asserting that our approach will have a substantial impact. For example there has been some preliminary work done already looking at modular addition and grokking using the learning coefficient.

---

### Official Review · Reviewer_tD3S · 2023-11-04

**Soundness:** 3 good
**Presentation:** 2 fair
**Contribution:** 3 good
**Rating:** 6
**Confidence:** 3

**Summary:**

If I understand well, they characterize some critical points of the likelihood function for the TMS in the high sparsity limit. They then use SLT to get approximate theoretical predictions on the phase transitions (for the sample size $n$ large enough) that the Bayesian posterior undergoes when $n$ increases. They confirm these predictions experimentally (Figure 2).

Then it is stated that the phase transition of the Bayesian posterior is a good indicator of the dynamical transitions (the Bayesian Antecedent Hypothesis)

**Strengths:**

The ideas are interesting and lead to intriguing experimental results. The paper seems to introduce novel findings on TMS in the high sparsity limit through the characterization of critical points.

**Weaknesses:**

The paper contains a lot of information and there seems to be a lot of prerequisites for understanding it. It is also difficult to get an idea of the context. It would have been nice to have more context on why this problem (TMS) matters and what is already known about its critical points.

The theoretical results could be made more formal; for example, the authors emphasize that applying SLT in Section 4.1 is a strong point of the paper compared to the usual use of the Laplace approximation for example in (Wei et al., 2022b; Lau et al., 2023). However, the paper does not discuss whether the assumptions of SLT hold in their setting.

I would also find very interesting to have a theoretical discussion of the Bayesian Antecedent Hypothesis.

**Questions:**

Is it true that the conditions of (Watanabe, 2009) hold for the local free energy formula (9) for the Gaussian prior considered in the experiments?

---

> ### Author Response · Authors · 2023-11-16
>
> Regarding the context for the paper, the original work by Elhage et al motivates the Toy Model of Superposition as a simple model in which to study the phenomena of superposition, which they view as a key obstacle to mechanistic interpretability. Our motivations are different: we are interested in phase transitions, both in the context of Bayesian statistics (where the phenomena is understudied and lacks good examples) and in deep learning (where somewhat mysterious "phase transitions" over training, what we call dynamical transitions, are attracting increasing interest).
>
> Our theoretical and experimental results give strong reason to believe the Toy Model of Superposition is also a good Toy Model of Phase Transitions, in both of these senses (Bayesian and dynamical).
>
> Regarding what is known about the critical points in TMS. As far as we know nothing has been published on this that goes beyond the qualitative results in the original paper.
>
> Re: making the theoretical results more formal. This is a good question, but we view it as out of scope for the present work. The aim of this paper is to show that Bayesian phase transitions exist, and to explain the theoretical context in which they make sense. To prove that the technical hypotheses of SLT apply to this example, and to further clarify the exact mathematical nature of Bayesian phase transitions, is quite difficult and may be the subject of future work. We agree it is important to clarify the status of hypotheses such as "relative finite variance" (necessary to apply the results of Watanabe's 2018 book) in neural networks.
>
> Re: the BAH. The Bayesian Antecedent Hypothesis is, currently, just that - a hypothesis. This paper provides preliminary empirical evidence supporting the BAH. We are actively working on a rigorous theoretical foundation for the BAH, but it is beyond the scope of this paper.
>
> Re: whether the conditions of Watanabe 2009 apply to the Gaussian prior considered in experiments. We don't quite understand the specific role of the prior in this question, could you please elaborate?

---

### Official Review · Reviewer_DVUo · 2023-11-07

**Soundness:** 4 excellent
**Presentation:** 3 good
**Contribution:** 4 excellent
**Rating:** 8
**Confidence:** 2

**Summary:**

This work analyses the empirically observed phenomenon that, during stochastic-gradient-decent (SGD) based training of neural networks, there appear to be "phase transitions" in which the loss suddenly drops rapidly in between longer "plateaus" of relatively constant loss; and that these phase transitions typically go from parameter-space regions of lower complexity and lower loss to regions of higher complexity and higher loss. To characterise this phenomenon, the authors focus on a simple but analytically tractable model.

Specifically, for certain numbers of hidden and feature dimensions in this model, they classify the critical points in the parameter space. They then characterise Bayesian phase transitions in this model as training sample sizes at which the much of the posterior probability mass shifts from one region to another. Finally, they verify that critical points with lower loss are typically associated with higher weight-configuration complexity. Taken together, these findings lead the authors to conjecture that "phase transitions" observed in SGD training have underlying them a Bayesian phase transition.

**Strengths:**

**Originality**
To my knowledge the main results in this work are novel. As a side note: I am not familiar with the literature in this area. In particular, especially in the characterisation of Bayesian phase transitions in Section, it is hard for me to tell which results are novel and which are due to the various works of Sumio Watanabe. However, even if this characterisation is largely based on existing work, I would still argue that the work is sufficiently novel for publication in ICLR.

**Quality**
I also believe that this work is rigorous enough for publication in ICLR;

**Clarity**
I think the writing and presentation is overall reasonably clear. But please see some suggestions for improvement in the "weaknesses" below.

**Significance**
This work sheds some much needed light on the phase transitions observed in SGD training (at least in a simple, tractable scenario).

**Weaknesses:**

To make this work more self contained, it would be good to explain/define concepts from Singular Learning Theory at the beginning. In particular, I really missed a clear definition of the "local learning coefficient" which is central to this work, and therefore found this work quite difficult to read without going through [1] first which gives a definition of this concept.

Additionally, the fact that the guide for understanding some aspects of, e.g., Figure 1 is only found in Appendix B (rather than the main paper) is not optimal.

[1] Lau, E., Murfet, D., & Wei, S. (2023). Quantifying degeneracy in singular models via the learning coefficient. arXiv preprint arXiv:2308.12108.

**Questions:**

Do you have any intuition about whether/how the dynamic phase transitions would be affected if we used an optimiser with momentum instead of SGD?

---

> ### Author Response · Authors · 2023-11-16
>
> Thanks for your comments. Regarding the originality of results in Section 4 on Bayesian phase transitions: this is the first time this kind of treatment of Bayesian phase transitions with respect to the number of samples, with a division of the parameter space into phases, has appeared in the literature. We cite Watanabe's work because he has written about other kinds of phase transitions in Bayesian learning (where a hyperparameter changes in the prior or the true distribution). As far as we are aware the experiments in this paper are the first time clear phase transitions of the kind we are talking about have been shown to exist.
>
> Re: Weaknesses. We have uploaded a revision where the caption of Figure 1 is expanded to include all the details necessary to read it, and Section 3.3 has been expanded slightly to include some more context on the local learning coefficient. We couldn't find a way to add in an introduction to SLT without pushing important content, currently in the main text, into Appendices, so while we agree an introduction would be valuable we've settled for the status quo of referring the reader to other papers.
>
> Re: Question about momentum. Yes this is a natural question. I think you have in mind that if one uses an optimiser with momentum, it will tend to spend less time stuck near critical points and therefore may collapse the "plateaus" that we are displaying in the context of dynamical transitions. I expect this is correct but the plateaus will still be present. We're running the experiments now and will add a comment when we know more.

---

> > ### Author Response · Authors · 2023-11-23
> >
> > Re: momentum. Preliminary experiments suggest that with momentum = 0.9 trajectories continue to reliably plateau at a loss value associated with one of the critical points we have identified. However, the use of momentum substantially reduces the fraction of sgd trajectories which involve dynamical phase transitions from one phase to another. These transitions still occur, but substantially less often. This observation comes from a single set of experiments holding all the other training hyperparameters constant and adding in momentum. A thorough investigation of how different optimisers affect the story should involve exploring different hyperparameters and would be beyond the scope of this comment.

---

### Author Response · Authors · 2023-11-16
**Revised presentation and connection to real models**

We express our gratitude to the reviewers for providing valuable insights to enhance the clarity of our presentation. Both DVUo and K4kT recommended a more thorough explanation of Figure 1 within the main text. In line with this feedback, we have relocated the guide on interpreting Figure 1 from Appendix B to an earlier section.

Additionally, K4kT highlighted the importance of detailing the classification of critical points within the main text rather than relegating it to the appendices. Please note the definition of k-gon critical points is already presented in Section 3.2 of the main text. Although Figures A.1 to A.3 in Appendix A enhance the visualization of these k-gons, we regret that space constraints prevent their inclusion in the main text.

We thank tD3S for asking for clarification about how studying TMS is supposed to inform us about larger models. Although the model under examination may appear toy-like in nature, it's important to emphasize that our methodology is highly scalable and applicable to larger neural networks. Specifically, the efficacy of the local learning coefficient as an estimator for local degeneracy extends well beyond the confines of the TMS model.

Now, the question arises: why did we choose to focus on the TMS initially? While singular learning theory is promising as a theoretical framework for understanding deep learning, many of its most interesting predictions have not yet been verified experimentally. In our view one of the most surprising predictions of SLT is the existence of phase transitions in the sample size, and we were excited to find in TMS a toy model for studying these transitions. Having seen that the free energy formula works in this example, we have much more confidence that such phase transitions exist in larger systems. Obviously this needs to be checked in follow-up work, but it is clearly reasonable to study the phenomena first in smaller models where we are able to theoretically calculate the local learning coefficient.

---

### Meta-Review · Area_Chair_JwWr · 2023-12-05

**Metareview:**

This paper presents a toy model for which the critical points of the population landscape were identified and classified, this is a nice result. The part of the paper where transitions between the critical points are described is based on a rather uncontrolled approximation. It is not clear whether this would generalize to more realistic models. A big problem is the misuse of the concept of a phase transition that has been established in physics for decades and is related to a non-analyticity of a thermodynamic potential in the large size limit. Here the concept is used in another way that is different from the canonical usage in physics and is not properly defined in the paper.

**Justification For Why Not Higher Score:**

The presentation of the paper and the claimed connections to phase transitions are just wrong; the paper should not be published in this form.

**Justification For Why Not Lower Score:**

N/A

---

### Decision · Program_Chairs · 2024-01-16

Reject